# CART-Based Synthetic Tabular Data Generation for Imbalanced Regression

## Abstract

Handling imbalanced target distributions in regression tasks remains a significant challenge in tabular data settings where the underrepresentation of relevant regions can hinder model performance. Among data-level solutions, some proposals, such as random sampling and SMOTE-based approaches, propose adapting classification techniques to regression tasks. However, these methods typically rely on crisp, artificial thresholds over the target variable, a limitation inherited from classification settings that can introduce arbitrariness, often leading to non-intuitive and potentially misleading problem formulations. While recent generative models, such as GANs and VAEs, provide flexible sample synthesis, they come with high computational costs and limited interpretability. In this study, we propose adapting an existing CART-based synthetic data generation method, tailoring it for imbalanced regression. The new method integrates relevance and density-based mechanisms to guide sampling in sparse regions of the target space and employs a threshold-free, feature-driven generation process, making it suitable for heterogeneous data types, complex inter-correlations, and intricate column-wise distributions. Our experimental study focuses on the prediction of extreme target values across benchmark datasets. The results indicate that the proposed method is competitive with other resampling and generative strategies in terms of performance, while offering faster execution and greater transparency, providing the best trade-off between both aspects. These results highlight the method's potential as a transparent, scalable data-level strategy for improving regression models in imbalanced domains.

## 1 Introduction

Research on imbalanced domain learning has primarily focused on classification tasks, yet similar challenges arise in regression, where continuous target variables lead to difficulties in identifying and predicting rare but critical outcomes (Branco et al., 2016). Standard regression models assume uniform importance across target values and rely on metrics like Mean Squared Error (MSE), which favor average predictions and neglect extremes. This limitation is particularly relevant in domains such as extreme weather forecasting (Schultz et al., 2021), sea surface temperature prediction (Alerskans et al., 2022), oncology drug response (Lenhof et al., 2022), and financial anomaly detection.

Despite two decades of research on imbalanced learning (Ribeiro & Moniz, 2020; Branco et al., 2016; Fernández et al., 2018; López et al., 2013; Krawczyk, 2016; He & Ma, 2013), most efforts remain centered on classification, with imbalanced regression receiving limited attention. In this field, data-level approaches are widely used to mitigate imbalance by modifying data distributions during preprocessing, ensuring compatibility with standard algorithms (Branco et al., 2016).

In this paper, we propose an adaptation of a CART-based data augmentation method (Panagiotou et al., 2024) tailored for imbalanced regression tasks. Unlike existing techniques (Branco et al., 2017; Torgo et al., 2015; Branco et al., 2019), our approach eliminates the need for user-defined thresholds within the target domain.

The remainder of this paper is organized as follows: Section 2 reviews relevant literature, Section 3 describes our proposed method, Section 4 presents experimental results, and Section 5 concludes the paper.

## 2 PRELIMINARIES AND RELATED WORK

### 2.1 IMBALANCED REGRESSION

Supervised learning aims to approximate an unknown function $Y = f(X_1, \ldots, X_p)$ using a model $h$ trained on data $D = \{\langle x_i, y_i \rangle\}_{i=1}^n$, with tasks framed as classification or regression depending on whether $Y$ is discrete or continuous, and optimized through criteria such as error rate or squared error (Branco et al., 2016; Ribeiro & Moniz, 2020). In domains like environmental sciences, finance, and healthcare, predicting extreme values is critical but hindered by their underrepresentation, which biases models toward average predictions and neglects rare cases—an imbalanced regression scenario (Branco et al., 2016; Ribeiro, 2011). Addressing this requires (i) identifying critical target ranges, (ii) evaluating performance in these regions, and (iii) biasing learning towards rare instances (Ribeiro & Moniz, 2020). Existing studies address these issues from various angles (Ribeiro & Moniz, 2020; Branco et al., 2017; Torgo et al., 2015; Branco et al., 2019; Camacho et al., 2022; Torgo et al., 2013; Camacho & Bacao, 2024; Belhaouari et al., 2024; Tian et al., 2023; Liu & Tian, 2024). This work focuses on strategies for (i), applies suitable metrics for (ii), and proposes a novel data-level approach for (iii).

### 2.2 DEFINITION OF RELEVANT AND RARE CASES

A central challenge in imbalanced regression is defining non-uniform preferences over continuous domains. While full domain knowledge would ideally guide this process, it is rarely available, particularly for infinite target domains. Two main approaches address this issue. The first, proposed by Ribeiro (2011); Ribeiro & Moniz (2020), approximates a relevance function $\phi() \in [0, 1]$ via interpolation of domain-based control points; when knowledge is lacking, it employs a non-parametric, data-driven procedure that assumes extreme values are most important and derives control points from adjusted boxplot statistics (Hubert & Vandervieren, 2008). The second, DenseWeight (Steininger et al., 2021), assigns weights inversely proportional to the estimated density of target values using Kernel Density Estimation (KDE), emphasizing underrepresented outcomes by prioritizing low-density regions; all weights are positive and normalized to ensure stable gradient descent. Unlike the relevance-based approach, DenseWeight uses probabilistic estimation, which may not match user preferences when density imbalance does not reflect their priorities.

### 2.3 ERROR METRICS

Evaluating predictive performance in imbalanced regression is a methodological challenge, as standard metrics such as MSE are ill-suited for these settings, focusing solely on the magnitude of errors and ignoring their distributional relevance or location in the target space (Ribeiro & Moniz, 2020). To overcome these limitations, relevance-aware metrics have been proposed. Relevance-Weighted Root Mean Squared Error (RW-RMSE) (Branco & Tulon, 2025) (Eq. 1) extends RMSE by incorporating a relevance function $\phi()$, thus capturing both the magnitude and contextual importance of errors:

$$\text{RW-RMSE} = \sqrt{\frac{\sum_{i=1}^n \phi(y_i) \cdot (y_i - \hat{y}_i)^2}{\sum_{i=1}^n \phi(y_i)}} \tag{1}$$

Similarly, Squared Error–Relevance Area (SERA) (Ribeiro & Moniz, 2020) (Eq. 2) prioritizes errors associated with highly relevant instances across the entire target domain without relying on arbitrary thresholds, leveraging instead the continuous relevance function $\phi()$:

$$SERA = \int_0^1 \sum_{i \in \mathcal{D}_t} (\hat{y}_i - y_i)^2 \, dt, \quad \mathcal{D}_t = \{\langle x_i, y_i \rangle \in \mathcal{D} \,|\, \phi(y_i) \geq t\} \tag{2}$$

### 2.4 DATA-LEVEL STRATEGIES

Data-level strategies address imbalanced regression by modifying data distributions to better represent relevant instances, using sampling and data augmentation techniques (Silva, 2022).

Examples include Random Undersampling (RU), Random Oversampling (RO), and WEighted Relevance Combination Strategy (WERCS) (Branco et al., 2019), which employ relevance func-

tions for guided sampling, with the latter notably avoiding predefined thresholds. Gaussian Noise (GN) (Branco et al., 2019) augments data by perturbing relevant cases while undersampling common ones. Additionally, SMOTER (Torgo et al., 2013) adapts SMOTE for regression by interpolating synthetic samples based on relevance, while SMOGN (Branco et al., 2017) combines undersampling with SMOTER and GN to balance synthetic data fidelity and variability. Furthermore, G-SMOTER (Camacho et al., 2022) extends G-SMOTE using geometric transformations to diversify generated samples.

However, many of these methods stem from classification settings, often relying on arbitrary thresholds to partition continuous target spaces, which is ill-suited for regression tasks. WSMOTER (Camacho & Bacao, 2024) mitigates this by integrating DenseWeight with SMOTE, using probabilistic weighting to focus on sparse regions. KNNOR-REG (Belhaouari et al., 2024) enhances SMOTE with k-NN filtering to identify representative minority points, addressing intra-class imbalance and noise.

Generative models like GANs and VAEs have also been explored for synthetic data generation. DIRVAE (Tian et al., 2023) uses a dual-model GAN framework to improve generative performance on sparse regression data. IRGAN (Liu & Tian, 2024) integrates generation, correction, discrimination, and regression modules for synthetic sample creation. DAVID (Stocksieker et al., 2024) combines regression training with $\beta$-VAE and VAE architectures. Despite their flexibility, these generative models are computationally intensive and offer limited interpretability.

## 3 OUR PROPOSAL

### 3.1 CART-BASED SYNTHETIC DATA GENERATION

The Classification and Regression Trees (CART) algorithm (Breiman et al., 1984) is a non-parametric method that recursively partitions the predictor space using binary splits, aiming to create regions increasingly homogeneous with respect to the target variable. Split selection is based on minimizing gini impurity (classification) or the sum of squared errors (regression).

Reiter (2005) proposed CART for synthetic data generation to safeguard sensitive microdata, highlighting its advantages over parametric models in handling unknown distributions, complex interactions, and missing values without explicit imputation. CART's recursive partitioning also aids variable selection in high-dimensional datasets.

Building on these principles, Panagiotou et al. (2024) conducted a comparative study on synthetic tabular data generation methods focused on class imbalance and fairness. Their approach generated synthetic data sequentially, column-wise, by sampling within CART tree leaves, as depicted in Figure 1. This method effectively mitigated class imbalance, outperforming traditional techniques like SMOTE and GANs.

Similarly, Akiya et al. (2024) applied CART to generate synthetic patient data (SPD) for survival analysis in oncology clinical trials. CART outperformed methods like Random Forests (RF), Bayesian Networks (BN), and Conditional Tabular GANs (CTGAN), especially in low-data scenarios. They suggest future improvements through enhanced feature engineering and hybrid approaches.

Overall, CART-based data generation effectively captures complex dependencies in the original data avoiding the distortions often introduced by SMOTE and GANs.

### 3.2 ADAPTATION FOR IMBALANCED REGRESSION

We propose CARTGen-IR, a method that employs CART to generate synthetic tabular data to tackle imbalanced regression problems. We aim to leverage the good performance that CART sample generation has shown in imbalanced classification problems and apply it to imbalanced regression. Furthermore, we aim to develop a strategy that removes the need for arbitrary user-defined thresholds, especially when identifying relevant or rare cases. This strategy enables us to avoid crisp divisions over the continuous domain of the target variable into bins or partitions, thereby preventing domain discretization, which is necessary in other previously proposed algorithms, such as SMOTER (Torgo et al., 2013) or SMOGN (Branco et al., 2017).

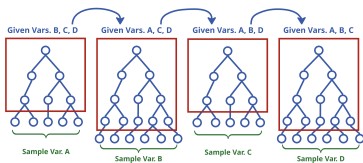

Figure 1: CART consecutive column-wise data generation

The main idea behind CARTGen-IR is as follows: given a dataset, a weighting mechanism is applied to the target domain values to rank them by rarity and relevance. Each instance is assigned a weight, with rarer and more relevant cases receiving a higher weight and vice versa. The original dataset is then resampled according to these weights. This ensures that relevant cases have a higher probability of being selected, and the final resampled dataset will primarily consist of rare cases related to the target variable domain. After this step, the algorithm is prepared to implement CART-based oversampling. To achieve this, we use the consecutive column-wise generation process, as described in Panagiotou et al. (2024). The algorithm first fits a CART tree for each column in the dataset, using the other columns as predictors, including the target variable. After fitting, it generates synthetic samples based on the constructed model.

The relevance-based weighting mechanism for rare target values is not new and is used in methods like WERCS (Branco et al., 2019) and G-SMOTER (Camacho et al., 2022) to prioritize relevant instances. However, CARTGen-IR stands out by resampling the original dataset and excluding cases with frequent target values from data augmentation. This approach minimizes the inclusion of frequent cases in the final training dataset, resulting in more focused augmentation.

The main steps of CARTGen-IR (cf. Algorithm 1) are:

1. **Assign a rarity score to each instance in the original dataset.** Our method employs three weighting techniques defined by the $density$ hyperparameter: a Kernel Density Estimation (KDE) method customized for this algorithm, the DenseWeight method (Steininger et al., 2021), and the relevance function (Ribeiro & Moniz, 2020). These techniques account for fluctuations in rarity across samples without assuming a specific distribution for rare cases or extremes. The density estimation helps compute a rarity score, which is determined by taking the inverse of the density (plus a small constant to avoid division by zero). However, any other weighting technique can be used. Final weight values are adjusted by a rarity exponent $\alpha$ and normalized, which involves dividing each score by the total sum of all weights.

2. **Resample the original dataset using rarity scores as weights.** The resampling process uses replacement to produce a dataset dominated by rare target values, with the number of selections linked to the desired synthetic samples. We introduced a hyperparameter $\eta$ (range [0,1]) to set the proportion of cases generated. For example, $\eta = 0.5$ creates 50% of the original dataset size, while higher values increase synthetic cases significantly. Following common practices in synthetic data generation for imbalanced learning (Chawla et al., 2002; He et al., 2008), we target five synthetic samples per selected instance to balance diversity and computational efficiency. The selection is based on rarity, ensuring minimal inclusion of common instances, without a preset threshold.

3. **Optionally input noise into duplicated instances.** The resampling process with replacement can create duplicated instances, potentially causing overfitting. To mitigate this, a small amount of noise can be added to the duplicates using the hyperparameter $\delta$.

4. **Fit CART to resampled dataset and generate synthetic samples.** To generate synthetic data, a CART decision tree is fitted for each column in the dataset, using the other columns as predictors in a sequential, column-wise generation process.

## 4 EXPERIMENTAL STUDY

Our primary objective is to evaluate CARTGen-IR performance and capabilities within the broader context of a comparative study against other state-of-the-art data-level strategies for imbalanced

**Algorithm 1:** CARTGen-IR Algorithm

---

**function** CARTGen-IR ($\mathbf{X}$: set of predictors, $Y$: target variable, $\alpha$: rarity exponent, $density \in \{"kde", "denseweight", "relevance"\}$, $\eta$: balance proportion, $\delta$: noise):

   |    $ws \leftarrow$ ComputeRarityWeights($Y, \alpha, density$)

   |    $N \leftarrow \lfloor \eta \cdot |Y| \rfloor$ ($\mathbf{X}_{\text{new}}, Y_{\text{new}}$) $\leftarrow$ sample($(\mathbf{X}, Y), N/5, ws, repl =$ True)

   |    **if** $\delta > 0$ **then**

   |      |   $\mathbf{X}_{\text{new}} \leftarrow$ AddGaussianNoise($\mathbf{X}_{\text{new}}, \delta$)

   |    **end**

   |    $model \leftarrow$ CART($\mathbf{X}_{\text{new}}, Y_{\text{new}}$)

   |    $(X_{\text{synth}}, Y_{\text{synth}}) \leftarrow$ CARTGen($model, N$)

   |    **return** $(\mathbf{X}, Y) \cup (\mathbf{X}_{synth}, Y_{synth})$

**function** ComputeRarityWeights($Y, \alpha, density$):

   |    **if** $density = "kde"$ **then**

   |      |   $ws \leftarrow (1/(\exp(\text{KDE}(Y, Gaussian) + 1e^{-5}))$

   |    **else if** $density = "denseweight"$ **then**

   |      |   $ws \leftarrow$ DenseWeight($Y$)

   |    **else if** $density = "relevance"$ **then**

   |      |   $ws \leftarrow \phi(Y)$

   |    $ws \leftarrow ws^{\alpha}$

   |    **return** $ws / \sum ws$

**function** AddGaussianNoise($\mathbf{X}, \delta$):

   |    $idx \leftarrow$ FindDuplicates($\mathbf{X}$)

   |    $attrs \leftarrow$ GetNumericAttrs($\mathbf{X}$)

   |    $\varepsilon \sim \mathcal{N}(0, \delta^2)$ of shape $|idx| \times |attrs|$ $\mathbf{X}[idx, attrs] \leftarrow \mathbf{X}[idx, attrs] + \varepsilon$

   |    **return** $\mathbf{X}$

---

regression tasks. With the experimental study, we aim to answer the following research questions: (RQ1) Is CARTGen-IR effective for imbalanced regression scenarios? (RQ2) How does it compare to the main state-of-the-art data-level methods proposed for imbalanced regression tasks for tabular data?; and, finally (RQ3) What tradeoff do these methods offer regarding predictive performance and execution time?

This section provides detailed information about the study, including experimental results and their discussion.

## 4.1 EXPERIMENTAL SETUP

We have created a publicly accessible repository with 62 datasets for imbalanced regression research from various application domains. This collection includes datasets referenced in studies such as Branco et al. (2017); Torgo et al. (2015); Branco et al. (2019); Camacho & Bacao (2024) and can be found in the supplementary material. Each dataset is briefly described and available for download in ARFF and CSV formats. The repository aims to serve as a standardized benchmark for investigating imbalanced regression issues across diverse real-world contexts.

In this study, we used 15 regression datasets from the repository. The key attributes of these datasets are summarized in Table 1, which also reports the absolute and relative frequencies of rare instances, defined according to a relevance threshold of 0.8. To accomplish this, we derived a relevance function for each dataset using the automated approach outlined in Ribeiro & Moniz (2020). The chosen datasets present a broad range of characteristics such as numeric and nominal features, instances, types of extremes, and rare occurrences, carefully chosen, being representative of the entire pool of benchmark datasets in terms of these characteristics.

We evaluated a comprehensive set of preprocessing strategies to address data imbalance in regression tasks. The approaches considered include RU, RO, WERCS, GN, SMOTER, SMOGN, WSMOTER, G-SMOTER, DAVID, KNNOR-REG and CARTGen-IR. A complete overview of the 72 resampling configurations is presented in Table 2. It is important to note that a direct comparison with the DIRVAE approach proposed by Tian et al. (2023) as well as the IRGAN approach proposed by Liu & Tian (2024) was not possible, as both source codes are not publicly available.

All these methods are sampling algorithms or ones that enhance the training set by generating additional synthetic instances. To evaluate their effectiveness, they must be coupled with a predictive learning model. In this study, we employed three such models: Random Forest (RF), Support Vector Regressor (SVR), and XGBoost (XGB). The experimental setup, shown in Table 3, comprised 14 different hyperparameter combinations for tuning these learning models. Each model was evaluated

Table 1: Benchmark regression datasets

| Dataset | N | Nom. Feat. | Num. Feat. | Type of Extreme | # Rare | % Rare |
|---|---|---|---|---|---|---|
| strikes | 625 | 0 | 6 | High | 15 | 2.40 |
| forestFires | 517 | 0 | 12 | High | 15 | 2.90 |
| ele-1 | 495 | 0 | 2 | High | 21 | 4.24 |
| cpuSm | 8192 | 0 | 12 | Low | 371 | 4.53 |
| airfoil | 1503 | 0 | 5 | High | 80 | 5.32 |
| fuelConsumption | 1764 | 12 | 25 | Both | 167 | 9.47 |
| heat | 7400 | 3 | 8 | Both | 833 | 11.26 |
| sensory | 576 | 0 | 11 | Both | 69 | 11.98 |
| mortgage | 1049 | 0 | 15 | Low | 133 | 12.68 |
| maxTorque | 1802 | 13 | 19 | Both | 235 | 13.04 |
| treasury | 1049 | 0 | 15 | Low | 137 | 13.06 |
| availablePower | 1802 | 7 | 8 | Both | 305 | 16.93 |
| housingBoston | 506 | 0 | 13 | Both | 105 | 20.75 |
| abalone | 4177 | 1 | 7 | Both | 1033 | 24.73 |
| servo | 167 | 4 | 0 | Both | 59 | 35.33 |

Table 2: Preprocessing strategies and hyperparameters

| Strategy | Hyperparameters |
|---|---|
| None | – |
| RU | %u = {balance, extreme} |
| RO | %o = {balance, extreme} |
| WERCS | %u = {0.5, 0.75}, %o = {0.5, 0.75} |
| GN | %u/%o = {balance, extreme}, $\delta$ = {0.05, 0.1, 0.5} |
| SMOTER | %u/%o = {balance, extreme} |
| SMOGN | %u/%o = {balance, extreme}, $\delta$ = {0.05, 0.1, 0.5} |
| WSMOTER | ratio = {1.5, 1.75}, $\beta$ = {1, 2} |
| G-SMOTER | strategy = {minority, majority, combined}, $\xi$ = {0.7}, $\eta$ = {0.75}, $\theta$ = {-0.5, 0.5}, $k$ = {5} |
| DAVID | $\alpha$ = {1, 2} |
| KNNOR-REG | – |
| CARTGen-IR | $\alpha$ = {1, 1.5, 2.0}, $\eta$ = {0.5, 0.75}, density = {kde, denseweight, relevance}, $\delta$ = {0.001, 0} |

across 15 regression datasets under all 72 preprocessing conditions, resulting in a total of 15.120 experiments ($14 \times 15 \times 72$).

In this study, we used both SERA and RW-RMSE as evaluation metrics to better assess model performance under imbalanced conditions. We also included RMSE as a standard regression error metric. Additionally, we adapted both RMSE and SERA to use DenseWeight as their weighting mechanism - DW-RMSE and DW-SERA - ensuring a fair comparison for methods that rely on this algorithm for weighting. All evaluation metrics were computed using a stratified repeated 2×5-fold cross-validation procedure to ensure robust and reliable performance estimates.

## 4.2 ERROR PERFORMANCE RESULTS

Table 4 displays the number of datasets where each combination of resampling strategy and learner achieves the best error estimates for each metric. For RMSE, CARTGen-IR stands out as the best-performing technique, surpassing WSMOTER, the second best rated strategy. This superiority is consistently confirmed across RW-RMSE, SERA, DW-RMSE and DW-SERA metrics, where CARTGen-IR leads with 21, 22, 21 and 18 top rankings, respectively, with no other method nearing its dominance. Performance across preprocessing strategies is uniformly distributed among learners, with no model significantly outperforming the others overall. However, for CARTGen-IR, ensemble models like RF and XGBoost yield better results, while SVR consistently underperforms. Interestingly, this trend is not observed with other strategies, which frequently achieve their best results with SVR. A detailed table of the best score values is provided in Appendix A.1.2.

Figure 2 summarizes the wins and losses, including statistically significant outcomes, of each data augmentation technique for all metrics. This analysis complements Table 4, which highlights results per learner, by evaluating strategies independent of learner, using the original imbalanced data as the baseline. Statistical significance was assessed through a Wilcoxon Signed Rank Test with a 95% significance threshold. Given the large number of parameter configurations (36) for CARTGen-IR, only its six best-performing versions are included in the figure, alongside all versions of the competing strategies.

Table 3: Learner hyperparameter configurations

| Model | Hyperparameters |
|---|---|
| Random Forest (RF) | n_estimators = {100, 200}, max_features = {sqrt, log2} |
| Support Vector Regressor (SVR) | kernel = {rbf}, C = {1, 10, 100}, epsilon = {0.1, 0.5} |
| XGBoost (XGB) | n_estimators = {100, 200}, max_depth = {3, 6} |

Table 4: Counts of best scores by preprocessing strategy / learner for each metric

| Metric | Learner | None | RU | RO | WERCS | GN | SMOTER | SMOGN | WSMOTER | G-SMOTER | DAVID | KNNOR-REG | CARTGen-IR |
|---|---|---|---|---|---|---|---|---|---|---|---|---|---|
| RMSE | RF | 2 | 0 | 1 | 0 | 0 | 0 | 0 | 3 | 0 | 0 | 2 | **7** |
| | SVR | 1 | 0 | 4 | 0 | 0 | 0 | 0 | 3 | 1 | 0 | 1 | **5** |
| | XGBoost | 1 | 0 | 1 | 0 | 0 | 0 | 0 | 3 | 3 | 0 | 0 | **7** |
| RW-RMSE | RF | 0 | 0 | 0 | 4 | 0 | 1 | 0 | 2 | 0 | 0 | 0 | **8** |
| | SVR | 0 | 0 | 4 | 3 | 0 | 0 | 1 | 1 | 0 | 0 | 0 | **6** |
| | XGBoost | 1 | 0 | 1 | 1 | 0 | 0 | 0 | 3 | 2 | 0 | 0 | **7** |
| SERA | RF | 0 | 0 | 0 | 3 | 0 | 1 | 1 | 2 | 0 | 0 | 0 | **8** |
| | SVR | 0 | 0 | 4 | 2 | 0 | 0 | 2 | 2 | 0 | 0 | 0 | **5** |
| | XGBoost | 1 | 0 | 0 | 1 | 0 | 0 | 0 | 3 | 1 | 0 | 0 | **9** |
| DW-RMSE | RF | 0 | 0 | 1 | 2 | 0 | 0 | 0 | 2 | 0 | 0 | 0 | **10** |
| | SVR | 0 | 0 | 3 | 2 | 0 | 0 | 0 | 4 | 1 | 0 | 0 | **5** |
| | XGBoost | 1 | 0 | 1 | 0 | 1 | 0 | 0 | 5 | 1 | 0 | 0 | **6** |
| DW-SERA | RF | 0 | 0 | 1 | 0 | 0 | 0 | 1 | 4 | 0 | 0 | 1 | **8** |
| | SVR | 0 | 0 | 3 | 0 | 1 | 0 | 0 | **5** | 2 | 0 | 0 | 4 |
| | XGBoost | 1 | 0 | 1 | 0 | 0 | 1 | 0 | 4 | 2 | 0 | 0 | **6** |

From this analysis, WSMOTER emerges as the most consistent performer across metrics, followed by KNNOR-REG and G-SMOTER. CARTGen-IR ranks fourth in overall consistency, with a level of robustness not observed in other methods. For instance, WERCS performs well in its oversampling-dominant versions but declines when undersampling becomes more prominent. Notably, although CARTGen-IR is not the most frequent winner, it exhibits a superior significant win-to-loss ratio compared to similarly performing methods, indicating that its victories are generally more meaningful. Additionally, the most effective CARTGen-IR variants share key characteristics: the introduction of Gaussian noise into numerical variables and higher $\alpha$ values (1.5 and 2), which appear beneficial. Regarding other hyperparameters, DenseWeight and Relevance outperform alternative rarity scoring methods, while the percentage of synthetic samples generated ($\eta$) shows minimal impact, as both tested values yielded similar outcomes.

Additionally, we applied multiple statistical tests to compare the resampling strategies across 15 datasets. The Friedman F-test revealed significant performance differences among the strategies for all metrics, leading to the rejection of the null hypothesis at a 95% confidence level. Consequently, we performed a post-hoc Nemenyi test to identify which methods differed significantly, using the same significance threshold. The results were visualized through Critical Difference (CD) diagrams (Demsar, 2006), which depict average rankings and highlight non-significant differences between strategies. These diagrams are available in Appendix A.1.4. Although overall differences among methods were not statistically significant, CARTGen-IR consistently ranked as the best-performing strategy across all metrics, particularly excelling in RW-RMSE and SERA, followed by WSMOTER and G-SMOTER. Even when CARTGen-IR was not the top-ranked method (e.g., with SVR under RMSE or DW-SERA), its performance remained statistically comparable to the best. SVR was the weakest learner for CARTGen-IR, while DAVID and RU consistently ranked as the least effective strategies.

Given that CARTGen-IR and WSMOTER exhibited superior results in previous analyses, we conducted a Bayesian Signed-Rank test to compare them. Unlike Critical Difference (CD) diagrams (Demsar, 2006), which depict average rank differences but lack probabilistic quantification, the Bayesian signed-rank test estimates the posterior probabilities that one method outperforms another, is inferior, or performs equivalently, incorporating a Region of Practical Equivalence (ROPE) (Benavoli et al., 2017; 2014). This allows for a more nuanced interpretation beyond binary hypothesis testing. For this study, we defined the ROPE interval as $[-1\%, 1\%]$. The results are fully presented in Appendix A.1.5, and a representative portion is shown in Figure 3 with six Bayesian ternary plots. CARTGen-IR demonstrated a clear advantage over WSMOTER with RF, achieving a 99% posterior probability of superiority across all metrics (Figures 3a and 3b). With

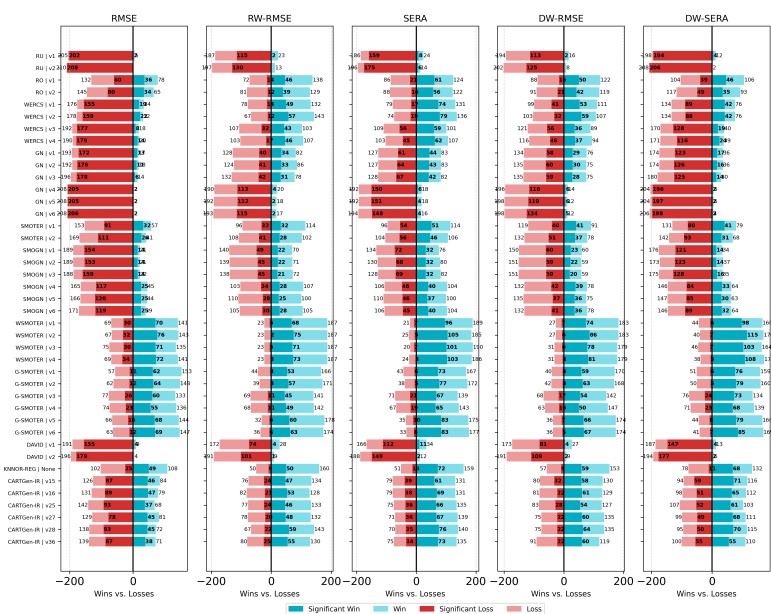

Figure 2: Comparison of (significant) wins and losses

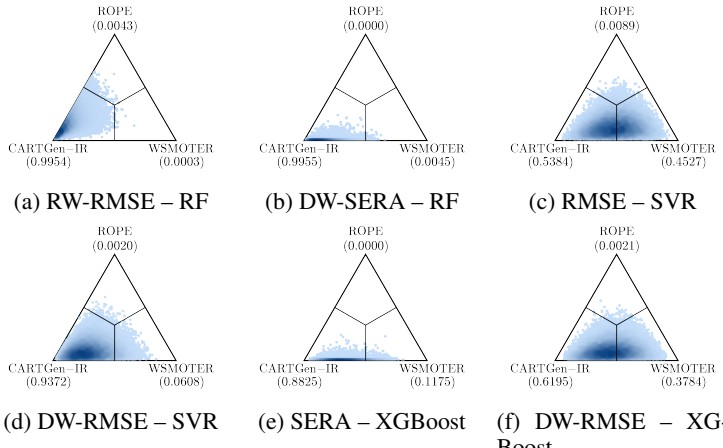

| (a) RW-RMSE – RF | (b) DW-SERA – RF | (c) RMSE – SVR |
| (d) DW-RMSE – SVR | (e) SERA – XGBoost | (f) DW-RMSE – XG-Boost |

Figure 3: Bayesian Posterior Ternary Plots

SVR, results were more balanced for RMSE and RW-RMSE, with CARTGen-IR favored in 53% of cases, while WSMOTER had a 46% advantage (Figure 3c), though CARTGen-IR dominated in SERA and DW-RMSE with over 90% probability (Figure 3d). For XGBoost, CARTGen-IR consistently outperformed WSMOTER, with posterior probabilities surpassing 70% for most metrics, and peaking at 90% for SERA (Figure 3e). The lowest probability for CARTGen-IR's superiority with XGBoost was 61% for DW-RMSE (Figure 3f). Overall, the Bayesian Signed-Rank test confirms CARTGen-IR's superior performance across metrics and learners, especially with ensemble models, while remaining competitive against WSMOTER when paired with SVR.

## 4.3 RUNTIME RESULTS

To conclude our experimental study, we performed a runtime comparison across all data augmentation techniques. For fairness, we measured execution times exclusively for the data augmentation procedures under identical parallelization conditions. Table 5 presents the runtime values (in seconds) for each strategy. CARTGen-IR stands out as one of the fastest techniques among those generating synthetic data. Sampling-based methods such as random undersampling, random oversam-

Table 5: Aggregated average runtime per preprocessing strategy

|  | RU | RO | WERCS | GN | SMOTER | SMOGN | WSMOTER | G-SMOTER | DAVID | KNNOR-REG | CARTGen-IR |
|---|---|---|---|---|---|---|---|---|---|---|---|
| Average | 0.027 | 0.058 | 0.002 | 1.518 | 5.147 | 5.350 | 0.314 | 0.244 | 24.077 | 0.034 | 0.183 |
| Std. Dev. | 0.000 | 0.000 | 0.000 | 0.008 | 0.016 | 0.019 | 0.003 | 0.001 | 0.083 | 0.005 | 0.001 |

pling, and WERCS exhibit the lowest runtimes, as they do not synthesize new data. Among augmentation methods, only KNNOR-REG surpasses CARTGen-IR in speed, though CARTGen-IR demonstrates lower standard deviation, indicating greater consistency. WSMOTER and G-SMOTER have runtimes similar to CARTGen-IR, while other SMOTER-based methods are significantly slower. DAVID is the slowest, taking about 31 times longer than CARTGen-IR.

## 4.4 DISCUSSION

In response to RQ1 and RQ2, CARTGen-IR has proven to be effective for imbalanced regression tasks. It consistently achieves strong performance across all datasets and ranks the highest overall. When compared to the leading state-of-the-art resampling strategies, CARTGen-IR either improves upon or matches these methods, particularly in balancing the focus between rare-valued and common-valued cases.For the RW-RMSE and SERA metrics, which are specifically designed to assess performance in imbalanced regression tasks, WSMOTER, KNORRR-REG, and CARTGen-IR emerge as clear winners, also demonstrating solid performance for RMSE. This indicates that CARTGen-IR generalizes well across the entire domain without compromising the overall predictive performance. Furthermore, it exhibits a significantly superior win-to-loss ratio compared to the other two methods. Concerning RQ3, when analyzing the characteristics of each preprocessing strategy, we can categorize the methods into sampling and augmentation techniques. Sampling techniques, due to their simplicity in implementation, result in the lowest execution times; however, they also yield inconsistent and lower-ranked scores. The other methods, which generate new synthetic samples, can then be compared. The best method in terms of runtime is KNNOR-REG, which, as previously mentioned, struggles with rare-valued target cases. CARTGen-IR ranks second in execution time and exhibits strong performance, effectively balancing the trade-off between the two.

## 5 CONCLUSIONS

This work focuses on imbalanced regression problems, where the objective is to predict rare values of the continuous target variable, a still-challenging problem in machine learning. It introduces CARTGen-IR, a non-parametric method specifically designed for imbalanced regression, which is based on a previously proposed CART-based synthetic generation method for tabular data (Reiter, 2005; Panagiotou et al., 2024). Unlike other state-of-the-art resampling strategies proposed for imbalanced regression, this approach does not rely on any specific threshold set for the continuous target variable. By leveraging a CART-based mechanism, CARTGen-IR can model complex relationships in the data while maintaining computational efficiency and interpretability. The empirical evaluation, conducted across a diverse set of benchmark datasets and state-of-the-art methods, confirms the competitiveness of the proposed approach in imbalanced regression scenarios.

As additional contributions, this work includes the extension and adaptation of relevance-based techniques, such as random under- and oversampling, SMOTER and SMOGN, to utilize adjusted boxplot statistics for accounting for skewness; the adaptation of both RMSE and SERA metrics to use DenseWeight as their weighting mechanism; and the public release of a curated repository comprising 62 benchmark datasets for imbalanced regression. The code used in this study is available in the supplementary material and is fully reproducible.

Even though CARTGen-IR demonstrates promise for tackling imbalanced regression problems, several directions for future work include: (i) expanding the comparative study to a broader dataset pool, particularly those containing target domain rare intervals that are not extremes; (ii) incorporating learners with a cost-sensitive approach to imbalanced regression-specific metrics, such as SERA; and (iii) identifying techniques to address the unsuitability of CARTGen-IR for parametric distributions, as partition boundaries may underperform in effectively capturing the true data relationships.

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

# A APPENDIX

## A.1 EXPERIMENTAL RESULTS

### A.1.1 DATA-LEVEL ALGORITHMS VARIANTS AND HYPERPARAMETERS

Table 6: Preprocessing strategy variants and respective hyperparameters

| Strategy | Variant | Hyperparameters Combination |
|---|---|---|
| RO/RU | v1 | (%u / %o = 'balance') |
| | v2 | (%u / %o = 'extreme') |
| WERCS | v1 | (%u = 0.5, %o = 0.5) |
| | v2 | (%u = 0.5, %o = 0.75) |
| | v3 | (%u = 0.75, %o = 0.5) |
| | v4 | (%u = 0.75, %o = 0.75) |
| GN | v1 | (%u/%o = 'balance', $\delta = 0.05$) |
| | v2 | (%u/%o = 'balance', $\delta = 0.1$) |
| | v3 | (%u/%o = 'balance', $\delta = 0.5$) |
| | v4 | (%u/%o = 'extreme', $\delta = 0.05$) |
| | v5 | (%u/%o = 'extreme', $\delta = 0.1$) |
| | v6 | (%u/%o = 'extreme', $\delta = 0.5$) |
| SMOTER | v1 | (%u/%o = 'balance', $k = 5$) |
| | v2 | (%u/%o = 'extreme', $k = 5$) |
| SMOGN | v1 | (%u/%o = 'balance', $k = 5$, $\delta = 0.05$) |
| | v2 | (%u/%o = 'balance', $k = 5$, $\delta = 0.1$) |
| | v3 | (%u/%o = 'balance', $k = 5$, $\delta = 0.5$) |
| | v4 | (%u/%o = 'extreme', $k = 5$, $\delta = 0.05$) |
| | v5 | (%u/%o = 'extreme', $k = 5$, $\delta = 0.1$) |
| | v6 | (%u/%o = 'extreme', $k = 5$, $\delta = 0.5$) |
| WSMOTER | v1 | ($\beta = 1$, ratio = 1.5, $k = 5$) |
| | v2 | ($\beta = 1$, ratio = 1.75, $k = 5$) |
| | v3 | ($\beta = 2$, ratio = 1.5, $k = 5$) |
| | v4 | ($\beta = 2$, ratio = 1.75, $k = 5$) |
| G-SMOTER | v1 | ($\xi = 0.7$, $k = 5$, $\eta = 0.75$, strategy = 'combined', $\theta$ = -0.5) |
| | v2 | ($\xi = 0.7$, $k = 5$, $\eta = 0.75$, strategy = 'combined', $\theta = 0.5$) |
| | v3 | ($\xi = 0.7$, $k = 5$, $\eta = 0.75$, strategy = 'majority', $\theta$ = -0.5) |
| | v4 | ($\xi = 0.7$, $k = 5$, $\eta = 0.75$, strategy = 'majority', $\theta = 0.5$) |
| | v5 | ($\xi = 0.7$, $k = 5$, $\eta = 0.75$, strategy = 'minority', $\theta$ = -0.5) |
| | v6 | ($\xi = 0.7$, $k = 5$, $\eta = 0.75$, strategy = 'minority', $\theta = 0.5$) |
| DAVID | v1 | ($\alpha = 1$) |
| | v2 | ($\alpha = 2$) |
| KNNOR-REG | v1 | - |

(Continued on the next page)

| Strategy | Variant | Hyperparameters Combination |
|---|---|---|
| CARTGen-IR | v1 | ($\alpha$ = 1, density = 'denseweight', $\delta$ = 0.00, $\eta$ = 'balance') |
| | v2 | ($\alpha$ = 1, density = 'denseweight', $\delta$ = 0.00, $\eta$ = 'extreme') |
| | v3 | ($\alpha$ = 1, density = 'denseweight', $\delta$ = 0.01, $\eta$ = 'balance') |
| | v4 | ($\alpha$ = 1, density = 'denseweight', $\delta$ = 0.01, $\eta$ = 'extreme') |
| | v5 | ($\alpha$ = 1, density = 'kde_baseline', $\delta$ = 0.00, $\eta$ = 'balance') |
| | v6 | ($\alpha$ = 1, density = 'kde_baseline', $\delta$ = 0.00, $\eta$ = 'extreme') |
| | v7 | ($\alpha$ = 1, density = 'kde_baseline', $\delta$ = 0.01, $\eta$ = 'balance') |
| | v8 | ($\alpha$ = 1, density = 'kde_baseline', $\delta$ = 0.01, $\eta$ = 'extreme') |
| | v9 | ($\alpha$ = 1, density = 'relevance', $\delta$ = 0.00, $\eta$ = 'balance') |
| | v10 | ($\alpha$ = 1, density = 'relevance', $\delta$ = 0.00, $\eta$ = 'extreme') |
| | v11 | ($\alpha$ = 1, density = 'relevance', $\delta$ = 0.01, $\eta$ = 'balance') |
| | v12 | ($\alpha$ = 1, density = 'relevance', $\delta$ = 0.01, $\eta$ = 'extreme') |
| | v13 | ($\alpha$ = 1.5, density = 'denseweight', $\delta$ = 0.00, $\eta$ = 'balance') |
| | v14 | ($\alpha$ = 1.5, density = 'denseweight', $\delta$ = 0.00, $\eta$ = 'extreme') |
| | v15 | ($\alpha$ = 1.5, density = 'denseweight', $\delta$ = 0.01, $\eta$ = 'balance') |
| | v16 | ($\alpha$ = 1.5, density = 'denseweight', $\delta$ = 0.01, $\eta$ = 'extreme') |
| | v17 | ($\alpha$ = 1.5, density = 'kde_baseline', $\delta$ = 0.00, $\eta$ = 'balance') |
| | v18 | ($\alpha$ = 1.5, density = 'kde_baseline', $\delta$ = 0.00, $\eta$ = 'extreme') |
| | v19 | ($\alpha$ = 1.5, density = 'kde_baseline', $\delta$ = 0.01, $\eta$ = 'balance') |
| | v20 | ($\alpha$ = 1.5, density = 'kde_baseline', $\delta$ = 0.01, $\eta$ = 'extreme') |
| | v21 | ($\alpha$ = 1.5, density = 'relevance', $\delta$ = 0.00, $\eta$ = 'balance') |
| | v22 | ($\alpha$ = 1.5, density = 'relevance', $\delta$ = 0.00, $\eta$ = 'extreme') |
| | v23 | ($\alpha$ = 1.5, density = 'relevance', $\delta$ = 0.01, $\eta$ = 'balance') |
| | v24 | ($\alpha$ = 1.5, density = 'relevance', $\delta$ = 0.01, $\eta$ = 'extreme') |
| | v25 | ($\alpha$ = 2.0, density = 'denseweight', $\delta$ = 0.00, $\eta$ = 'balance') |
| | v26 | ($\alpha$ = 2.0, density = 'denseweight', $\delta$ = 0.00, $\eta$ = 'extreme') |
| | v27 | ($\alpha$ = 2.0, density = 'denseweight', $\delta$ = 0.01, $\eta$ = 'balance') |
| | v28 | ($\alpha$ = 2.0, density = 'denseweight', $\delta$ = 0.01, $\eta$ = 'extreme') |
| | v29 | ($\alpha$ = 2.0, density = 'kde_baseline', $\delta$ = 0.00, $\eta$ = 'balance') |
| | v30 | ($\alpha$ = 2.0, density = 'kde_baseline', $\delta$ = 0.00, $\eta$ = 'extreme') |
| | v31 | ($\alpha$ = 2.0, density = 'kde_baseline', $\delta$ = 0.01, $\eta$ = 'balance') |
| | v32 | ($\alpha$ = 2.0, density = 'kde_baseline', $\delta$ = 0.01, $\eta$ = 'extreme') |
| | v33 | ($\alpha$ = 2.0, density = 'relevance', $\delta$ = 0.00, $\eta$ = 'balance') |
| | v34 | ($\alpha$ = 2.0, density = 'relevance', $\delta$ = 0.00, $\eta$ = 'extreme') |
| | v35 | ($\alpha$ = 2.0, density = 'relevance', $\delta$ = 0.01, $\eta$ = 'balance') |
| | v36 | ($\alpha$ = 2.0, density = 'relevance', $\delta$ = 0.01, $\eta$ = 'extreme') |

### A.1.2 Aggregated Results

Table 7: Best RMSE scores (mean ± std. dev.) across learners, strategies, and datasets. Best results per learner are bolded. The overall best per dataset is shaded in light blue.

| Learner | Strategy | abalone | airfoil | availablePower | cpuSm | ele-1 | forestFires | fuelConsumption | heat | housingBoston | maxTorque | mortgage | sensory | servo | strikes | treasury |
|---|---|---|---|---|---|---|---|---|---|---|---|---|---|---|---|---|
| RF | None | **2.157 (0.087)** | 1.774 (0.091) | 6.322 (1.884) | 2.826 (0.102) | 646.809 (78.54) | **55.657 (36.566)** | 0.486 (0.134) | 2.265 (0.177) | 3.236 (0.64) | 12.28 (2.774) | 0.136 (0.019) | 0.717 (0.038) | 0.607 (0.129) | **506.85 (143.141)** | 0.217 (0.043) |
| | RU | 2.897 (0.604) | 9.907 (0.307) | 26.469 (1.298) | 25.323 (2.222) | 1079.04 (110.776) | 225.951 (47.534) | 1.42 (0.135) | 13.573 (0.813) | 8.676 (2.178) | 56.236 (4.354) | 0.653 (0.058) | 0.973 (0.222) | 1.09 (0.31) | 2337.521 (234.62) | 0.657 (0.054) |
| | RO | 2.181 (0.089) | 1.87 (0.114) | **6.045 (1.713)** | 2.861 (0.13) | 651.264 (82.385) | 62.492 (33.583) | 0.484 (0.14) | 2.218 (0.205) | 3.21 (0.556) | 11.43 (2.457) | 0.138 (0.02) | 0.721 (0.034) | 0.558 (0.124) | 520.809 (146.245) | 0.219 (0.043) |
| | WERCS | 2.301 (0.075) | 2.612 (0.208) | 6.968 (1.969) | 2.915 (0.094) | 666.804 (69.497) | 71.909 (27.18) | 0.521 (0.118) | 2.844 (0.155) | 3.239 (0.48) | 13.362 (3.069) | 0.199 (0.017) | 0.745 (0.051) | 0.736 (0.16) | 631.342 (133.774) | 0.284 (0.106) |
| | GN | 2.324 (0.072) | 2.504 (0.167) | 6.608 (2.03) | 2.936 (0.078) | 743.947 (100.74) | 77.188 (26.563) | 0.559 (0.122) | 3.71 (0.185) | 3.494 (0.4) | 15.804 (3.339) | 0.199 (0.027) | 0.744 (0.051) | 0.6 (0.117) | 561.74 (128.888) | 0.252 (0.042) |
| | SMOTER | 2.561 (0.117) | 1.87 (0.128) | 7.061 (2.992) | 2.882 (0.14) | 700.95 (103.09) | 59.493 (35.126) | 0.484 (0.131) | 2.189 (0.205) | 3.247 (0.512) | 11.924 (1.971) | 0.153 (0.026) | 0.725 (0.032) | 0.546 (0.145) | 508.507 (140.601) | 0.235 (0.045) |
| | SMOGN | 2.611 (0.166) | 1.98 (0.109) | 6.122 (1.81) | 2.838 (0.089) | 710.91 (110.186) | 67.621 (32.052) | 0.482 (0.137) | 2.213 (0.204) | 3.215 (0.512) | 11.127 (1.657) | 0.166 (0.019) | 0.712 (0.029) | 0.553 (0.123) | 525.012 (143.887) | 0.247 (0.041) |
| | WSMOTER | 2.206 (0.083) | **1.742 (0.124)** | 6.256 (1.908) | 2.81 (0.087) | 656.571 (84.306) | 61.585 (33.542) | 0.472 (0.137) | 2.104 (0.168) | **3.037 (0.499)** | 11.517 (2.808) | **0.133 (0.019)** | 0.713 (0.026) | 0.526 (0.12) | 511.707 (139.193) | 0.202 (0.035) |
| | G-SMOTER | 2.158 (0.09) | 1.763 (0.096) | 6.238 (1.895) | 2.793 (0.086) | 641.887 (76.037) | 55.873 (37.16) | 0.481 (0.141) | 2.127 (0.186) | 3.146 (0.576) | 11.529 (3.092) | 0.135 (0.022) | 0.768 (0.046) | 0.526 (0.12) | 507.028 (139.136) | 0.215 (0.041) |
| | DAVID | 2.313 (0.095) | 4.113 (0.233) | 39.369 (8.618) | 3.608 (0.159) | 645.271 (72.465) | 77.981 (25.867) | 0.666 (0.355) | 27.501 (1.204) | 3.953 (0.818) | 24.59 (26.382) | 0.197 (0.022) | 0.721 (0.037) | 1.701 (0.26) | 639.218 (133) | 0.297 (0.072) |
| | KNNOR-REG | 2.175 (0.077) | 1.786 (0.11) | 6.05 (1.828) | 2.826 (0.105) | **641.016 (76.971)** | 56.116 (36.575) | 0.486 (0.14) | 2.17 (0.168) | 3.237 (0.627) | 12.533 (3.054) | 0.137 (0.019) | **0.709 (0.029)** | 0.607 (0.165) | 507.018 (141.403) | 0.215 (0.038) |
| | CARTGen-IR | 2.161 (0.087) | 1.778 (0.088) | 6.05 (1.828) | **2.777 (0.063)** | 649.240 (85.39) | 58.644 (35.935) | **0.471 (0.135)** | **1.949 (0.178)** | 3.045 (0.546) | **11.06 (3.324)** | 0.137 (0.019) | 0.709 (0.029) | **0.497 (0.12)** | 507.314 (142.911) | **0.199 (0.035)** |
| SVR | None | **2.118 (0.084)** | 2.789 (0.155) | 8.216 (1.551) | 3.481 (0.21) | 736.059 (125.845) | 50.486 (42.224) | 0.592 (0.153) | 0.754 (0.227) | 3.234 (0.678) | 14.09 (5.586) | 0.066 (0.008) | 0.769 (0.054) | 0.478 (0.072) | 518.313 (126.646) | 0.183 (0.032) |
| | RU | 2.403 (0.095) | 6.298 (0.391) | 18.158 (1.758) | 8.118 (0.317) | 1036.317 (93.201) | 50.028 (41.956) | 0.951 (0.108) | 6.726 (0.229) | 5.403 (0.757) | 30.91 (3.156) | 1.376 (0.194) | 0.947 (0.152) | 1.013 (0.253) | 527.223 (126.247) | 1.632 (0.209) |
| | RO | 2.399 (0.157) | 3.74 (0.336) | 7.727 (1.385) | 3.584 (0.244) | 764.579 (100.299) | 50.753 (39.733) | 0.603 (0.142) | **0.441 (0.057)** | 3.291 (0.558) | **10.886 (1.964)** | **0.066 (0.007)** | 0.813 (0.055) | **0.465 (0.076)** | 533.552 (138.892) | 0.184 (0.031) |
| | WERCS | 2.269 (0.067) | 3.423 (0.264) | 8.529 (1.175) | 3.398 (0.183) | 715.259 (65.68) | 56.143 (36.115) | 0.629 (0.128) | 0.502 (0.104) | 3.621 (0.616) | 13.478 (3.21) | 0.097 (0.016) | 0.893 (0.086) | 0.714 (0.136) | 562.514 (135.315) | 0.213 (0.04) |
| | GN | 2.389 (0.083) | 4.324 (0.397) | 9.633 (1.717) | 4.353 (0.522) | 867.891 (88.391) | 61.972 (33.534) | 0.668 (0.116) | 1.621 (0.219) | 3.714 (0.525) | 15.745 (2.889) | 0.082 (0.012) | 0.877 (0.074) | 0.589 (0.106) | 569.876 (145.308) | 0.201 (0.038) |
| | SMOTER | 2.531 (0.157) | 3.795 (0.337) | 11.929 (1.266) | 3.557 (0.265) | 733.421 (75.814) | 51.306 (39.317) | 0.607 (0.141) | 0.547 (0.106) | 3.399 (0.617) | 14.604 (4.146) | 0.066 (0.008) | 0.85 (0.089) | 0.483 (0.096) | 531.828 (151.555) | 0.184 (0.032) |
| | SMOGN | 2.871 (0.355) | 5.14 (0.519) | 13.559 (1.848) | 3.751 (0.275) | 816.114 (105.264) | 57.019 (35.126) | 0.607 (0.15) | 0.544 (0.106) | 3.4 (0.558) | 15.426 (5.098) | 0.076 (0.023) | 0.816 (0.055) | 0.5 (0.099) | 546.138 (156.987) | 0.188 (0.031) |
| | WSMOTER | 2.127 (0.07) | 2.786 (0.171) | **7.688 (1.373)** | 3.422 (0.216) | 666.744 (115.59) | 50.207 (42.222) | 0.598 (0.146) | 0.509 (0.158) | **3.128 (0.5)** | 12.716 (3.592) | 0.066 (0.009) | 0.807 (0.047) | 0.466 (0.073) | 503.601 (133.424) | **0.182 (0.031)** |
| | G-SMOTER | **2.113 (0.084)** | **2.785 (0.184)** | 8.103 (0.467) | 3.391 (0.185) | 698.015 (120.957) | 50.378 (42.188) | 0.591 (0.146) | 0.619 (0.188) | 3.179 (0.58) | 12.311 (3.068) | 0.066 (0.008) | 0.784 (0.052) | 0.488 (0.078) | 514.786 (127.707) | 0.181 (0.033) |
| | DAVID | 3.306 (0.33) | 5.14 (0.519) | 42.333 (7.702) | 4.841 (0.519) | 736.059 (125.845) | 149.361 (37.514) | 0.757 (0.367) | 22.273 (7.918) | 3.785 (0.56) | 28.259 (28.828) | 0.133 (0.014) | 0.926 (0.086) | 1.686 (0.263) | 523.404 (119.47) | 0.26 (0.064) |
| | KNNOR-REG | 2.143 (0.08) | 2.842 (0.142) | 8.328 (0.631) | 3.363 (0.205) | 698.7 (132.074) | 50.48 (42.297) | 0.591 (0.144) | 0.821 (0.21) | 3.222 (0.712) | 14.825 (5.008) | 0.07 (0.009) | **0.782 (0.052)** | 0.485 (0.074) | 507.162 (134.736) | 0.187 (0.035) |
| | CARTGen-IR | 2.141 (0.078) | 2.871 (0.156) | 8.458 (0.584) | **3.209 (0.174)** | **643.363 (70.038)** | **49.285 (41.758)** | **0.585 (0.13)** | 0.692 (0.122) | 3.237 (0.56) | 12.738 (2.314) | 0.068 (0.008) | 0.788 (0.052) | 0.483 (0.083) | **503.230 (133.99)** | 0.192 (0.032) |
| XGBoost | None | 2.199 (0.075) | 1.45 (0.108) | 6.078 (0.546) | 2.766 (0.171) | 727.643 (101.967) | 63.309 (33.893) | 0.462 (0.107) | **1.203 (0.057)** | 3.222 (0.642) | 5.742 (2.153) | 0.142 (0.021) | 0.722 (0.04) | 0.298 (0.073) | 571.419 (152.277) | 0.245 (0.027) |
| | RU | 3.135 (0.421) | 7.915 (0.217) | 22.452 (2.441) | 6.001 (1.868) | 996.917 (67.866) | 325.664 (123.877) | 1.571 (0.297) | 10.357 (0.979) | 9.773 (2.491) | 46.341 (6.211) | 0.495 (0.046) | 1.293 (0.131) | 1.481 (0.683) | 2799.445 (346.28) | 0.541 (0.097) |
| | RO | 2.339 (0.08) | 1.528 (0.177) | 5.921 (1.763) | 2.835 (0.188) | 715.421 (104.689) | 65.558 (37.552) | 0.468 (0.125) | 1.352 (0.05) | 3.227 (0.534) | 6.925 (1.975) | 0.14 (0.016) | 0.729 (0.034) | **0.298 (0.077)** | 587.919 (162.925) | 0.234 (0.036) |
| | WERCS | 2.452 (0.08) | 2.076 (0.204) | 7.67 (1.65) | 2.98 (0.152) | 733.381 (93.748) | 90.426 (29.594) | 0.557 (0.1) | 1.823 (0.072) | 3.289 (0.532) | 7.981 (2.107) | 0.212 (0.038) | 0.805 (0.05) | 0.652 (0.319) | 700.592 (117.604) | 0.311 (0.069) |
| | GN | 2.348 (0.086) | 2.22 (0.261) | 7.801 (1.656) | 2.99 (0.073) | 797.44 (113.26) | 88.621 (26.068) | 0.598 (0.119) | 3.043 (0.158) | 3.866 (0.552) | 12.211 (3.728) | 0.178 (0.028) | 0.789 (0.08) | 0.336 (0.084) | 701.053 (160.419) | 0.271 (0.049) |
| | SMOTER | 2.701 (0.132) | 1.594 (0.231) | 7.359 (2.044) | 2.916 (0.069) | 750.828 (108.07) | 68.845 (28.836) | 0.486 (0.123) | 1.604 (0.133) | 3.503 (0.704) | 9.041 (4.452) | 0.164 (0.02) | 0.762 (0.047) | 0.358 (0.097) | 583.903 (145.187) | 0.272 (0.046) |
| | SMOGN | 2.689 (0.12) | 1.721 (0.183) | 7.445 (1.785) | 2.99 (0.08) | 776.211 (109.248) | 83.963 (30.147) | 0.477 (0.13) | 1.627 (0.086) | 3.455 (0.688) | 7.859 (2.963) | 0.178 (0.023) | 0.715 (0.031) | 0.31 (0.07) | 604.471 (159.955) | 0.29 (0.036) |
| | WSMOTER | 2.252 (0.075) | 1.514 (0.13) | 5.807 (1.51) | 2.733 (0.054) | 712.924 (97.447) | 70.059 (37.382) | 0.454 (0.127) | 1.5 (0.054) | **2.976 (0.444)** | 6.338 (2.354) | 0.135 (0.023) | 0.705 (0.032) | 0.31 (0.07) | 551.085 (169.158) | **0.213 (0.043)** |
| | G-SMOTER | 2.188 (0.076) | **1.421 (0.127)** | 5.681 (1.206) | **2.721 (0.068)** | 710.517 (90.44) | 62.276 (35.402) | 0.455 (0.128) | 1.302 (0.077) | 2.979 (0.541) | **5.39 (1.261)** | **0.133 (0.018)** | 0.705 (0.032) | 0.305 (0.077) | 567.979 (137.351) | 0.23 (0.041) |
| | DAVID | 2.452 (0.108) | 3.709 (0.33) | 35.467 (8.861) | 3.668 (0.27) | 727.643 (101.967) | 99.689 (46.34) | 0.624 (0.297) | 15.342 (1.739) | 4.017 (0.882) | 17.015 (24.148) | 0.21 (0.029) | 0.817 (0.059) | 1.701 (0.247) | 729.166 (165.934) | 0.301 (0.063) |
| | KNNOR-REG | 2.232 (0.069) | 1.448 (0.104) | 5.658 (1.734) | 2.75 (0.054) | 713.414 (85.503) | 64.284 (33.102) | 0.474 (0.137) | 1.513 (0.098) | 3.116 (0.575) | 6.707 (2.628) | 0.143 (0.015) | 0.721 (0.034) | 0.298 (0.084) | 563.362 (140.201) | 0.238 (0.048) |
| | CARTGen-IR | **2.181 (0.075)** | 1.488 (0.117) | **5.303 (1.503)** | 2.771 (0.08) | **696.308 (84.072)** | **61.971 (37.88)** | **0.442 (0.112)** | 1.8 (0.077) | 3.11 (0.442) | 6.595 (2.144) | 0.141 (0.02) | **0.69 (0.04)** | 0.311 (0.056) | **547.091 (139.975)** | 0.230 (0.056) |

Table 8: Best RW-RMSE scores (mean ± std. dev.) across learners, strategies, and datasets. Best results per learner are bolded. The overall best per dataset is shaded in light blue.

| Learner | Strategy | abalone | airfoil | availablePower | cpuSm | ele-1 | forestFires | fuelConsumption | heat | housingBoston | maxTorque | mortgage | sensory | servo | strikes | treasury |
|---|---|---|---|---|---|---|---|---|---|---|---|---|---|---|---|---|
| RF | None | 2.778 (0.167) | 1.869 (0.293) | 10.717 (5.699) | 4.014 (0.553) | 829.122 (371.593) | 202.643 (145.721) | 0.743 (0.4) | 3.083 (0.717) | 3.65 (0.869) | 18.333 (9.397) | 0.066 (0.013) | 1.018 (0.097) | 0.533 (0.33) | 1730.972 (363.864) | 0.174 (0.131) |
| | RU | 3.04 (0.702) | 5.182 (3.675) | 22.117 (6.964) | 12.219 (5.504) | 1081.435 (148.981) | 238.235 (88.858) | 1.316 (0.247) | 10.759 (2.166) | 7.342 (2.269) | 47.947 (22.273) | 0.641 (0.127) | 1.263 (0.389) | 1.04 (0.782) | 1657.321 (176.062) | 0.633 (0.114) |
| | RO | 2.744 (0.17) | 1.871 (0.315) | 9.778 (5.138) | 4.165 (0.719) | 802.176 (354.137) | 199.648 (145.895) | 0.729 (0.399) | 3.029 (0.68) | 3.598 (0.825) | 16.631 (8.159) | 0.063 (0.01) | 1.005 (0.122) | 0.495 (0.274) | 1726.502 (374.669) | 0.173 (0.127) |
| | WERCS | **2.537 (0.107)** | 1.909 (0.915) | 9.926 (5.811) | 3.676 (0.389) | 795.194 (365.743) | **185.06 (140.214)** | **0.671 (0.377)** | 2.916 (0.623) | 3.483 (0.855) | 16.873 (9.273) | 0.063 (0.01) | 1.01 (0.137) | 0.673 (0.451) | **1621.218 (400.789)** | 0.265 (0.343) |
| | GN | 2.619 (0.138) | 2.248 (0.447) | 11.101 (5.56) | 4.028 (0.501) | 762.669 (340.907) | 188.98 (148.155) | 0.76 (0.384) | 4.259 (1.045) | 3.671 (0.934) | 21.696 (13.888) | 0.078 (0.015) | 0.981 (0.137) | 0.503 (0.232) | 1701.047 (378.476) | 0.204 (0.149) |
| | SMOTER | 2.738 (0.213) | 1.863 (0.291) | **9.481 (3.742)** | 4.268 (0.763) | 782.471 (336.469) | 197.622 (147.312) | 0.724 (0.398) | 2.955 (0.679) | 3.564 (0.739) | 18.797 (11.292) | 0.129 (0.051) | 1.073 (0.131) | 0.493 (0.296) | 1726.165 (364.242) | 0.253 (0.086) |
| | SMOGN | 2.779 (0.25) | 1.947 (0.292) | 10.245 (7.264) | 3.991 (0.443) | 760.554 (341.624) | 193.565 (146.612) | 0.671 (0.377) | 2.983 (0.609) | 3.514 (0.742) | 16.803 (8.648) | 0.139 (0.039) | 1.014 (0.11) | 0.473 (0.282) | 1713.167 (367.849) | 0.268 (0.093) |
| | WSMOTER | 2.638 (0.175) | **1.813 (0.36)** | 10.182 (5.314) | 3.778 (0.485) | 781.626 (337.697) | 197.737 (146.218) | 0.703 (0.401) | 2.769 (0.608) | 3.337 (0.625) | 17.235 (9.008) | **0.059 (0.012)** | 0.963 (0.126) | 0.489 (0.298) | 1718.533 (371.403) | 0.16 (0.117) |
| | G-SMOTER | 2.755 (0.164) | 1.845 (0.283) | 10.354 (5.642) | 3.895 (0.548) | 790.756 (343.329) | 200.671 (144.599) | 0.726 (0.398) | 2.858 (0.623) | 3.457 (0.854) | 17.387 (9.034) | 0.063 (0.013) | 0.979 (0.128) | 0.448 (0.287) | 1716.164 (358.437) | 0.169 (0.12) |
| | DAVID | 3.112 (0.231) | 5.087 (0.727) | 68.533 (35.766) | 5.792 (0.992) | 820.955 (363.69) | 195.877 (140.247) | 1.058 (0.635) | 39.636 (12.43) | 4.605 (1.045) | 45.672 (65.65) | 0.107 (0.022) | 1.089 (0.13) | 1.174 (1.225) | 1641.071 (387.724) | 0.249 (0.139) |
| | KNNOR-REG | 2.711 (0.163) | 1.839 (0.281) | 10.651 (5.462) | 3.913 (0.513) | 810.077 (353.151) | 201.724 (145.162) | 0.738 (0.406) | 2.926 (0.584) | 3.577 (0.856) | 19.098 (9.721) | 0.066 (0.012) | 1.004 (0.101) | 0.522 (0.313) | 1717.077 (383.916) | 0.174 (0.127) |
| | CARTGen-IR | 2.690 (0.145) | 1.842 (0.333) | 9.831 (5.5) | **3.631 (0.436)** | **672.071 (262.872)** | 190.003 (144.594) | 0.688 (0.406) | **2.606 (0.592)** | **3.299 (0.863)** | **15.498 (8.073)** | 0.061 (0.013) | **0.948 (0.15)** | **0.418 (0.244)** | 1629.396 (343.249) | **0.151 (0.11)** |
| SVR | None | 2.786 (0.149) | 3.458 (0.704) | 12.749 (4.45) | 6.021 (1.33) | 1138.685 (599.032) | 212.25 (144.593) | 0.9 (0.402) | 1.402 (0.636) | 3.614 (1.033) | 20.616 (12.343) | 0.033 (0.004) | 1.029 (0.18) | 0.439 (0.14) | 1829.381 (318.857) | 0.154 (0.086) |
| | RU | 2.849 (0.257) | 3.939 (2.371) | 16.353 (4.232) | 8.617 (1.441) | 1449.322 (249.568) | 188.064 (122.603) | 1.108 (0.211) | 5.963 (2.066) | 5.396 (0.855) | 40.559 (21.023) | 0.282 (0.116) | 1.365 (0.194) | 1.028 (0.763) | 1536.925 (309.118) | 1.2 (1.41) |
| | RO | 2.638 (0.179) | 3.632 (0.599) | **11.446 (4.361)** | 5.735 (1.041) | **720.32 (277.912)** | 186.487 (142.406) | 0.894 (0.397) | **0.671 (0.117)** | 3.522 (0.983) | **12.529 (4.876)** | 0.032 (0.003) | 1.016 (0.196) | 0.387 (0.16) | 1439.71 (332.128) | 0.157 (0.084) |
| | WERCS | **2.516 (0.189)** | **2.196 (0.784)** | 11.945 (3.786) | 4.239 (0.587) | 791.668 (246.73) | **178.177 (141.263)** | 0.857 (0.355) | 0.689 (0.375) | 3.921 (1.009) | 15.767 (7.686) | **0.031 (0.003)** | 1.063 (0.232) | 0.636 (0.356) | 1468.433 (340.399) | 0.184 (0.148) |
| | GN | 2.644 (0.178) | 3.561 (1.063) | 13.241 (4.864) | 5.847 (0.955) | 760.188 (245.113) | 182.057 (144.429) | 0.915 (0.349) | 2.496 (0.691) | 3.941 (0.865) | 20.442 (10.857) | 0.05 (0.01) | 1.018 (0.113) | 0.515 (0.198) | 1431.701 (332.061) | 0.167 (0.094) |
| | SMOTER | 2.667 (0.228) | 3.58 (0.6) | 17.628 (5.636) | 5.594 (1.023) | 739.805 (293.592) | 189.302 (143.316) | 0.9 (0.376) | 0.919 (0.238) | 3.611 (1.085) | 19.501 (8.368) | 0.032 (0.004) | 1.129 (0.254) | 0.4 (0.216) | 1437.286 (330.412) | 0.157 (0.096) |
| | SMOGN | 2.811 (0.29) | 3.518 (0.627) | 19.047 (6.278) | 5.644 (0.82) | 722.55 (274.979) | 183.349 (144.711) | 0.89 (0.382) | 0.914 (0.241) | 3.577 (1.002) | 16.805 (8.586) | 0.032 (0.004) | 1.031 (0.123) | 0.412 (0.196) | **1429.1 (335.515)** | 0.16 (0.087) |
| | WSMOTER | 2.503 (0.133) | 3.171 (0.717) | 11.64 (4.237) | 5.558 (1.263) | 951.804 (490.344) | 203.598 (147.167) | 0.907 (0.391) | 0.886 (0.388) | 3.511 (0.882) | 17.361 (8.413) | 0.032 (0.002) | 0.982 (0.192) | 0.401 (0.226) | 782.977 (334.862) | **0.15 (0.078)** |
| | G-SMOTER | 2.681 (0.161) | 3.256 (0.646) | 12.537 (4.642) | 5.483 (1.138) | 1065.396 (548.715) | 211.461 (144.689) | 0.886 (0.394) | 1.097 (0.448) | 3.514 (1.028) | 16.493 (7.293) | 0.034 (0.004) | 0.993 (0.187) | 0.425 (0.263) | 1819.209 (322.482) | 0.153 (0.087) |
| | DAVID | 3.352 (0.316) | 4.925 (0.838) | 74.915 (40.359) | 7.558 (1.235) | 1138.685 (599.032) | 209.139 (110.381) | 1.2 (0.665) | 31.314 (9.895) | 4.345 (1.089) | 52.33 (73.147) | 0.077 (0.013) | 1.1 (0.16) | 1.163 (1.215) | 1448.585 (328.368) | 0.222 (0.118) |
| | KNNOR-REG | 2.603 (0.14) | 3.378 (0.658) | 12.86 (4.747) | 5.508 (1.228) | 1037.281 (534.642) | 211.601 (144.67) | 0.89 (0.399) | 1.528 (0.568) | 3.52 (1.034) | 21.275 (12.001) | 0.035 (0.004) | 1.025 (0.187) | 0.437 (0.306) | 1790.726 (333.022) | 0.159 (0.097) |
| | CARTGen-IR | **2.468 (0.132)** | 2.401 (0.128) | 12.693 (3.431) | **4.122 (0.681)** | **662.75 (259.706)** | **179.673 (123.084)** | **0.812 (0.345)** | 1.015 (0.314) | **3.347 (0.922)** | 15.545 (5.803) | 0.036 (0.01) | **0.971 (0.164)** | **0.377 (0.221)** | 1616.316 (324.765) | 0.158 (0.084) |
| XGBoost | None | 2.797 (0.162) | 1.555 (0.322) | 9.864 (4.017) | 3.875 (0.936) | 909.944 (455.037) | 208.38 (137.905) | 0.68 (0.38) | **1.403 (0.263)** | 3.69 (0.84) | 7.237 (4.967) | 0.075 (0.009) | 0.923 (0.126) | 0.259 (0.197) | 1760.546 (336.539) | 0.211 (0.128) |
| | RU | 3.17 (0.52) | 4.399 (2.459) | 19.629 (4.362) | 6.249 (0.848) | 993.745 (365.984) | 329.65 (139.384) | 1.485 (0.353) | 7.151 (0.734) | 7.859 (2.467) | 36.73 (9.396) | 0.444 (0.203) | 1.413 (0.145) | 1.142 (1.084) | 2045.225 (454.847) | 0.505 (0.125) |
| | RO | 2.732 (0.187) | 1.529 (0.328) | 10.006 (3.927) | 4.082 (1.092) | 901.859 (454.269) | 200.918 (139.944) | 0.665 (0.4) | 1.615 (0.296) | 3.731 (0.697) | 8.944 (6.66) | 0.081 (0.01) | 0.914 (0.147) | **0.259 (0.175)** | 1771.008 (355.222) | 0.21 (0.141) |
| | WERCS | 2.611 (0.118) | 1.795 (0.38) | 11.217 (5.912) | 3.831 (0.916) | 886.384 (438.004) | 200.592 (135.764) | 0.686 (0.352) | 1.68 (0.371) | 3.516 (0.869) | 9.316 (6.442) | 0.079 (0.022) | 0.953 (0.119) | 0.568 (0.626) | **1652.766 (415.57)** | 0.286 (0.265) |
| | GN | 2.657 (0.116) | 1.99 (0.257) | 10.994 (5.187) | 4.25 (0.736) | 815.815 (380.997) | 200.999 (136.731) | 0.756 (0.387) | 3.276 (0.534) | 3.797 (0.494) | 15.713 (7.801) | 0.084 (0.012) | 0.927 (0.114) | 0.29 (0.21) | 1738.66 (397.418) | 0.222 (0.164) |
| | SMOTER | 2.876 (0.272) | 1.618 (0.304) | 10.434 (4.663) | 4.339 (0.521) | 870.453 (433.712) | 199.127 (145.613) | 0.705 (0.393) | 1.81 (0.41) | 3.82 (0.884) | 14.163 (16.995) | 0.157 (0.031) | 0.992 (0.135) | 0.298 (0.234) | 1756.15 (364.143) | 0.295 (0.14) |
| | SMOGN | 2.85 (0.247) | 1.702 (0.323) | 11.716 (6.999) | 4.413 (0.576) | 826.594 (387.609) | 200.615 (143.492) | 0.687 (0.354) | 1.814 (0.319) | 3.685 (0.849) | 11.814 (14.91) | 0.155 (0.039) | 0.926 (0.07) | 0.295 (0.274) | 1752.211 (311.355) | 0.314 (0.13) |
| | WSMOTER | 2.653 (0.194) | 1.621 (0.348) | 9.221 (4.387) | **3.558 (0.421)** | 869.937 (409.268) | 202.679 (146.806) | 0.665 (0.381) | 1.673 (0.311) | 3.4 (0.571) | **5.899 (3.401)** | **0.07 (0.008)** | 0.893 (0.145) | 0.28 (0.227) | 1744.496 (313.715) | 0.186 (0.118) |
| | G-SMOTER | **2.188 (0.076)** | **1.492 (0.349)** | 8.833 (3.833) | 3.65 (0.458) | 880.081 (433.029) | 206.272 (143.831) | 0.665 (0.378) | 1.51 (0.23) | 3.4 (0.787) | 7.96 (5.141) | 0.075 (0.01) | 0.903 (0.141) | 0.266 (0.209) | 1718.038 (313.928) | 0.258 (0.138) |
| | DAVID | 3.264 (0.289) | 4.596 (0.922) | 64.914 (37.018) | 5.499 (1.139) | 909.944 (455.037) | 224.515 (144.464) | 0.954 (0.571) | 24.79 (5.089) | 4.56 (0.868) | 34.043 (61.589) | 0.115 (0.016) | 0.969 (0.175) | 1.191 (1.198) | 1792.512 (308.193) | 0.258 (0.138) |
| | KNNOR-REG | 2.743 (0.155) | 1.555 (0.308) | 9.425 (4.562) | 3.609 (0.454) | 882.475 (406.867) | 203.695 (145.579) | 0.675 (0.404) | 1.723 (0.407) | 3.541 (0.711) | 9.474 (6.868) | 0.075 (0.014) | 0.918 (0.169) | 0.275 (0.201) | 1750.294 (359.657) | 0.215 (0.159) |
| | CARTGen-IR | 2.716 (0.161) | 1.554 (0.382) | **8.499 (4.582)** | 3.563 (0.43) | 720.549 (304.087) | **184.155 (141.16)** | **0.637 (0.382)** | 2.088 (0.352) | **3.336 (0.61)** | 8.432 (6.654) | 0.074 (0.009) | **0.854 (0.147)** | 0.260 (0.196) | 1676.084 (364.082) | **0.178 (0.101)** |

Table 9: Best SERA scores (mean ± std. dev.) across learners, strategies, and datasets. Best results per learner are bolded. The overall best per dataset is shaded in light blue.

| Learner | Strategy | abalone | airfoil | availablePower | cpuSm | ele-1 | forestFires | fuelConsumption | heat | housingBoston | maxTorque | mortgage | sensory | servo | strikes | treasury |
|---|---|---|---|---|---|---|---|---|---|---|---|---|---|---|---|---|
| RF | None | 2344.629 (329.112) | 159.874 (104.774) | 11842.204 (10101.166) | 3450.568 (727.225) | 16367575.153 (12686092.018) | 381956.514 (594335.436) | 57.768 (602.279) | 3879.483 (1140.02) | 412.586 (292.205) | 36044.139 (36404.312) | 0.232 (0.111) | 12.235 (16.742) | 5.334 (6.336) | 25381108.028 (20326797.014) | 2.547 (3.431) |
| | RU | 2986.899 (1689.722) | 1539.924 (2280.611) | 44649.644 (16301.308) | 71514.552 (143308.467) | 26871543.795 (14303528.629) | 419649.224 (464583.668) | 161.974 (41.712) | 47088.893 (8743.784) | 1601.324 (870.252) | 200265.979 (96480.171) | 21.205 (13.041) | 22.961 (42.111) | 17.361 (19.906) | 25124734.817 (15364389.077) | 23.109 (9.65) |
| | RO | 2287.8 (328.378) | 159.585 (102.068) | 9893.052 (8241.677) | 3735.555 (1073.682) | 15134441.125 (11563659.547) | 378064.642 (590846.781) | 55.768 (57.248) | 3753.244 (1178.119) | 409.321 (320.28) | 29022.864 (25808.217) | 0.212 (0.084) | 12.188 (17.24) | 4.581 (5.499) | 25307620.666 (20410124.679) | 2.469 (3.225) |
| | WERCS | **1956.194 (221.654)** | 179.906 (184.53) | 10443.236 (9913.754) | 2993.994 (935.359) | 19968789.183 (6617308.456) | 361156.676 (575640.712) | **47.942 (54.044)** | 3519.447 (30809.908) | 372.906 (296.079) | 46320.301 (36996.721) | 0.328 (0.162) | 11.711 (16.642) | 7.301 (7.506) | 22787844.929 (18708879.658) | 8.546 (18.943) |
| | GN | 2083.289 (268.835) | 232.506 (151.967) | 12210.932 (9645.757) | 3455.946 (550.262) | 13694771.614 (9134749.211) | 378625.72 (595499.223) | 54.795 (55.265) | 3601.57 (1227.412) | 431.813 (294.159) | 31342.497 (21364.349) | 0.924 (0.821) | 13.283 (17.67) | 4.164 (4.31) | 25243992.48 (20261619.591) | 3.41 (4.259) |
| | SMOTER | 2281.524 (374.114) | 158.863 (102.624) | **9130.566 (6466.139)** | 3939.978 (1218.266) | 19523047.611 (7582370.172) | 368072.609 (575257.301) | 54.679 (56.628) | 3681.22 (1182.118) | 392.205 (264.063) | **26050.38 (17780.248)** | 1.04 (0.639) | 12.36 (17.289) | 4.76 (6.248) | 24872393.023 (19819229.069) | 4.034 (2.862) |
| | SMOGN | 2353.047 (400.998) | 171.942 (104.984) | 10932.5 (7141.813) | 3422.919 (691.123) | 1750136.036 (11472349.9) | 375758.535 (585849.453) | 51.693 (54.736) | 3148.818 (949.233) | 388.919 (263.076) | 32071.863 (30908.145) | **0.189 (0.085)** | 10.986 (15.051) | 4.308 (5.374) | 25039470.188 (20107614.304) | 4.506 (3.078) |
| | WSMOTER | 2115.173 (281.047) | **153.412 (111.868)** | 10610.548 (9394.498) | 3062.003 (635.043) | 14670397.943 (11017383.605) | 381117.043 (598012.436) | 55.406 (58.757) | 3362.736 (1069.953) | 338.573 (211.384) | 32648.701 (29620.273) | 0.212 (0.099) | 11.462 (15.925) | 4.663 (5.999) | 24967333.226 (20042081.677) | 2.053 (2.625) |
| | G-SMOTER | 2305.142 (322.886) | 156.092 (103.836) | 11071.001 (10286.391) | 3249.952 (702.864) | 15797215.668 (11905308.992) | 364136.037 (587393.326) | 142.45 (173.653) | 711394.315 (75241.932) | 375.569 (272.512) | 298767.092 (555450.496) | 0.595 (0.284) | 13.905 (18.934) | 4.057 (5.201) | 23572002.803 (19642431.759) | 2.314 (2.965) |
| | DAVID | 2949.324 (501.444) | 1153.73 (680.296) | 463592.284 (254649.999) | 7188.322 (1854.144) | 1561209.187 (12687715.203) | 384958.607 (592964.064) | 56.981 (60.512) | 3526.569 (1021.95) | 673.693 (500.009) | 38831.558 (36039.75) | 0.231 (0.102) | 11.999 (16.523) | 5.521 (7.446) | 25064060.335 (20304976.391) | 4.307 (3.906) |
| | KNNOR-REG | 2230.848 (300.32) | 154.855 (101.328) | 11577.116 (9745.126) | 3298.454 (765.286) | **9641359.653 (5305522.995)** | 356122.41 (590855.567) | 58.711 (56.387) | **2803.56 (971.341)** | 405.258 (317.901) | 27661.357 (28286.847) | 0.202 (0.094) | **10.74 (14.975)** | — | **22785445.93 (18799517.479)** | 2.409 (2.956) |
| | CARTGen-IR | 2199.465 (285.904) | 156.399 (106.685) | 10126.531 (9286.489) | **2819.941 (501.566)** | — | 412490.983 (628747.3) | 48.97 (51.719) | — | **315.151 (261.439)** | — | — | — | **3.399 (4.221)** | — | **1.675 (2.235)** |
| SVR | None | 2355.9 (310.263) | 554.399 (376.766) | 15351.21 (9992.859) | 7527.234 (1887.381) | 32064626.075 (26203374.519) | 412490.983 (628747.3) | 81.528 (74.531) | 833.438 (534.35) | 418.307 (307.444) | 51280.201 (67675.031) | 0.059 (0.023) | 12.695 (17.584) | 2.862 (2.168) | 28055720.18 (21851073.743) | 1.639 (1.641) |
| | RU | 2492.661 (558.615) | 824.673 (1038.763) | 24329.207 (9871.24) | 18256.144 (5921.535) | 47996792.684 (28190828.129) | 329495.359 (485462.252) | 118.019 (51.651) | 13906.587 (3191.954) | 859.162 (460.116) | 143234.883 (91885.511) | 6.853 (4.259) | 23.633 (36.225) | 16.677 (19.806) | 21757880.642 (17095214.904) | 159.81 (229.183) |
| | RO | 2117.475 (323.715) | 596.363 (359.891) | **1761.124 (6796.224)** | 6868.794 (1205.497) | 1122497.97 (64448306.638) | 343367.67 (541688.101) | 79.765 (67.691) | **195.967 (89.426)** | 407.207 (327.602) | **14604.24 (6648.162)** | **0.052 (0.018)** | 12.91 (18.327) | 2.47 (2.845) | 18521496.436 (14994794.633) | 1.67 (1.589) |
| | WERCS | 1928.343 (319.295) | **230.799 (175.604)** | 12880.305 (6349.914) | 3904.678 (983.664) | 1413500.213 (915208.193) | **327776.497 (524862.456)** | 73.156 (61.376) | 203.172 (156.828) | 483.092 (341.423) | 26571.895 (25840.841) | 0.056 (0.019) | 13.596 (18.879) | 6.078 (5.216) | 18789098.822 (15261494.873) | 2.615 (3.557) |
| | GN | 2127.906 (312.399) | 585.888 (404.364) | 15601.596 (9208.897) | 7208.147 (1251.531) | 12376319.269 (5692001.862) | 334001.257 (488164.602) | 81.682 (56.094) | 2573.382 (1009.264) | 418.359 (312.224) | 37612.765 (24720.492) | 0.129 (0.06) | 13.185 (19.2) | 4.154 (3.921) | 1836496.0064 (14872271.472) | 1.892 (2.099) |
| | SMOTER | 2169.691 (393.304) | 580.944 (353.118) | 27903.371 (12219.001) | 6548.497 (1215.25) | 11583303.921 (6065409.576) | 351271.414 (551697.757) | 80.762 (65.027) | 365.242 (18.385) | 406.262 (291.589) | 40727.981 (35869.241) | 0.054 (0.022) | 14.692 (19.54) | 3.04 (4.015) | 18416233.475 (14861776.589) | 1.693 (1.599) |
| | SMOGN | 2414.716 (496.354) | 563.024 (347.75) | 32985.734 (16279.546) | 6713.197 (949.984) | **10966667.486 (5475754.39)** | 394428.678 (612752.095) | 79.517 (68.581) | 362.744 (180.353) | 385.739 (252.113) | 37022.391 (48892.196) | 0.057 (0.02) | 13.153 (18.67) | 2.75 (3.192) | **18305541.987 (14893393.673)** | 1.708 (1.692) |
| | WSMOTER | **1903.951 (257.877)** | 477.638 (340.714) | 12344.66 (7705.377) | 6449.697 (1760.759) | 2398679.4798 (19925692.542) | 410544.395 (626935.762) | 81.99 (70.694) | 350.447 (266.03) | 401.777 (312.53) | 32825.982 (32491.021) | 0.055 (0.019) | 11.666 (16.296) | 2.803 (2.188) | 2671639.875 (21000882.466) | **1.502 (1.373)** |
| | G-SMOTER | 2183.212 (311.589) | 492.017 (336.495) | 14550.106 (9346.16) | 6259.215 (1411.275) | 27946286.58 (22866952.192) | 363256.139 (489475.382) | 78.855 (71.712) | 532.524 (388.526) | 552.653 (299.609) | 25572.641 (20945.364) | 0.06 (0.023) | 11.861 (16.508) | 3.01 (1.763) | 27773504.303 (21692906.103) | 1.624 (1.646) |
| | DAVID | 3424.804 (655.687) | 1128.355 (752.581) | 520367.479 (271239.123) | 12110.664 (3375.807) | 32064626.075 (26203374.519) | 362356.139 (489475.382) | 175.109 (199.657) | 39281.404 (117931.623) | 399.308 (284.112) | 382397.394 (689362.773) | 0.309 (0.133) | 14.867 (21.049) | 38.335 (51.951) | 18883708.766 (14739834.783) | 3.383 (3.074) |
| | KNNOR-REG | 2058.452 (270.144) | 526.847 (355.705) | 15375.059 (9466.357) | 6325.31 (1677.963) | 27232262.334 (23458432.973) | 391747.858 (608570.172) | 80.065 (72.016) | 980.978 (529.666) | 342.082 (312.657) | 52870.274 (65651.06) | 0.064 (0.024) | 12.631 (17.536) | 2.921 (2.167) | 26947129.081 (21155835.418) | 1.795 (1.82) |
| | CARTGen-IR | 1969.096 (256.007) | 270.408 (177.737) | 14952.781 (8199.041) | **3593.039 (657.41)** | 11686222.95 (8975257.553) | 306885.035 (548003.586) | **62.952 (42.19)** | 429.148 (197.123) | **310.266 (79.017)** | 24691.581 (19290.679) | 0.071 (0.059) | **11.617 (17.394)** | **2.316 (2.805)** | 22264488.25 (18081697.318) | 1.682 (1.61) |
| XGBoost | None | 2374.63 (315.825) | 114.114 (84.907) | 10333.806 (7106.522) | 3309.938 (1695.902) | 20309229.0044 (16549918.56) | 392611.838 (600577.722) | 45.416 (40.823) | 804.596 (168.172) | 384.962 (164.654) | 6716.264 (9788.335) | 0.299 (0.13) | 10.015 (13.427) | 1.408 (1.707) | 2611402.5238 (20530250.69) | 3.206 (3.501) |
| | RU | 3159.822 (1237.196) | 1016.727 (1160.408) | 36797.841 (16101.916) | 8836.559 (3119.928) | 20884157.891 (88882706.261) | 773824.908 (628287.664) | 216.884 (79.959) | 22056.092 (3828.349) | 18709.0 (137.224) | 121705.089 (46014.264) | 11.713 (12.075) | 25.151 (38.408) | 29.278 (43.458) | 38489061.935 (27418018.346) | 15.565 (7.265) |
| | RO | 2269.276 (339.902) | 109.777 (82.774) | 10837.97 (7786.519) | 3685.209 (1917.929) | 20174218.594 (71052237.315) | 374657.306 (564693.61) | 48.235 (50.288) | 1057.109 (145.916) | 380.169 (98.899) | 10298.889 (13196.577) | 0.346 (0.151) | 9.992 (13.848) | 1.412 (1.801) | 26388078.191 (20442964.158) | 3.309 (4.045) |
| | WERCS | **2068.621 (223.018)** | 148.521 (94.004) | 13155.86 (9354.292) | 3304.37 (1765.408) | 18417245.406 (1341295.046) | 363848.51 (524111.045) | 47.852 (45.87) | 1134.906 (158.51) | 335.859 (133.114) | 9920.806 (9541.495) | 0.361 (0.117) | 10.695 (14.151) | 6.148 (10.857) | **22924043.178 (19940055.66)** | 7.28 (11.116) |
| | GN | 2145.011 (262.828) | 180.22 (103.939) | 11396.207 (7530.917) | 3839.478 (851.278) | 13909400.051 (6218362.772) | 371582.52 (544714.258) | 57.433 (49.686) | 4400.29 (823.995) | 419.463 (181.487) | 24274.669 (18845.454) | 0.384 (0.167) | 10.426 (14.325) | 1.664 (1.902) | 25664595.998 (20226280.29) | 3.925 (4.984) |
| | SMOTER | 2525.622 (496.765) | 123.666 (91.756) | 12592.249 (8220.734) | 4015.236 (688.486) | 17699029.194 (13833270.398) | 379066.795 (583625.135) | 50.77 (51.274) | 1349.953 (411.495) | 405.006 (194.848) | 22094.709 (36893.24) | 1.242 (0.622) | 11.505 (15.437) | 2.013 (2.153) | 25960036.535 (20511157.366) | 5.564 (5.064) |
| | SMOGN | 2477.367 (453.462) | 123.262 (85.61) | 11969.874 (8672.033) | 4191.919 (972.52) | 14795899.44 (9233563.429) | 380365.974 (568255.764) | 50.615 (40.505) | 1342.113 (229.843) | 411.589 (246.251) | 14785.074 (22289.047) | 1.24 (0.922) | 10.304 (13.869) | 1.751 (2.148) | 25868632.476 (20132394.918) | 6.216 (4.349) |
| | WSMOTER | 2144.37 (357.838) | 123.478 (90.783) | 8337.861 (5503.207) | **2705.588 (469.048)** | 18396564.726 (15335163.144) | 383546.067 (584262.211) | 45.913 (48.298) | 1133.376 (149.84) | 321.923 (98.189) | 7873.349 (10802.499) | **0.263 (0.115)** | 9.102 (11.79) | 1.598 (1.829) | 25057588.615 (20344865.898) | 2.5 (3.021) |
| | G-SMOTER | 2312.018 (284.14) | **103.29 (68.257)** | 7813.816 (5962.692) | 2830.653 (324.49) | 18732280.531 (13900104.982) | 395086.438 (597406.663) | 45.926 (40.395) | 944.259 (247.455) | 310.932 (106.89) | **4014.111 (5050.882)** | 0.291 (0.118) | 9.562 (12.731) | 1.502 (1.863) | 25157588.615 (20344865.898) | 3.076 (3.88) |
| | DAVID | 3243.745 (605.703) | 951.449 (618.198) | 394455.063 (249480.83) | 6280.156 (1282.365) | 20309229.004 (16549918.56) | 448793.315 (623442.57) | 112.37 (133.031) | 2493130.73 (46005.809) | 627.154 (410.414) | 216714.301 (445559.766) | 0.688 (0.28) | 11.915 (18.141) | 38.568 (51.737) | 27182079.895 (21291380.413) | 4.462 (4.169) |
| | KNNOR-REG | 2283.211 (289.559) | 111.598 (76.382) | 9368.686 (7900.754) | 2832.752 (435.948) | 18752687.214 (14360894.651) | 391747.858 (608570.172) | 49.385 (47.831) | 1205.362 (302.127) | 356.006 (141.084) | 11562.583 (16617.648) | 0.304 (0.155) | 9.77 (13.154) | 1.533 (1.835) | 25905696.216 (20724694.494) | 3.529 (4.802) |
| | CARTGen-IR | 2242.262 (304.122) | 115.159 (84.221) | **7237.189 (5866.996)** | 2765.906 (578.441) | **9777311.079 (7080594.562)** | **306885.035 (548003.586)** | **41.407 (39.829)** | 1725.244 (411.988) | — | 9196.557 (12388.776) | 0.291 (0.15) | **8.941 (13.217)** | **1.39 (1.599)** | 24286622.221 (19592553.796) | **2.11 (2.555)** |

Table 10: Best DW-RMSE scores (mean ± std. dev.) across learners, strategies, and datasets. Best results per learner are bolded. The overall best per dataset is shaded in light blue.

| Learner | Strategy | abalone | airfoil | availablePower | cpuSm | ele-1 | forestFires | fuelConsumption | heat | housingBoston | maxTorque | mortgage | sensory | servo | strikes | treasury |
|---|---|---|---|---|---|---|---|---|---|---|---|---|---|---|---|---|
| RF | None | 2.92 (0.198) | 2.123 (0.178) | 10.127 (3.588) | 3.615 (0.18) | 924.336 (137.244) | 235.845 (222.075) | 0.676 (0.302) | 3.361 (0.341) | 4.253 (1.193) | 19.157 (6.816) | 0.193 (0.027) | 1.012 (0.047) | 1.022 (0.256) | 1199.886 (290.26) | 0.304 (0.079) |
| | RU | 3.067 (0.299) | 11.131 (0.847) | 27.704 (1.806) | 19.296 (2.36) | 1173.218 (109.322) | 285.517 (140.677) | 1.461 (0.116) | 14.583 (0.81) | 7.18 (1.379) | 65.831 (6.575) | 0.338 (0.035) | 1.205 (0.425) | 1.22 (0.539) | 1865.492 (137.6) | 0.441 (0.094) |
| | RO | 2.894 (0.208) | 2.236 (0.191) | **9.522 (3.192)** | 3.679 (0.259) | 915.572 (138.442) | 234.203 (219.853) | 0.665 (0.301) | 3.283 (0.349) | 4.219 (1.038) | 17.552 (5.66) | 0.197 (0.031) | 1.006 (0.056) | 0.878 (0.267) | 1204.221 (277.818) | 0.309 (0.092) |
| | WERCS | **2.729 (0.145)** | 3 (0.287) | 10.091 (3.791) | 3.431 (0.169) | 900.222 (113.68) | 223.233 (207.953) | 0.634 (0.279) | 3.583 (0.275) | 3.937 (1.445) | 18.973 (6.365) | 0.296 (0.029) | 0.994 (0.048) | 0.957 (0.287) | 1181.453 (275.702) | 0.432 (0.197) |
| | GN | 2.764 (0.186) | 2.962 (0.31) | 11.509 (3.076) | 3.67 (0.117) | 944.196 (121.59) | 228.284 (217.014) | 0.713 (0.27) | 5.09 (0.401) | 4.199 (0.998) | 23.715 (6.762) | 0.227 (0.036) | 0.98 (0.067) | 0.861 (0.186) | 1203.569 (269.139) | 0.366 (0.092) |
| | SMOTER | 2.962 (0.189) | 2.242 (0.19) | 10.451 (3.954) | 3.731 (0.295) | 927.668 (118.191) | 232.165 (222.47) | 0.661 (0.3) | 3.219 (0.355) | 4.131 (1.157) | 18.174 (5.002) | 0.214 (0.042) | 1.052 (0.061) | 0.942 (0.309) | 1203.622 (279.281) | 0.333 (0.097) |
| | SMOGN | 3.006 (0.186) | 2.361 (0.191) | 11.274 (5.565) | 3.621 (0.15) | 936.438 (133.245) | 233.534 (214.512) | 0.661 (0.301) | 3.264 (0.383) | 4.059 (0.975) | 16.976 (3.792) | 0.229 (0.03) | 1.011 (0.056) | 0.885 (0.239) | 1208.075 (269.481) | 0.349 (0.081) |
| | WSMOTER | 2.767 (0.192) | **2.049 (0.206)** | 9.788 (3.467) | 3.48 (0.165) | 902.408 (150.317) | 231.265 (221.475) | 0.639 (0.296) | 3.072 (0.327) | 3.877 (0.983) | 18.032 (5.732) | **0.19 (0.033)** | 0.957 (0.04) | 0.943 (0.254) | 1197.397 (290.735) | 0.281 (0.074) |
| | G-SMOTER | 2.902 (0.201) | 2.09 (0.172) | 10.02 (3.651) | 3.541 (0.168) | 903.714 (131.571) | 233.809 (222.232) | 0.661 (0.3) | 3.134 (0.366) | 4.159 (1.13) | 18.017 (6.746) | 0.195 (0.034) | 0.981 (0.034) | 0.905 (0.287) | 1196.563 (285.214) | 0.308 (0.084) |
| | DAVID | 3.402 (0.237) | 5.336 (0.489) | 66.921 (15.534) | 4.92 (0.389) | 922.214 (124.793) | 233.284 (214.59) | 0.986 (0.613) | 45.255 (1.942) | 5.244 (1.469) | 40.225 (44.842) | 0.275 (0.04) | 1.101 (0.083) | 3.37 (0.467) | 1196.801 (281.741) | 0.396 (0.103) |
| | KNNOR-REG | 2.857 (0.187) | 2.087 (0.169) | 9.955 (3.623) | 3.586 (0.19) | 890.623 (138.064) | 233.137 (222.583) | 0.666 (0.298) | 3.17 (0.31) | 4.279 (1.16) | 19.697 (6.647) | 0.194 (0.026) | 1.005 (0.063) | 1.046 (0.329) | 1194.647 (281.591) | 0.3 (0.068) |
| | CARTGen-IR | 2.827 (0.167) | 2.074 (0.168) | 9.641 (3.443) | **3.429 (0.133)** | **815.268 (97.456)** | 225.149 (222.263) | **0.627 (0.29)** | **2.908 (0.384)** | **3.676 (1.048)** | **16.846 (6.433)** | 0.195 (0.035) | **0.95 (0.043)** | **0.836 (0.238)** | **1157.292 (280.529)** | **0.272 (0.083)** |
| SVR | None | 2.979 (0.187) | 3.363 (0.331) | 12.043 (3.177) | 4.72 (0.426) | 1193.388 (236.711) | 244.989 (225.197) | 0.819 (0.303) | 1.289 (0.422) | 4.118 (1.284) | 20.231 (11.535) | 0.091 (0.015) | 1.064 (0.037) | 0.565 (0.163) | 1268.993 (268.458) | 0.226 (0.043) |
| | RU | 3.036 (0.181) | 8.048 (0.605) | 17.426 (1.854) | 8.22 (0.435) | 1408.921 (198.08) | 227.137 (219.324) | 1.056 (0.185) | 8.592 (0.517) | 5.568 (0.628) | 39.835 (10.793) | 2.036 (0.226) | 1.266 (0.245) | 1.185 (0.721) | 1247.205 (271.103) | 2.596 (0.27) |
| | RO | 2.881 (0.219) | 4.498 (0.631) | 10.966 (2.495) | 4.685 (0.315) | 881.783 (102.697) | 220.44 (213.163) | 0.82 (0.293) | **0.686 (0.112)** | 4.103 (1.123) | **12.75 (3.073)** | 0.09 (0.015) | 1.053 (0.083) | **0.541 (0.157)** | 1169.337 (289.106) | 0.225 (0.042) |
| | WERCS | 2.752 (0.164) | 3.703 (0.414) | 11.646 (2.201) | 3.935 (0.279) | 883.457 (110.487) | **214.851 (209.279)** | 0.803 (0.275) | 0.72 (0.212) | 4.095 (1.067) | 16.62 (6.643) | 0.139 (0.032) | 1.068 (0.069) | 0.773 (0.283) | **124.905 (254.144)** | 0.28 (0.063) |
| | GN | 2.871 (0.219) | 5.243 (0.71) | 13.076 (2.85) | 5.299 (0.554) | 937.751 (79.562) | 217.107 (214.597) | 0.847 (0.239) | 2.561 (0.403) | 4.305 (0.935) | 20.47 (5.015) | 0.114 (0.022) | 1.058 (0.083) | 0.641 (0.156) | 149.565 (266.568) | 0.254 (0.062) |
| | SMOTER | 2.94 (0.209) | 4.48 (0.618) | 17.385 (2.576) | 4.626 (0.324) | 876.158 (74.85) | 223.178 (214.966) | 0.824 (0.287) | 0.896 (0.207) | 4.157 (1.083) | 20.086 (8.602) | 0.09 (0.015) | 1.088 (0.106) | 0.633 (0.167) | 1177.734 (287.773) | 0.226 (0.042) |
| | SMOGN | 3.129 (0.187) | 4.53 (0.633) | 19.303 (3.379) | 4.774 (0.218) | 909.489 (116.769) | 217.993 (215.087) | 0.82 (0.318) | 0.887 (0.198) | 4.25 (1.278) | 20.347 (10.2) | 0.108 (0.046) | 1.06 (0.078) | 0.568 (0.211) | 1125.461 (262.899) | 0.234 (0.044) |
| | WSMOTER | 2.674 (0.156) | **3.158 (0.334)** | **10.896 (2.837)** | 4.449 (0.425) | 990.113 (239.432) | 237.521 (224.49) | 0.826 (0.3) | 0.84 (0.301) | 3.865 (1.017) | 17.599 (7.451) | **0.088 (0.014)** | **1.005 (0.032)** | 0.585 (0.159) | 1232.223 (277.49) | 0.222 (0.034) |
| | G-SMOTER | 2.885 (0.183) | 3.299 (0.374) | 11.862 (3.235) | 4.468 (0.34) | 1097.764 (237.155) | 244.231 (225.109) | 0.806 (0.299) | 1.041 (0.35) | 4 (1.195) | 16.325 (6.088) | 0.091 (0.015) | 1.034 (0.029) | 0.562 (0.151) | 1261.193 (271.767) | **0.221 (0.043)** |
| | DAVID | 4.428 (0.564) | 5.629 (0.835) | 71.247 (14.023) | 6.45 (0.773) | 1193.388 (236.711) | 258.632 (164.379) | 1.118 (0.643) | 37.555 (13.323) | 4.358 (0.733) | 44.886 (50.002) | 0.181 (0.025) | 1.32 (0.219) | 3.339 (0.494) | 1181.84 (270.945) | 0.337 (0.078) |
| | KNNOR-REG | 2.814 (0.167) | 3.36 (0.321) | 11.945 (3.174) | 4.444 (0.413) | 1068.214 (262.096) | 244.316 (225.01) | 0.813 (0.297) | 1.407 (0.367) | 4.074 (1.267) | 20.332 (11.229) | 0.096 (0.024) | 1.059 (0.043) | 0.574 (0.166) | 1238.466 (274.88) | 0.232 (0.047) |
| | CARTGen-IR | **2.597 (0.158)** | 3.301 (0.31) | 12.195 (3.166) | **3.881 (0.265)** | 827.663 (123.126) | 218.193 (213.738) | **0.764 (0.276)** | 1.096 (0.266) | **3.843 (1.18)** | **16.505 (4.212)** | 0.092 (0.014) | 1.011 (0.068) | 0.561 (0.264) | 1142.667 (265.98) | 0.233 (0.044) |
| XGBoost | None | 2.969 (0.179) | 1.614 (0.158) | 9.518 (3.062) | 3.488 (0.391) | 1055.46 (189.036) | 240.286 (217.404) | 0.62 (0.24) | **1.656 (0.117)** | 3.955 (0.919) | 8.153 (4.392) | 0.192 (0.036) | 0.923 (0.038) | 0.447 (0.145) | 1238.849 (250.243) | 0.334 (0.054) |
| | RU | 3.243 (0.226) | 8.37 (0.852) | 25.208 (7.247) | 6.174 (0.842) | 1133.98 (140.053) | 371.205 (148.127) | 1.647 (0.254) | 10.312 (0.788) | 7.561 (1.471) | 49.59 (10.673) | 0.355 (0.081) | 1.32 (0.219) | 1.529 (0.581) | 2316.066 (346.842) | 0.469 (0.093) |
| | RO | 2.906 (0.165) | 1.726 (0.301) | 9.958 (2.979) | 3.619 (0.445) | 986.264 (176.578) | 232.777 (212.071) | 0.626 (0.276) | 1.908 (0.102) | 3.8 (0.534) | 10.427 (4.703) | 0.187 (0.026) | 0.897 (0.029) | **0.444 (0.133)** | 1252.491 (257.86) | 0.308 (0.077) |
| | WERCS | 2.829 (0.158) | 2.235 (0.28) | 11.265 (3.412) | 3.583 (0.346) | 990.351 (196.679) | 239.831 (202.87) | 0.658 (0.244) | 2.282 (0.092) | 3.621 (0.55) | 10.903 (4.483) | 0.285 (0.044) | 0.937 (0.064) | 0.824 (0.452) | 1208.675 (295.492) | 0.446 (0.132) |
| | GN | **2.81 (0.172)** | 2.389 (0.298) | 11.465 (3.127) | 3.8 (0.21) | 992.515 (165.465) | 239.246 (202.604) | 0.722 (0.265) | 4.111 (0.222) | 3.914 (0.422) | 18.687 (7.229) | 0.249 (0.045) | 0.919 (0.074) | 0.507 (0.133) | 1272.351 (208.576) | 0.354 (0.078) |
| | SMOTER | 3.153 (0.186) | 1.795 (0.292) | 11.347 (4.139) | 3.809 (0.123) | 1010.044 (180.685) | 233.312 (216.032) | 0.646 (0.288) | 2.187 (0.254) | 4.098 (0.848) | 14.027 (9.235) | 0.21 (0.029) | 0.944 (0.024) | 0.547 (0.209) | 1245.079 (242.758) | 0.367 (0.082) |
| | SMOGN | 3.12 (0.187) | 1.919 (0.24) | 12.367 (4.43) | 3.842 (0.209) | 1015.726 (140.732) | 235.342 (211.526) | 0.639 (0.289) | 2.205 (0.207) | 3.921 (0.773) | 11.856 (7.763) | 0.229 (0.03) | 0.894 (0.024) | 0.454 (0.261) | 1260.953 (235.563) | 0.394 (0.066) |
| | WSMOTER | 2.811 (0.215) | 1.653 (0.217) | 8.692 (2.713) | **3.317 (0.096)** | 1005.623 (196.776) | 238.441 (221.651) | 0.606 (0.272) | 1.989 (0.09) | **3.341 (0.46)** | **7.129 (2.393)** | **0.181 (0.043)** | 0.888 (0.054) | 0.481 (0.137) | 1229.51 (234.82) | 0.276 (0.066) |
| | G-SMOTER | 2.93 (0.181) | **1.582 (0.174)** | 8.505 (2.549) | 3.383 (0.087) | 1014.73 (173.499) | 239.669 (222.817) | 0.609 (0.278) | 1.798 (0.149) | 3.525 (0.641) | 9.031 (5.366) | 0.184 (0.026) | 0.905 (0.041) | 0.458 (0.162) | 1213.51 (266.302) | **0.276 (0.066)** |
| | DAVID | 3.599 (0.289) | 4.561 (0.426) | 60.275 (16.373) | 4.893 (0.498) | 1055.46 (189.036) | 273.904 (210.792) | 0.884 (0.526) | 25.595 (2.767) | 4.543 (0.978) | 28.123 (42.066) | 0.274 (0.051) | 1.037 (0.131) | 3.363 (0.432) | 1308.543 (269.219) | 0.383 (0.091) |
| | KNNOR-REG | 2.935 (0.186) | 1.634 (0.152) | 9.112 (3.134) | 3.42 (0.12) | 1006.931 (175.13) | 235.075 (221.809) | 0.631 (0.301) | 2.057 (0.192) | 3.492 (0.654) | 9.832 (4.621) | 0.187 (0.025) | 0.914 (0.056) | 0.446 (0.148) | 1228.85 (273.657) | 0.322 (0.086) |
| | CARTGen-IR | 2.874 (0.166) | 1.659 (0.203) | **8.124 (2.951)** | 3.402 (0.161) | 868.638 (140.029) | 232.449 (223.512) | **0.581 (0.249)** | 2.417 (0.138) | 3.634 (0.407) | 10.251 (4.151) | 0.182 (0.026) | **0.871 (0.014)** | 0.46 (0.127) | **1185.64 (251.872)** | 0.299 (0.072) |

Table 11: Best DW-SERA scores (mean ± std. dev.) across learners, strategies, and datasets. Best results per learner are bolded. The overall best per dataset is shaded in light blue.

| Learner | Strategy | abalone | airfoil | availablePower | cpuSim | dc-1 | forestFires | fuelConsumption | heat | housingBoston | maxTorque | mortgage | sensory | servo | strikes | treasury |
|---|---|---|---|---|---|---|---|---|---|---|---|---|---|---|---|---|
| RF | None | 3063.686 (315.567) | 696.008 (105.798) | 14713.931 (9766.508) | 9961.877 (911.263) | 34293590.29 (8683724.258) | 406490.503 (587012.675) | 75.245 (59.643) | 6799.771 (1201.566) | 869.292 (384.754) | 52087.907 (32284.98) | 3.469 (1.019) | 50.93 (5.608) | 10.414 (4.414) | 3075888.253 (1644461.9) | 8.123 (3.204) |
| | RU | 4299.561 (1516.909) | 21220.032 (2621.136) | 156095.039 (16578.953) | 499994.338 (127210.123) | 78468121.194 (12720372.17) | 1679923.024 (532266.52) | 485.048 (86.975) | 172591.69 (18146.244) | 3589.373 (1570.649) | 835750.859 (150587.857) | 15.99 (3.263) | 80.02 (39.502) | 24.332 (17.003) | 275477357.56 (68380758.65) | 20.974 (6.348) |
| | RO | 3065.51 (301.521) | 783.79 (151.284) | **13100.489 (7768.404)** | 10232.126 (1202.465) | 34821453.445 (9307480.269) | 408137.772 (573676.113) | 6467.945 (236.04) | 9215.508 (1113.08) | 847.403 (322.089) | 44443.819 (29918.945) | 3.628 (1.205) | 50.616 (4.889) | 8.376 (3.603) | 313892002.18 (16255770.193) | 8.414 (3.787) |
| | WERCS | 2959.675 (207.059) | 1479.499 (224.23) | 16557.348 (10317.141) | 9537.697 (809.029) | 34593510.668 (7928284.03) | 401364.681 (537730.289) | 72.764 (51.549) | 6696.483 (2137.052) | 782.165 (330.776) | 57050.658 (294789.958) | 7.703 (1.676) | 51.558 (6.942) | 12.133 (5.471) | 35727168.08 (14874742.142) | 16.906 (16.908) |
| | GN | 3053.995 (250.21) | 1423.719 (262.475) | 20407.339 (8918.28) | 10518.551 (678.645) | 44669338.558 (13046225.859) | 439062.939 (549014.415) | 90.507 (53.978) | 6268.72 (1262.93) | 903.787 (290.339) | 83385.485 (38262.197) | 5.287 (1.958) | 51.187 (4.865) | 8.693 (3.271) | 32966239.557 (15518535.774) | 11.315 (4.685) |
| | SMOTER | 3618.018 (215.39) | 787.967 (149.989) | 16147.696 (10926.356) | 10415.504 (1347.517) | 39210217.772 (10162457.362) | 403192.372 (581634.279) | 73.009 (54.402) | 6451.886 (1385.662) | 842.75 (323.934) | 46815.903 (19426.14) | 4.573 (1.5) | 53.967 (6.865) | 9.236 (4.813) | 31075711.149 (16106083.976) | 10.19 (4.908) |
| | SMOGN | 3743.384 (354.3) | 883.306 (126.237) | 19886.543 (18814.497) | 9978.262 (708.448) | 41678512.651 (13277891.352) | 405600.051 (571410.523) | 73.151 (57.031) | 5798.785 (1038.561) | 808.415 (293.586) | **40652.475 (13315.052)** | 5.347 (1.299) | 51.128 (5.233) | 8.271 (3.558) | 31768613.086 (15905984.564) | 11.234 (4.599) |
| | WSMOTER | **2926.104 (284.182)** | **659.873 (112.609)** | 13708.831 (8990.19) | 9460.025 (783.669) | 33614395.868 (8432997.905) | 403833.96 (588603.93) | 68.979 (55.062) | 5967.876 (1213.052) | 741.104 (300.192) | 46226.715 (27108.993) | **3.366 (1.241)** | **48.397 (4.229)** | 8.849 (4.366) | 30833899.785 (16545628.291) | 7.097 (2.796) |
| | G-SMOTER | 3048.844 (312.81) | 679.742 (102.513) | 14432.445 (9271.957) | 9638.079 (769.178) | 34185729.541 (8330204.17) | 433342.481 (553054.27) | 72.595 (58.184) | 5967.876 (1213.052) | 833.056 (355.506) | 46914.128 (29649.93) | 3.533 (1.337) | 49.287 (3.991) | 8.29 (4.269) | 30636518.495 (16713376.146) | 8.307 (3.465) |
| | DAVID | 4054.506 (460.701) | 4297.799 (637.376) | 576551.459 (228043.705) | 17690.194 (2167.274) | 329039421.165 (89691120.849) | 398913.094 (385324.72) | 179.862 (210.905) | 436497.205 (823621.156) | 1317.152 (620.311) | 436497.205 (823621.156) | 7.174 (1.601) | 60.742 (8.603) | 97.2 (28.104) | 40027896.083 (14666619.393) | 15.538 (9.117) |
| | KNNOR-REG | 3020.846 (271.948) | 678.972 (114.274) | 14272.09 (9630.148) | 9959.417 (948.97) | 33442589.029 (8894899.313) | **392015.386 (585516.798)** | 73.513 (57.844) | 6121.449 (1053.311) | 880.985 (370.019) | 54733.937 (31051.347) | 3.475 (0.993) | 51.194 (5.911) | 10.778 (4.965) | 30521607.079 (16196990.7) | 7.815 (2.637) |
| | CARTGen-IR | 2965.266 (256.169) | 685.842 (101.507) | 13449.293 (9041.481) | **9366.835 (522.581)** | 32903942.165 (8969112.849) | 334423589.029 (8894899.313) | **67.141 (54.483)** | **5166.143 (1235.402)** | **693.745 (240.824)** | 42500.815 (26985.19) | 3.564 (1.279) | 48.485 (3.976) | **7.159 (3.358)** | 30035702.278 (15272267.781) | **6.753 (3.628)** |
| SVR | None | 3133.162 (282.18) | 1704.437 (296.798) | 21215.813 (9631.805) | 15802.58 (2185.367) | 49266539.354 (18641734.057) | 428340.194 (620606.619) | 107.779 (74.885) | 900.773 (533.074) | 872.3 (421.234) | 66500.803 (69098.884) | 0.79 (0.221) | 59.754 (8.778) | 4.329 (2.194) | 34539305.394 (16242455.211) | 5.172 (1.557) |
| | RU | 3612.783 (394.585) | 9798.548 (1154.732) | 61161.186 (8480.996) | 62122.903 (5168.52) | 88637652.007 (16072261.585) | 406079 (558632.711) | 221.16 (47.231) | 56608.487 (4371.987) | 1877.492 (314.46) | 262055.677 (87620.53) | 342.887 (84.416) | 81.769 (24.858) | 21.985 (16.985) | 33579597.822 (16314199.926) | 539.475 (143.188) |
| | RO | 3344.041 (403.79) | 3261.675 (824.571) | 18405.417 (7631.957) | 16234.465 (2408.097) | 41674561.106 (11174845.348) | 372648.505 (531868.677) | 108.366 (71.212) | 275.025 (77.875) | 859.516 (352.209) | 28964.563 (10027.699) | 0.786 (0.224) | 60.644 (6.53) | 4.074 (2.513) | 31697957.904 (17121353.202) | 5.164 (1.543) |
| | WERCS | 2985.447 (211.841) | 2516.912 (578.197) | 20986.3 (6458.226) | 12955.123 (1820.231) | 36629377.569 (6945822.52) | 370343.631 (523685.734) | 108.796 (63.756) | 344.941 (161.371) | 895.165 (344.564) | 47997.32 (28301.153) | 1.859 (0.745) | 69.521 (13.508) | 9.868 (5.692) | 30447341.443 (15595640.417) | 7.435 (2.444) |
| | GN | 3262.398 (285.215) | 4435.079 (1086.47) | 27368.104 (10389.467) | 23266.086 (5293.89) | 50879027.197 (10183809.423) | 360985.455 (530011.997) | 126.649 (56.292) | 3657.072 (1011.7) | 973.575 (281.63) | 67821.754 (26798.749) | 1.236 (0.411) | 67.258 (10.918) | 5.915 (2.689) | 32401345.347 (18566737.826) | 6.387 (2.084) |
| | SMOTER | 3618.931 (385.485) | 3251.676 (832.143) | 43006.442 (10784.397) | 15942.046 (2364.68) | 39138062.137 (7934667.264) | 377672.574 (542139.848) | 109.88 (69.381) | 412.914 (170.561) | 878.228 (341.178) | 65759.843 (44729.199) | 1.179 (0.981) | 67.552 (13.695) | 5.053 (2.122) | 31505239.116 (18136764.505) | 5.209 (1.544) |
| | SMOGN | 4336.62 (692.399) | 3356.329 (812.403) | 54932.42 (16098.691) | 17393.287 (2107.584) | 46863318.03 (14481764.182) | 369011.4 (535417.864) | 110.642 (78.821) | 433.288 (164.435) | 927.536 (363.318) | 72051.909 (62370.346) | 1.179 (0.981) | 61.167 (7.238) | 4.717 (2.647) | 31918701.028 (18783531.702) | 5.49 (1.485) |
| | WSMOTER | **2750.039 (233.535)** | **1551.757 (300.481)** | 17782.557 (7859.63) | 14437.464 (2143.644) | 37220480.484 (14756629.173) | 413361.411 (603550.207) | 107.893 (72.968) | 404.143 (261.261) | **773.885 (305.115)** | 48492.712 (33853.15) | **0.776 (0.227)** | 58.514 (7.03) | 4.439 (2.157) | 31994604.979 (16538180.958) | 5.056 (1.445) |
| | G-SMOTER | 3027.473 (245.27) | 1675.542 (310.466) | 20551.811 (9076.08) | 14666.592 (1768.217) | 43048327.443 (16894442.268) | 426090.929 (619007.225) | 105.114 (73.016) | 603.58 (385.345) | 823.834 (360.763) | 42590.325 (29226.378) | 0.792 (0.219) | 75.156 (12.165) | 4.322 (2.02) | 34006930.454 (16288424.7) | **5.016 (1.59)** |
| | DAVID | 8787.029 (2115.836) | 5412.664 (1540.198) | 654191.803 (218824.092) | 31092.589 (6999.086) | 863529757 (361954646) | 438357.865 (599729.578) | 220.399 (235.784) | 817084.746 (344579.114) | 1019.792 (307.716) | 540407.8 (996717.516) | 3.241 (0.761) | 57.581 (7.816) | 95.48 (28.302) | 81404690.361 (33467428.057) | 11.885 (7.165) |
| | KNNOR-REG | 2989.753 (256.174) | 1730.94 (301.995) | 21277.056 (9634.957) | 14476.208 (2053.143) | 42393208.827 (18263106.37) | 426632.926 (619860.391) | 106.746 (72.165) | 1038.27 (495.56) | 856.064 (414.12) | 66982.465 (67183.603) | 0.904 (0.399) | 60.403 (8.638) | 4.461 (2.209) | 32419559.887 (16659721.191) | 5.414 (1.758) |
| | CARTGen-IR | 2772.314 (205.791) | 1781.062 (287.259) | 22127.625 (10007.737) | 12205.585 (1701.126) | 30627661.746 (6506506.35) | 369156.735 (533866.515) | 97.96 (63.135) | 711.83 (327.564) | 784.982 (382.48) | 45293.097 (16095.379) | 0.833 (0.221) | 59.842 (8.444) | 4.359 (2.013) | 30114540.04 (15934306.007) | 5.646 (1.812) |
| XGBoost | None | 3211.277 (275.142) | 421.725 (68.497) | 12903.247 (6541.686) | 9497.453 (1638.331) | 44648501.494 (13543124.514) | 433786.134 (577340.058) | 63.886 (39.895) | 1811.424 (226.283) | 796.514 (340.985) | 11249.385 (9326.267) | 3.786 (1.12) | 47.341 (5.557) | 2.557 (1.255) | 35007168.022 (15999962.145) | 9.851 (1.899) |
| | RU | 4835.697 (954.647) | 12957.799 (2466.872) | 137634.049 (86370.603) | 31902.839 (9389.234) | 684770527.52 (19029705.793) | 3423449.838 (2388622.93) | 617.763 (220.525) | 96021.637 (15062.128) | 4238.555 (1963.969) | 508976.891 (209636.653) | 16.555 (3.411) | 120.842 (26.887) | 46.218 (36.904) | 902230589.846 (102877533.396) | 24.021 (9.081) |
| | RO | 3259.597 (226.457) | 476.07 (169.042) | 13670.078 (7098.552) | 10086.404 (1876.009) | 40760938.311 (12653498.345) | 439309.662 (588876.388) | 66.826 (48.715) | 2344.623 (204.228) | 746.926 (190.26) | 16144.843 (11421.348) | 3.387 (0.804) | 45.864 (5.181) | 2.466 (0.985) | 36010791.833 (16533971.216) | 8.91 (2.792) |
| | WERCS | 3295.531 (185.858) | 887.565 (243.994) | 19285.769 (9457.554) | 10462.355 (1477.205) | 41183340.716 (11831690.179) | 548942.337 (483910.089) | 80.938 (44.735) | 3997.627 (303.867) | 715.016 (215.886) | 19983.44 (11548.668) | 8.757 (3.962) | 52.826 (5.526) | 13.383 (17.417) | 39738459.495 (44774878.941) | 17.389 (8.18) |
| | GN | 3096.112 (236.127) | 1011.338 (251.779) | 19803.988 (10880.334) | 11136.063 (808.446) | 50216326.391 (12566078.503) | 526676.051 (520530.609) | 97.853 (53.071) | 11524.673 (1067.856) | 916.805 (207.08) | 52799.049 (19140.972) | 5.94 (1.875) | 52.431 (5.269) | 3.198 (1.544) | 42992739.498 (15808988.569) | 11.86 (3.835) |
| | SMOTER | 4063.045 (266.769) | 520.186 (166.656) | 18247.342 (12624.62) | 10773.253 (584.444) | 45495969.55 (12182485.884) | 413876.439 (548848.654) | 69.283 (43.421) | 3218.123 (625.801) | 874.406 (319.815) | 21601.532 (21968.385) | 4.685 (1.119) | 52.531 (6.062) | 3.555 (2.396) | 36407874.255 (16099604.962) | 12.533 (3.558) |
| | SMOGN | 4028.257 (386.108) | 593.567 (141.467) | 22329.708 (13425.669) | 1079.3 (905.44) | 49952292.218 (15709157.054) | 486010.575 (513307.117) | 70.892 (53.128) | 3287.337 (526.036) | 823.324 (298.529) | 33591.808 (3992.33) | 5.6 (1.365) | 45.34 (3.312) | 2.853 (2.965) | 37472415.216 (17032090.828) | 14.3 (3.088) |
| | WSMOTER | 3062.493 (353.109) | 452.197 (93.053) | 11265.359 (5367.456) | 8808.972 (468.837) | 41357520.429 (13202960.592) | 490309.662 (588876.388) | 62.596 (47.418) | 2702.361 (231.521) | **610.88 (180.497)** | **8298.074 (4412.356)** | 3.411 (1.088) | 45.189 (4.711) | 2.747 (1.324) | 33399123.308 (17903244.522) | **7.337 (2.194)** |
| | G-SMOTER | 3131.928 (268.063) | **407.463 (76.815)** | 10793.339 (5310.669) | 9066.62 (585.782) | 41614455.932 (11892889.465) | 470014.389 (574011.262) | 64.179 (44.371) | 2127.186 (304.821) | 649.741 (216.466) | 13445.144 (9901.135) | 3.259 (0.955) | 45.485 (4.291) | 2.668 (1.357) | 34206650.029 (15066124.085) | 8.667 (3.029) |
| | DAVID | 4626.768 (581.155) | 3256.593 (753.657) | 18193.805 (29706.618) | 18193.805 (29706.618) | 41376123.988 (11853150.645) | 715992.967 (757284.92) | 146.946 (161.259) | 348548.201 (78033.023) | 1132.969 (466.031) | 296695.24 (596691.894) | 7.907 (2.704) | 63.412 (11.067) | 96.741 (27.238) | 53772661.548 (250861116.547) | 14.888 (6.586) |
| | KNNOR-REG | 3216.178 (262.747) | 445.828 (71.39) | 12522.729 (7219.348) | 9314.888 (515.396) | 41376123.988 (11853150.645) | 420521.129 (584041.049) | 64.371 (44.371) | 2849.068 (446.619) | 646.373 (219.594) | 14628.086 (10600.758) | 3.589 (0.82) | 46.576 (5.106) | 2.524 (1.446) | 34245056.462 (15922972.894) | 9.564 (3.732) |
| | CARTGen-IR | **3036.923 (260.672)** | 465.474 (99.858) | **9982.248 (6136.984)** | 9354.382 (704.922) | 35902236.762 (11882600.681) | 471193.347 (588393.216) | **58.864 (40.91)** | 3925.756 (456.985) | 691.773 (123.001) | 16169.796 (10889.623) | 3.514 (0.867) | **44.558 (3.427)** | 2.585 (1.048) | 32256480.806 (14715460.832) | 8.426 (2.757) |

Table 12: Best preprocessing strategy + learner combination per dataset and metric

| Dataset | RMSE | RW-RMSE | SERA | DW-RMSE | DW-SERA |
|---|---|---|---|---|---|
| abalone | None + SVR | G-SMOTER + XGBoost | WSMOTER + SVR | CARTGen-IR + SVR | WSMOTER + SVR |
| airfoil | G-SMOTER + XGBoost | G-SMOTER + XGBoost | G-SMOTER + XGBoost | G-SMOTER + XGBoost | G-SMOTER + XGBoost |
| availablePower | CARTGen-IR + XGBoost | CARTGen-IR + XGBoost | CARTGen-IR + XGBoost | CARTGen-IR + XGBoost | CARTGen-IR + XGBoost |
| cpuSm | G-SMOTER + XGBoost | WSMOTER + XGBoost | WSMOTER + XGBoost | WSMOTER + XGBoost | WSMOTER + XGBoost |
| ele-1 | KNNOR-REG + RF | CARTGen-IR + XGBoost | CARTGen-IR + RF | CARTGen-IR + RF | CARTGen-IR + SVR |
| forestFires | CARTGen-IR + SVR | WERCS + SVR | CARTGen-IR + XGBoost | WERCS + SVR | GN + SVR |
| fuelConsumption | CARTGen-IR + XGBoost | CARTGen-IR + XGBoost | CARTGen-IR + XGBoost | CARTGen-IR + XGBoost | CARTGen-IR + XGBoost |
| heat | RO + SVR | RO + SVR | RO + SVR | RO + SVR | RO + SVR |
| housingBoston | WSMOTER + XGBoost | CARTGen-IR + RF | CARTGen-IR + XGBoost | WSMOTER + XGBoost | WSMOTER + XGBoost |
| maxTorque | WSMOTER + XGBoost | WSMOTER + XGBoost | WSMOTER + XGBoost | WSMOTER + XGBoost | WSMOTER + XGBoost |
| mortgage | RO + SVR | WERCS + SVR | RO + SVR | WSMOTER + SVR | WSMOTER + SVR |
| sensory | CARTGen-IR + XGBoost | CARTGen-IR + XGBoost | CARTGen-IR + XGBoost | CARTGen-IR + XGBoost | CARTGen-IR + XGBoost |
| servo | RO + SVR | RO + XGBoost | CARTGen-IR + XGBoost | RO + XGBoost | RO + XGBoost |
| strikes | CARTGen-IR + SVR | SMOGN + SVR | SMOGN + SVR | WERCS + SVR | CARTGen-IR + RF |
| treasury | WSMOTER + SVR | WSMOTER + SVR | WSMOTER + SVR | G-SMOTER + SVR | G-SMOTER + SVR |

### A.1.3 WILCOXON SIGNED-RANK TEST RESULTS

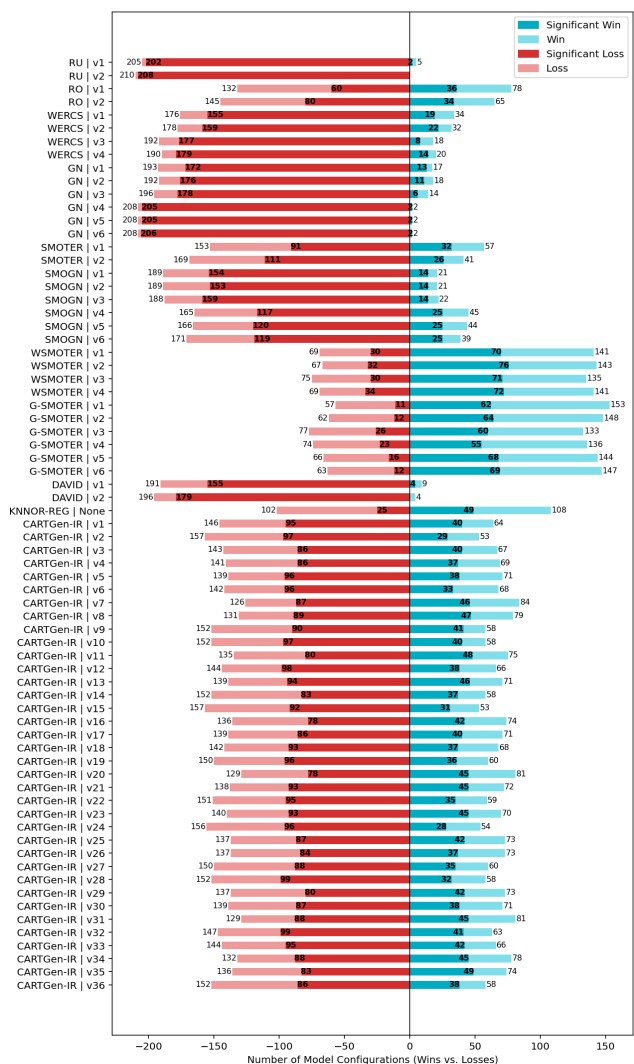

Figure 4: RMSE results by (significant) wins/losses vs. baseline scenario

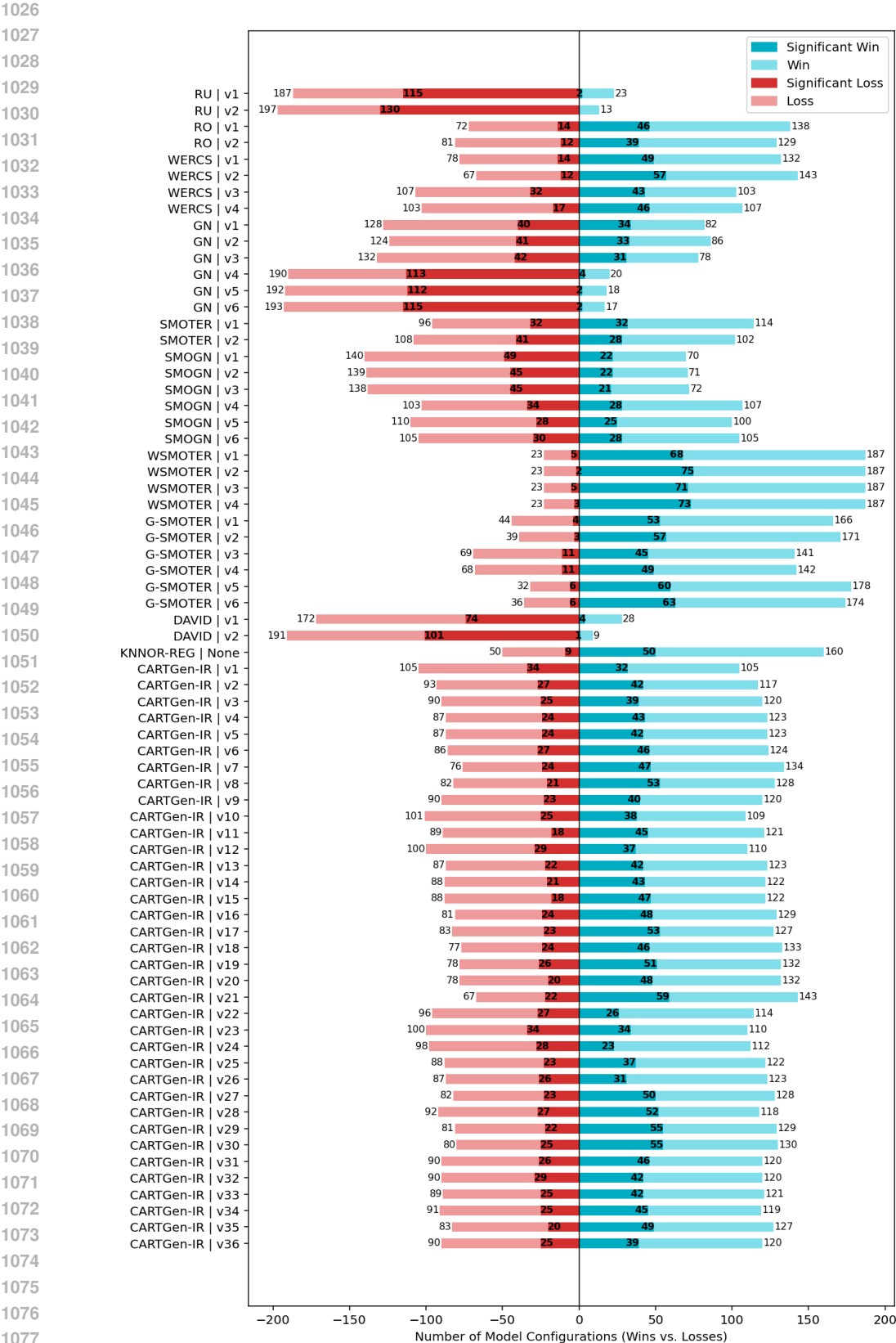

Figure 5: RW-RMSE results by (significant) wins/losses vs. baseline scenario

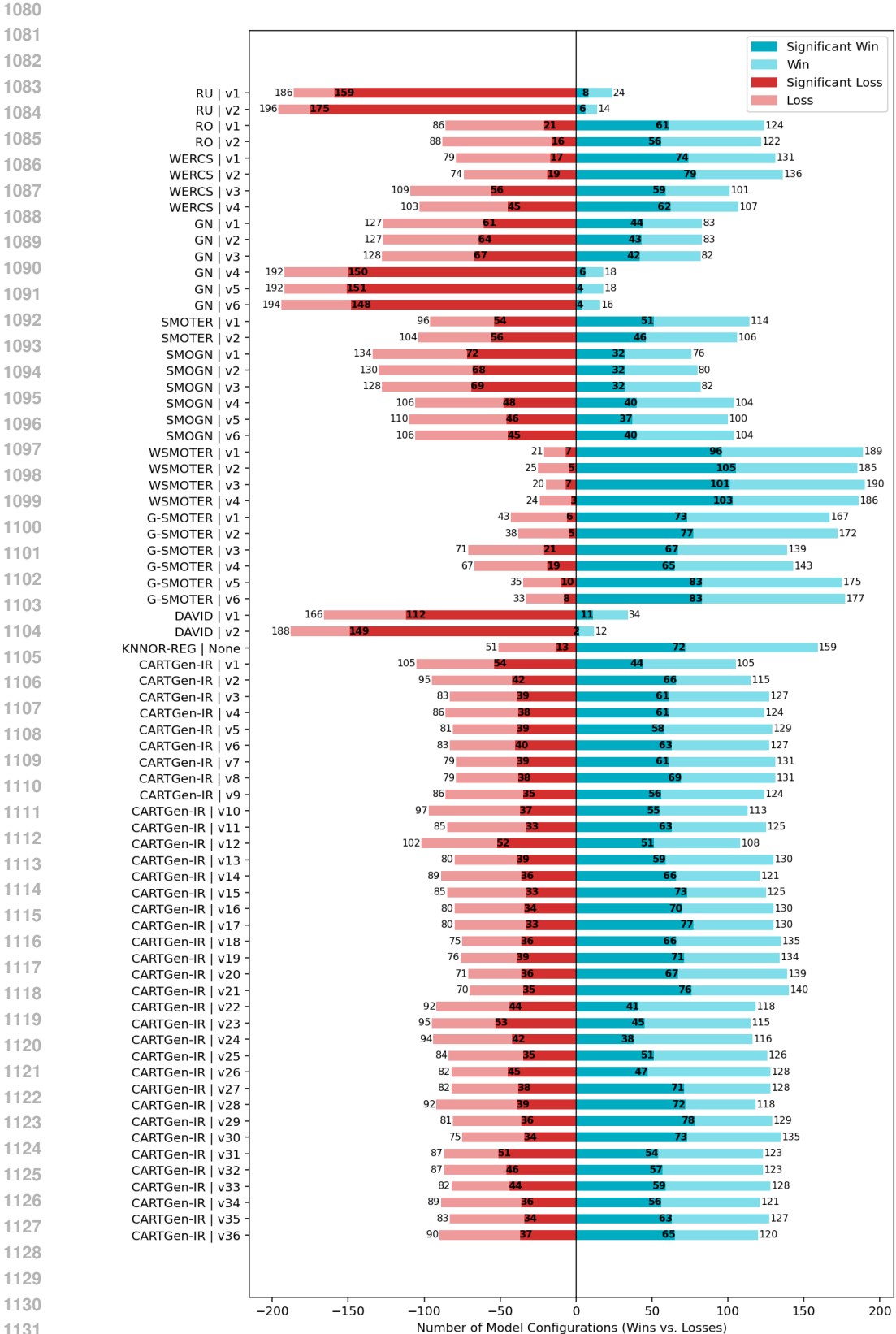

Figure 6: SERA results by (significant) wins/losses vs. baseline scenario

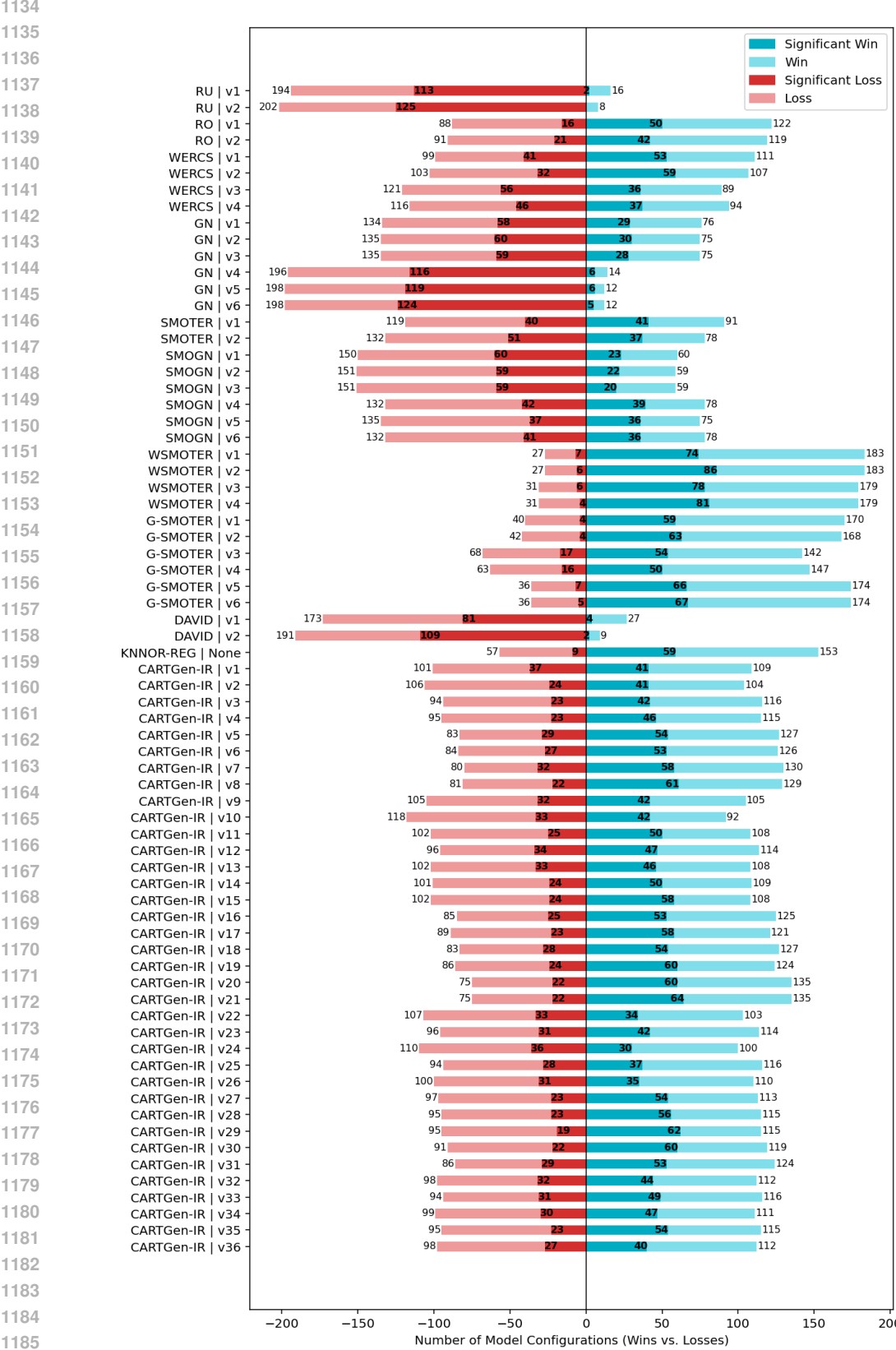

Figure 7: DW-RMSE results by (significant) wins/losses vs. baseline scenario

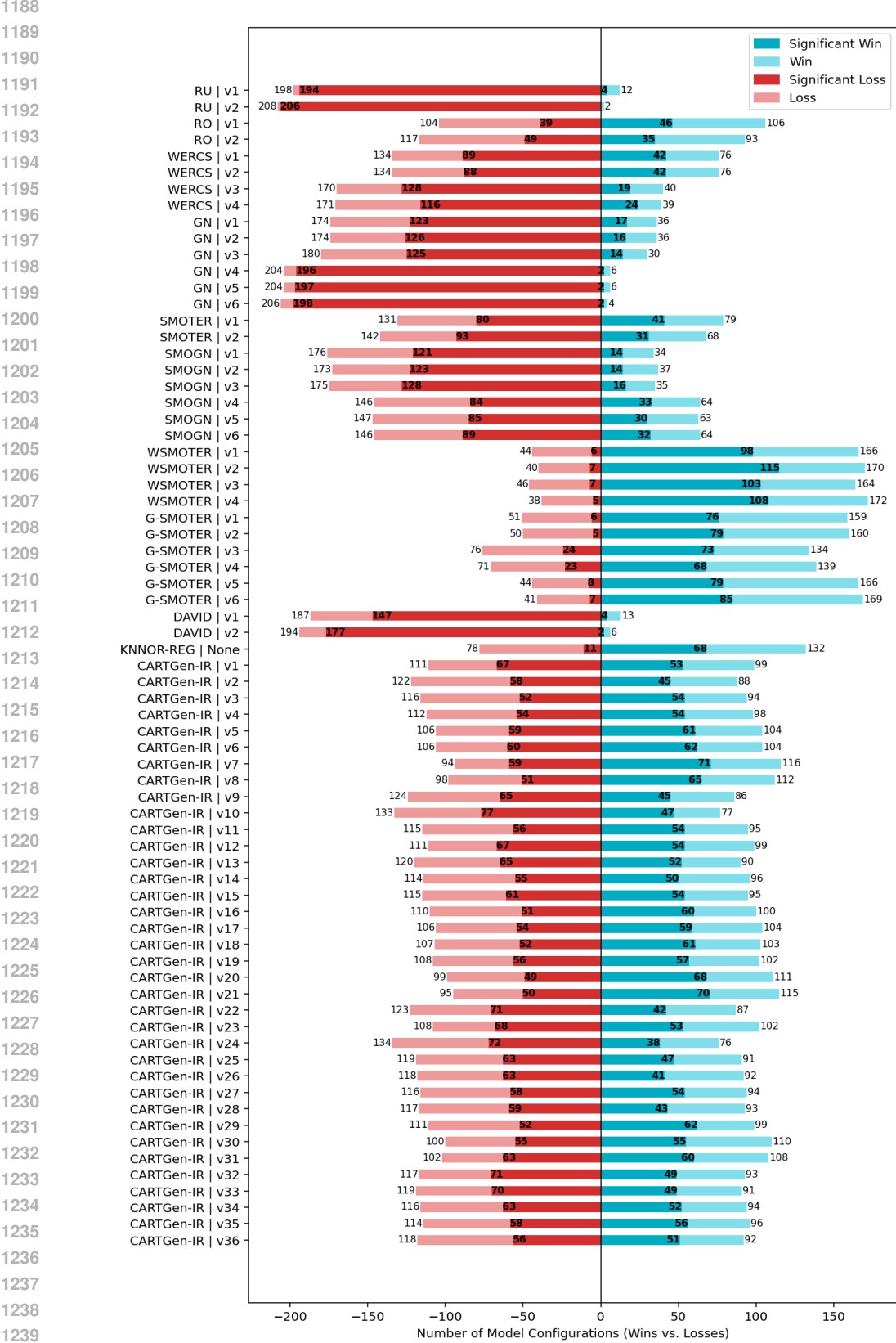

Figure 8: DW-SERA results by (significant) wins/losses vs. baseline scenario

Table 13: RMSE results by (significant) wins/losses vs. baseline scenario

| Strategy | RF (Max. Wins = 60) | | SVR (Max. Wins = 90) | | XGBoost (Max. Wins = 60) | |
|---|---|---|---|---|---|---|
| | Win | Loss | Win | Loss | Win | Loss |
| RU — v1 | 0(0) | 60(60) | 5(2) | 85(82) | 0(0) | 60(60) |
| RU — v2 | 0(0) | 60(60) | 0(0) | 90(88) | 0(0) | 60(60) |
| RO — v1 | 23(7) | 37(17) | 42(29) | 48(24) | 13(0) | 47(19) |
| RO — v2 | 19(6) | 41(17) | 34(26) | 56(42) | 12(2) | 48(21) |
| WERCS — v1 | 0(0) | 60(51) | 32(19) | 58(52) | 2(0) | 58(52) |
| WERCS — v2 | 2(0) | 58(53) | 30(22) | 60(54) | 0(0) | 60(52) |
| WERCS — v3 | 0(0) | 60(60) | 18(8) | 72(60) | 0(0) | 60(57) |
| WERCS — v4 | 0(0) | 60(60) | 20(14) | 70(61) | 0(0) | 60(58) |
| GN — v1 | 1(0) | 59(55) | 16(13) | 74(65) | 0(0) | 60(52) |
| GN — v2 | 2(0) | 58(56) | 16(11) | 74(67) | 0(0) | 60(53) |
| GN — v3 | 0(0) | 60(56) | 14(6) | 76(67) | 0(0) | 60(55) |
| GN — v4 | 0(0) | 60(60) | 2(2) | 88(85) | 0(0) | 60(60) |
| GN — v5 | 0(0) | 60(60) | 2(2) | 88(85) | 0(0) | 60(60) |
| GN — v6 | 0(0) | 60(60) | 2(2) | 88(86) | 0(0) | 60(60) |
| SMOTER — v1 | 15(6) | 45(28) | 38(26) | 52(27) | 4(0) | 56(36) |
| SMOTER — v2 | 15(5) | 45(31) | 24(21) | 66(41) | 2(0) | 58(39) |
| SMOGN — v1 | 0(0) | 60(46) | 21(14) | 69(62) | 0(0) | 60(46) |
| SMOGN — v2 | 0(0) | 60(44) | 21(14) | 69(62) | 0(0) | 60(47) |
| SMOGN — v3 | 0(0) | 60(46) | 22(14) | 68(62) | 0(0) | 60(51) |
| SMOGN — v4 | 17(5) | 43(33) | 22(19) | 68(52) | 6(1) | 54(32) |
| SMOGN — v5 | 17(5) | 43(32) | 22(19) | 68(53) | 5(1) | 55(35) |
| SMOGN — v6 | 15(4) | 45(31) | 20(20) | 70(52) | 4(1) | 56(36) |
| WSMOTER — v1 | 43(14) | 17(9) | 69(52) | 21(5) | 29(4) | 31(16) |
| WSMOTER — v2 | 43(19) | 17(8) | 66(50) | 24(11) | 34(7) | 26(13) |
| WSMOTER — v3 | 42(17) | 18(10) | 66(49) | 24(7) | 27(5) | 33(13) |
| WSMOTER — v4 | 43(20) | 17(11) | 66(47) | 24(9) | 32(5) | 28(14) |
| G-SMOTER — v1 | 38(11) | 22(2) | 78(49) | 12(5) | 37(2) | 23(4) |
| G-SMOTER — v2 | 35(12) | 25(3) | 77(50) | 13(5) | 36(2) | 24(4) |
| G-SMOTER — v3 | 36(9) | 24(5) | 66(43) | 24(14) | 31(8) | 29(7) |
| G-SMOTER — v4 | 35(12) | 25(1) | 66(41) | 24(15) | 35(2) | 25(7) |
| G-SMOTER — v5 | 34(16) | 26(5) | 70(49) | 20(1) | 40(3) | 20(10) |
| G-SMOTER — v6 | 39(14) | 21(2) | 69(49) | 21(1) | 39(6) | 21(9) |
| DAVID — v1 | 3(0) | 57(49) | 6(4) | 78(60) | 0(0) | 56(46) |
| DAVID — v2 | 4(0) | 56(55) | 0(0) | 84(74) | 0(0) | 56(50) |
| KNNOR-REG — None | 31(3) | 29(4) | 52(45) | 38(13) | 25(1) | 35(8) |
| CARTGen-IR — v1 | 9(3) | 51(30) | 40(34) | 50(38) | 15(3) | 45(27) |
| CARTGen-IR — v2 | 10(4) | 50(31) | 40(32) | 50(38) | 16(2) | 44(29) |
| CARTGen-IR — v3 | 18(6) | 42(25) | 38(35) | 52(41) | 14(4) | 46(27) |
| CARTGen-IR — v4 | 25(8) | 35(21) | 40(31) | 50(41) | 16(6) | 44(26) |
| CARTGen-IR — v5 | 16(7) | 44(26) | 37(31) | 53(39) | 10(3) | 50(34) |
| CARTGen-IR — v6 | 16(8) | 44(24) | 42(32) | 48(34) | 8(2) | 52(37) |
| CARTGen-IR — v7 | 24(8) | 36(23) | 41(32) | 49(37) | 13(5) | 47(28) |
| CARTGen-IR — v8 | 24(14) | 36(21) | 41(32) | 49(34) | 9(3) | 51(28) |
| CARTGen-IR — v9 | 16(6) | 44(24) | 34(30) | 56(40) | 8(2) | 52(22) |
| CARTGen-IR — v10 | 11(3) | 49(26) | 34(24) | 56(44) | 8(2) | 52(27) |
| CARTGen-IR — v11 | 22(8) | 38(22) | 31(30) | 59(42) | 14(2) | 46(22) |
| CARTGen-IR — v12 | 24(12) | 36(22) | 34(22) | 56(41) | 11(3) | 49(23) |
| CARTGen-IR — v13 | 17(4) | 43(29) | 44(31) | 46(36) | 10(3) | 50(31) |
| CARTGen-IR — v14 | 20(6) | 40(28) | 38(26) | 52(39) | 10(1) | 50(29) |
| CARTGen-IR — v15 | 25(9) | 35(22) | 45(32) | 45(38) | 14(5) | 46(27) |
| CARTGen-IR — v16 | 28(14) | 32(27) | 39(29) | 51(36) | 12(4) | 48(26) |
| CARTGen-IR — v17 | 10(6) | 50(19) | 40(33) | 50(34) | 8(2) | 52(37) |
| CARTGen-IR — v18 | 9(6) | 51(22) | 40(32) | 50(38) | 9(2) | 51(37) |
| CARTGen-IR — v19 | 22(13) | 38(17) | 40(33) | 50(35) | 13(2) | 47(28) |
| CARTGen-IR — v20 | 20(12) | 40(25) | 39(32) | 51(37) | 12(2) | 48(32) |
| CARTGen-IR — v21 | 13(5) | 47(19) | 34(30) | 56(42) | 11(2) | 49(22) |
| CARTGen-IR — v22 | 15(4) | 45(22) | 33(25) | 57(46) | 5(2) | 55(24) |
| CARTGen-IR — v23 | 26(9) | 34(19) | 34(32) | 56(40) | 14(1) | 46(19) |
| CARTGen-IR — v24 | 27(11) | 33(18) | 32(25) | 58(46) | 12(4) | 48(22) |
| CARTGen-IR — v25 | 13(4) | 47(29) | 44(32) | 46(34) | 11(1) | 49(30) |
| CARTGen-IR — v26 | 13(5) | 47(28) | 38(28) | 52(39) | 9(3) | 51(29) |
| CARTGen-IR — v27 | 26(8) | 34(19) | 42(32) | 48(33) | 13(5) | 47(26) |
| CARTGen-IR — v28 | 24(14) | 36(24) | 38(27) | 52(39) | 10(4) | 50(30) |
| CARTGen-IR — v29 | 10(6) | 50(21) | 41(28) | 49(38) | 8(1) | 52(36) |
| CARTGen-IR — v30 | 11(5) | 49(20) | 39(22) | 51(39) | 4(1) | 56(37) |
| CARTGen-IR — v31 | 21(12) | 39(19) | 39(28) | 51(39) | 13(2) | 47(29) |
| CARTGen-IR — v32 | 21(12) | 39(15) | 39(22) | 51(37) | 13(3) | 47(32) |
| CARTGen-IR — v33 | 20(5) | 40(23) | 35(28) | 55(40) | 5(2) | 55(25) |
| CARTGen-IR — v34 | 21(7) | 39(21) | 32(24) | 58(48) | 5(1) | 55(30) |
| CARTGen-IR — v35 | 25(9) | 35(20) | 34(30) | 56(41) | 14(3) | 46(19) |
| CARTGen-IR — v36 | 26(10) | 34(18) | 34(25) | 56(46) | 11(3) | 49(23) |

Table 14: RW-RMSE results by (significant) wins/losses vs. baseline scenario

| Strategy | RF (Max. Wins = 60) | | SVR (Max. Wins = 90) | | XGBoost (Max. Wins = 60) | |
|---|---|---|---|---|---|---|
| | Win | Loss | Win | Loss | Win | Loss |
| RU — v1 | 4(0) | 56(34) | 19(2) | 71(45) | 0(0) | 60(36) |
| RU — v2 | 0(0) | 60(40) | 13(0) | 77(50) | 0(0) | 60(40) |
| RO — v1 | 46(11) | 14(1) | 76(33) | 14(1) | 16(2) | 44(12) |
| RO — v2 | 41(8) | 19(1) | 71(28) | 19(2) | 17(3) | 43(9) |
| WERCS — v1 | 39(12) | 21(0) | 69(33) | 21(5) | 24(4) | 36(9) |
| WERCS — v2 | 51(17) | 9(0) | 69(36) | 21(2) | 23(4) | 37(10) |
| WERCS — v3 | 24(12) | 36(11) | 61(28) | 29(8) | 18(3) | 42(13) |
| WERCS — v4 | 28(12) | 32(4) | 66(30) | 24(5) | 13(4) | 47(8) |
| GN — v1 | 19(4) | 41(10) | 51(27) | 39(13) | 12(3) | 48(17) |
| GN — v2 | 20(4) | 40(9) | 52(27) | 38(12) | 14(2) | 46(20) |
| GN — v3 | 20(4) | 40(9) | 45(25) | 45(12) | 13(2) | 47(21) |
| GN — v4 | 0(0) | 60(34) | 19(4) | 71(41) | 1(0) | 59(38) |
| GN — v5 | 0(0) | 60(34) | 18(2) | 72(41) | 0(0) | 60(37) |
| GN — v6 | 1(0) | 59(33) | 16(2) | 74(46) | 0(0) | 60(36) |
| SMOTER — v1 | 35(4) | 25(8) | 64(28) | 26(6) | 15(0) | 45(18) |
| SMOTER — v2 | 33(6) | 27(8) | 56(22) | 34(12) | 13(0) | 47(21) |
| SMOGN — v1 | 15(0) | 45(13) | 48(22) | 42(14) | 7(0) | 53(22) |
| SMOGN — v2 | 15(0) | 45(10) | 48(22) | 42(14) | 8(0) | 52(21) |
| SMOGN — v3 | 21(0) | 39(10) | 46(21) | 44(14) | 5(0) | 55(21) |
| SMOGN — v4 | 32(6) | 28(8) | 63(22) | 27(8) | 12(0) | 48(18) |
| SMOGN — v5 | 30(3) | 30(4) | 63(22) | 27(6) | 7(0) | 53(18) |
| SMOGN — v6 | 35(6) | 25(4) | 60(22) | 30(7) | 10(0) | 50(19) |
| WSMOTER — v1 | 57(24) | 3(0) | 85(41) | 5(2) | 45(3) | 15(3) |
| WSMOTER — v2 | 60(30) | 0(0) | 84(39) | 6(0) | 43(6) | 17(2) |
| WSMOTER — v3 | 59(21) | 1(0) | 85(43) | 5(1) | 43(7) | 17(4) |
| WSMOTER — v4 | 59(28) | 1(0) | 85(40) | 5(0) | 43(5) | 17(3) |
| G-SMOTER — v1 | 49(14) | 11(0) | 82(39) | 8(0) | 35(0) | 25(4) |
| G-SMOTER — v2 | 50(16) | 10(0) | 81(39) | 9(0) | 40(2) | 20(3) |
| G-SMOTER — v3 | 44(12) | 16(0) | 69(30) | 21(8) | 28(3) | 32(3) |
| G-SMOTER — v4 | 46(14) | 14(0) | 68(30) | 22(9) | 28(5) | 32(2) |
| G-SMOTER — v5 | 56(18) | 4(0) | 83(38) | 7(0) | 39(4) | 21(6) |
| G-SMOTER — v6 | 54(20) | 6(0) | 83(38) | 7(0) | 37(5) | 23(6) |
| DAVID — v1 | 9(0) | 51(24) | 19(4) | 65(26) | 0(0) | 56(24) |
| DAVID — v2 | 7(1) | 53(31) | 2(0) | 82(40) | 0(0) | 56(30) |
| KNNOR-REG — None | 48(14) | 12(1) | 76(32) | 14(4) | 36(4) | 24(4) |
| CARTGen-IR — v1 | 26(5) | 34(5) | 53(26) | 37(17) | 26(1) | 34(12) |
| CARTGen-IR — v2 | 33(10) | 27(3) | 50(26) | 40(13) | 27(1) | 33(13) |
| CARTGen-IR — v3 | 37(7) | 23(4) | 51(26) | 39(17) | 22(1) | 38(13) |
| CARTGen-IR — v4 | 41(14) | 19(0) | 52(30) | 38(14) | 27(2) | 33(12) |
| CARTGen-IR — v5 | 42(12) | 18(0) | 58(29) | 32(12) | 20(1) | 40(17) |
| CARTGen-IR — v6 | 40(13) | 20(1) | 64(28) | 26(9) | 17(1) | 43(15) |
| CARTGen-IR — v7 | 41(14) | 19(2) | 61(29) | 29(11) | 17(2) | 43(12) |
| CARTGen-IR — v8 | 45(19) | 15(0) | 69(29) | 21(8) | 13(1) | 47(12) |
| CARTGen-IR — v9 | 42(12) | 18(1) | 57(27) | 33(15) | 21(0) | 39(9) |
| CARTGen-IR — v10 | 38(11) | 22(1) | 57(30) | 33(12) | 22(1) | 38(14) |
| CARTGen-IR — v11 | 43(11) | 17(1) | 56(26) | 34(14) | 21(2) | 39(10) |
| CARTGen-IR — v12 | 45(13) | 15(2) | 59(30) | 31(9) | 19(0) | 41(13) |
| CARTGen-IR — v13 | 43(11) | 17(0) | 61(31) | 29(11) | 19(0) | 41(13) |
| CARTGen-IR — v14 | 42(14) | 18(2) | 58(31) | 32(11) | 24(1) | 36(14) |
| CARTGen-IR — v15 | 47(14) | 13(1) | 61(31) | 29(12) | 26(2) | 34(11) |
| CARTGen-IR — v16 | 47(18) | 13(1) | 59(33) | 31(11) | 22(2) | 38(9) |
| CARTGen-IR — v17 | 40(15) | 20(1) | 64(24) | 26(7) | 16(1) | 44(15) |
| CARTGen-IR — v18 | 36(12) | 24(0) | 62(26) | 28(7) | 11(0) | 49(18) |
| CARTGen-IR — v19 | 43(21) | 17(0) | 62(23) | 28(7) | 16(1) | 44(11) |
| CARTGen-IR — v20 | 44(17) | 16(0) | 63(25) | 27(5) | 16(0) | 44(17) |
| CARTGen-IR — v21 | 44(12) | 16(1) | 59(28) | 31(12) | 19(3) | 41(8) |
| CARTGen-IR — v22 | 47(13) | 13(0) | 58(31) | 32(10) | 17(3) | 43(8) |
| CARTGen-IR — v23 | 46(18) | 14(1) | 59(28) | 31(14) | 24(2) | 36(9) |
| CARTGen-IR — v24 | 49(18) | 11(1) | 58(30) | 32(13) | 20(5) | 40(9) |
| CARTGen-IR — v25 | 46(11) | 14(1) | 65(35) | 25(10) | 22(0) | 38(13) |
| CARTGen-IR — v26 | 45(16) | 15(0) | 65(34) | 25(12) | 22(1) | 38(14) |
| CARTGen-IR — v27 | 51(12) | 9(0) | 64(34) | 26(9) | 17(2) | 43(11) |
| CARTGen-IR — v28 | 51(20) | 9(2) | 66(34) | 24(9) | 26(5) | 34(11) |
| CARTGen-IR — v29 | 37(6) | 23(1) | 64(20) | 26(10) | 13(0) | 47(16) |
| CARTGen-IR — v30 | 37(5) | 23(0) | 62(18) | 28(9) | 13(0) | 47(19) |
| CARTGen-IR — v31 | 42(15) | 18(1) | 62(20) | 28(6) | 18(2) | 42(16) |
| CARTGen-IR — v32 | 46(14) | 14(0) | 58(17) | 32(9) | 19(0) | 41(17) |
| CARTGen-IR — v33 | 45(14) | 15(2) | 60(33) | 30(13) | 23(3) | 37(8) |
| CARTGen-IR — v34 | 43(19) | 17(2) | 59(31) | 31(12) | 16(2) | 44(13) |
| CARTGen-IR — v35 | 48(19) | 12(2) | 60(32) | 30(11) | 21(4) | 39(9) |
| CARTGen-IR — v36 | 50(19) | 10(2) | 60(33) | 30(12) | 20(3) | 40(11) |

Table 15: SERA results by (significant) wins/losses vs. baseline scenario

| Strategy | RF (Max. Wins = 60) | | SVR (Max. Wins = 90) | | XGBoost (Max. Wins = 60) | |
|---|---|---|---|---|---|---|
| | Win | Loss | Win | Loss | Win | Loss |
| RU — v1 | 2(0) | 58(46) | 20(8) | 70(57) | 2(0) | 58(56) |
| RU — v2 | 0(0) | 60(57) | 14(6) | 76(62) | 0(0) | 60(56) |
| RO — v1 | 40(13) | 20(1) | 71(46) | 19(3) | 13(2) | 47(17) |
| RO — v2 | 38(10) | 22(2) | 69(43) | 21(2) | 15(3) | 45(12) |
| WERCS — v1 | 34(17) | 26(0) | 70(50) | 20(5) | 27(7) | 33(12) |
| WERCS — v2 | 45(20) | 15(0) | 69(54) | 21(3) | 22(5) | 38(16) |
| WERCS — v3 | 24(19) | 36(14) | 58(38) | 32(19) | 19(2) | 41(23) |
| WERCS — v4 | 35(16) | 25(9) | 58(40) | 32(17) | 14(6) | 46(19) |
| GN — v1 | 20(5) | 40(18) | 51(36) | 39(19) | 12(3) | 48(24) |
| GN — v2 | 22(4) | 38(19) | 50(37) | 40(18) | 11(2) | 49(27) |
| GN — v3 | 24(5) | 36(17) | 46(35) | 44(23) | 12(2) | 48(27) |
| GN — v4 | 0(0) | 60(45) | 18(6) | 72(52) | 0(0) | 60(53) |
| GN — v5 | 0(0) | 60(46) | 18(4) | 72(52) | 0(0) | 60(53) |
| GN — v6 | 0(0) | 60(42) | 16(4) | 74(55) | 0(0) | 60(51) |
| SMOTER — v1 | 38(4) | 22(16) | 62(45) | 28(11) | 14(2) | 46(27) |
| SMOTER — v2 | 36(6) | 24(16) | 57(40) | 33(14) | 13(0) | 47(26) |
| SMOGN — v1 | 21(0) | 39(22) | 49(32) | 41(22) | 6(0) | 54(28) |
| SMOGN — v2 | 22(0) | 38(18) | 49(32) | 41(23) | 9(0) | 51(27) |
| SMOGN — v3 | 28(0) | 32(18) | 49(32) | 41(22) | 5(0) | 55(29) |
| SMOGN — v4 | 30(5) | 30(12) | 63(35) | 27(9) | 11(0) | 49(27) |
| SMOGN — v5 | 30(3) | 30(9) | 64(34) | 26(10) | 6(0) | 54(27) |
| SMOGN — v6 | 31(6) | 29(8) | 62(34) | 28(10) | 11(0) | 49(27) |
| WSMOTER — v1 | 60(29) | 0(0) | 85(64) | 5(2) | 44(3) | 16(5) |
| WSMOTER — v2 | 60(36) | 0(0) | 85(62) | 5(0) | 40(7) | 20(5) |
| WSMOTER — v3 | 60(28) | 0(0) | 86(65) | 4(2) | 44(8) | 16(5) |
| WSMOTER — v4 | 59(33) | 1(0) | 86(65) | 4(0) | 41(5) | 19(3) |
| G-SMOTER — v1 | 48(14) | 12(0) | 84(57) | 6(0) | 35(2) | 25(6) |
| G-SMOTER — v2 | 49(18) | 11(0) | 84(57) | 6(0) | 39(2) | 21(5) |
| G-SMOTER — v3 | 45(14) | 15(3) | 68(50) | 22(13) | 26(3) | 34(5) |
| G-SMOTER — v4 | 45(15) | 15(2) | 68(49) | 22(13) | 30(1) | 30(4) |
| G-SMOTER — v5 | 53(21) | 7(0) | 84(54) | 6(0) | 38(8) | 22(10) |
| G-SMOTER — v6 | 54(22) | 6(0) | 85(55) | 5(0) | 38(6) | 22(8) |
| DAVID — v1 | 12(1) | 48(38) | 22(10) | 62(39) | 0(0) | 56(35) |
| DAVID — v2 | 8(2) | 52(45) | 4(0) | 80(58) | 0(0) | 56(46) |
| KNNOR-REG — None | 47(18) | 13(0) | 75(51) | 15(6) | 37(3) | 23(7) |
| CARTGen-IR — v1 | 26(6) | 34(8) | 55(37) | 35(26) | 24(1) | 36(20) |
| CARTGen-IR — v2 | 30(12) | 30(8) | 52(38) | 38(23) | 26(1) | 34(21) |
| CARTGen-IR — v3 | 38(7) | 22(7) | 53(37) | 37(26) | 24(1) | 36(20) |
| CARTGen-IR — v4 | 44(12) | 16(4) | 52(41) | 38(26) | 27(1) | 33(21) |
| CARTGen-IR — v5 | 41(14) | 19(2) | 63(42) | 27(17) | 19(1) | 41(27) |
| CARTGen-IR — v6 | 42(14) | 18(4) | 65(45) | 25(18) | 21(0) | 39(22) |
| CARTGen-IR — v7 | 42(12) | 18(2) | 61(42) | 29(15) | 18(2) | 42(19) |
| CARTGen-IR — v8 | 44(19) | 16(1) | 68(44) | 22(12) | 15(0) | 45(21) |
| CARTGen-IR — v9 | 39(19) | 21(1) | 58(45) | 32(21) | 23(1) | 37(15) |
| CARTGen-IR — v10 | 37(17) | 23(3) | 58(48) | 32(19) | 20(1) | 40(20) |
| CARTGen-IR — v11 | 43(16) | 17(4) | 57(44) | 33(18) | 27(1) | 33(17) |
| CARTGen-IR — v12 | 45(13) | 15(2) | 61(48) | 29(18) | 18(0) | 42(18) |
| CARTGen-IR — v13 | 44(12) | 16(0) | 61(46) | 29(18) | 24(0) | 36(21) |
| CARTGen-IR — v14 | 40(17) | 20(1) | 61(44) | 29(18) | 26(2) | 34(21) |
| CARTGen-IR — v15 | 44(14) | 16(2) | 62(45) | 28(18) | 25(2) | 35(19) |
| CARTGen-IR — v16 | 48(20) | 12(2) | 63(47) | 27(17) | 20(2) | 40(19) |
| CARTGen-IR — v17 | 42(17) | 18(3) | 63(39) | 27(11) | 19(0) | 41(21) |
| CARTGen-IR — v18 | 38(14) | 22(2) | 62(41) | 28(11) | 13(0) | 47(24) |
| CARTGen-IR — v19 | 43(24) | 17(0) | 63(38) | 27(14) | 19(1) | 41(19) |
| CARTGen-IR — v20 | 47(21) | 13(3) | 64(38) | 26(10) | 19(0) | 41(26) |
| CARTGen-IR — v21 | 40(17) | 20(0) | 59(47) | 31(20) | 22(2) | 38(16) |
| CARTGen-IR — v22 | 46(18) | 14(1) | 60(51) | 30(18) | 19(4) | 41(14) |
| CARTGen-IR — v23 | 43(22) | 17(2) | 59(45) | 31(17) | 28(3) | 32(15) |
| CARTGen-IR — v24 | 48(20) | 12(1) | 59(50) | 31(16) | 23(7) | 37(16) |
| CARTGen-IR — v25 | 46(15) | 14(2) | 68(51) | 22(15) | 21(0) | 39(19) |
| CARTGen-IR — v26 | 45(18) | 15(1) | 68(50) | 22(16) | 21(3) | 39(22) |
| CARTGen-IR — v27 | 51(16) | 9(0) | 68(49) | 22(17) | 20(2) | 40(19) |
| CARTGen-IR — v28 | 50(21) | 10(1) | 66(50) | 24(16) | 24(5) | 36(18) |
| CARTGen-IR — v29 | 39(8) | 21(5) | 66(33) | 24(16) | 13(0) | 47(23) |
| CARTGen-IR — v30 | 39(8) | 21(1) | 61(30) | 29(14) | 16(0) | 44(27) |
| CARTGen-IR — v31 | 43(17) | 17(3) | 65(32) | 25(10) | 18(2) | 42(22) |
| CARTGen-IR — v32 | 48(14) | 12(3) | 61(31) | 29(17) | 19(2) | 41(25) |
| CARTGen-IR — v33 | 46(17) | 14(3) | 59(51) | 31(21) | 23(3) | 37(14) |
| CARTGen-IR — v34 | 43(20) | 17(1) | 59(49) | 31(18) | 16(3) | 44(20) |
| CARTGen-IR — v35 | 46(22) | 14(5) | 62(51) | 28(16) | 21(5) | 39(15) |
| CARTGen-IR — v36 | 50(17) | 10(3) | 63(52) | 27(16) | 22(4) | 38(15) |

Table 16: DW-RMSE results by (significant) wins/losses vs. baseline scenario

| Strategy | RF (Max. Wins = 60) | | SVR (Max. Wins = 90) | | XGBoost (Max. Wins = 60) | |
|---|---|---|---|---|---|---|
| | Win | Loss | Win | Loss | Win | Loss |
| RU — v1 | 0(0) | 60(32) | 16(2) | 74(43) | 0(0) | 60(38) |
| RU — v2 | 0(0) | 60(36) | 8(0) | 82(49) | 0(0) | 60(40) |
| RO — v1 | 39(13) | 21(0) | 65(36) | 25(3) | 18(1) | 42(13) |
| RO — v2 | 35(12) | 25(1) | 62(29) | 28(6) | 22(1) | 38(14) |
| WERCS — v1 | 31(12) | 29(8) | 60(37) | 30(8) | 20(4) | 40(25) |
| WERCS — v2 | 34(15) | 26(8) | 61(42) | 29(5) | 12(2) | 48(19) |
| WERCS — v3 | 28(8) | 32(16) | 55(28) | 35(12) | 6(0) | 54(28) |
| WERCS — v4 | 32(9) | 28(13) | 54(28) | 36(7) | 8(0) | 52(26) |
| GN — v1 | 18(6) | 42(21) | 50(20) | 40(15) | 8(3) | 52(22) |
| GN — v2 | 15(7) | 45(18) | 48(21) | 42(20) | 12(2) | 48(22) |
| GN — v3 | 19(5) | 41(20) | 45(21) | 45(18) | 11(2) | 49(21) |
| GN — v4 | 4(0) | 56(32) | 10(6) | 80(44) | 0(0) | 60(40) |
| GN — v5 | 4(0) | 56(33) | 8(6) | 82(47) | 0(0) | 60(39) |
| GN — v6 | 4(0) | 56(35) | 8(5) | 82(50) | 0(0) | 60(39) |
| SMOTER — v1 | 28(7) | 32(8) | 55(34) | 35(10) | 8(0) | 52(22) |
| SMOTER — v2 | 24(9) | 36(12) | 47(28) | 43(16) | 7(0) | 53(23) |
| SMOGN — v1 | 11(0) | 49(16) | 44(23) | 46(18) | 5(0) | 55(26) |
| SMOGN — v2 | 9(0) | 51(14) | 44(22) | 46(19) | 6(0) | 54(26) |
| SMOGN — v3 | 11(0) | 49(14) | 44(20) | 46(20) | 4(0) | 56(25) |
| SMOGN — v4 | 22(11) | 38(9) | 48(28) | 42(13) | 8(0) | 52(20) |
| SMOGN — v5 | 22(9) | 38(5) | 48(27) | 42(13) | 5(0) | 55(19) |
| SMOGN — v6 | 22(8) | 38(5) | 48(28) | 42(13) | 8(0) | 52(23) |
| WSMOTER — v1 | 59(25) | 1(0) | 83(45) | 7(2) | 41(4) | 19(5) |
| WSMOTER — v2 | 59(30) | 1(0) | 81(45) | 9(1) | 43(11) | 17(5) |
| WSMOTER — v3 | 59(24) | 1(0) | 80(46) | 10(2) | 40(8) | 20(4) |
| WSMOTER — v4 | 57(31) | 3(0) | 80(44) | 10(0) | 42(6) | 18(4) |
| G-SMOTER — v1 | 49(12) | 11(0) | 80(46) | 10(0) | 41(1) | 19(4) |
| G-SMOTER — v2 | 45(16) | 15(0) | 80(46) | 10(0) | 43(1) | 17(4) |
| G-SMOTER — v3 | 45(13) | 15(0) | 69(36) | 21(10) | 28(5) | 32(7) |
| G-SMOTER — v4 | 46(12) | 14(1) | 69(36) | 21(8) | 32(2) | 28(7) |
| G-SMOTER — v5 | 48(20) | 12(0) | 85(45) | 5(0) | 41(1) | 19(7) |
| G-SMOTER — v6 | 52(18) | 8(0) | 85(45) | 5(0) | 37(4) | 23(5) |
| DAVID — v1 | 11(0) | 49(29) | 16(4) | 68(30) | 0(0) | 56(22) |
| DAVID — v2 | 4(0) | 56(35) | 5(2) | 79(42) | 0(0) | 56(32) |
| KNNOR-REG — None | 48(15) | 12(1) | 73(40) | 17(3) | 32(4) | 28(5) |
| CARTGen-IR — v1 | 29(9) | 31(5) | 57(31) | 33(18) | 23(1) | 37(14) |
| CARTGen-IR — v2 | 37(10) | 23(1) | 56(35) | 34(17) | 21(2) | 39(16) |
| CARTGen-IR — v3 | 38(10) | 22(0) | 55(31) | 35(17) | 21(1) | 39(14) |
| CARTGen-IR — v4 | 47(13) | 13(0) | 56(36) | 34(17) | 21(4) | 39(12) |
| CARTGen-IR — v5 | 43(14) | 17(0) | 54(30) | 36(13) | 15(0) | 45(19) |
| CARTGen-IR — v6 | 44(14) | 16(1) | 57(34) | 33(11) | 15(1) | 45(19) |
| CARTGen-IR — v7 | 43(15) | 17(1) | 55(30) | 35(13) | 13(2) | 47(16) |
| CARTGen-IR — v8 | 45(19) | 15(1) | 58(34) | 32(9) | 12(1) | 48(13) |
| CARTGen-IR — v9 | 39(10) | 21(2) | 53(29) | 37(14) | 20(1) | 40(11) |
| CARTGen-IR — v10 | 37(10) | 23(0) | 50(30) | 40(10) | 17(1) | 43(14) |
| CARTGen-IR — v11 | 45(12) | 15(0) | 52(28) | 38(12) | 19(2) | 41(11) |
| CARTGen-IR — v12 | 47(14) | 13(0) | 50(31) | 40(11) | 18(1) | 42(12) |
| CARTGen-IR — v13 | 47(15) | 13(0) | 60(37) | 30(12) | 20(2) | 40(17) |
| CARTGen-IR — v14 | 44(15) | 16(0) | 59(35) | 31(13) | 23(3) | 37(14) |
| CARTGen-IR — v15 | 48(19) | 12(0) | 59(37) | 31(14) | 23(2) | 37(18) |
| CARTGen-IR — v16 | 50(21) | 10(0) | 59(38) | 31(11) | 20(2) | 40(11) |
| CARTGen-IR — v17 | 36(12) | 24(0) | 53(30) | 37(12) | 16(0) | 44(20) |
| CARTGen-IR — v18 | 33(12) | 27(0) | 49(30) | 41(14) | 10(0) | 50(19) |
| CARTGen-IR — v19 | 42(20) | 18(1) | 52(29) | 38(13) | 14(1) | 46(11) |
| CARTGen-IR — v20 | 40(17) | 20(1) | 52(29) | 38(13) | 16(0) | 44(19) |
| CARTGen-IR — v21 | 40(14) | 20(0) | 52(32) | 38(11) | 17(4) | 43(13) |
| CARTGen-IR — v22 | 42(16) | 18(0) | 52(37) | 38(12) | 14(5) | 46(12) |
| CARTGen-IR — v23 | 47(19) | 13(0) | 56(31) | 34(14) | 22(3) | 38(11) |
| CARTGen-IR — v24 | 48(17) | 12(0) | 55(34) | 35(11) | 18(7) | 42(12) |
| CARTGen-IR — v25 | 44(15) | 16(1) | 63(38) | 27(11) | 20(1) | 40(16) |
| CARTGen-IR — v26 | 41(17) | 19(0) | 63(39) | 27(9) | 20(4) | 40(15) |
| CARTGen-IR — v27 | 53(19) | 7(0) | 66(38) | 24(10) | 16(3) | 44(12) |
| CARTGen-IR — v28 | 47(22) | 13(0) | 63(38) | 27(9) | 25(4) | 35(13) |
| CARTGen-IR — v29 | 37(10) | 23(0) | 54(24) | 36(13) | 12(0) | 48(20) |
| CARTGen-IR — v30 | 38(6) | 22(0) | 50(24) | 40(12) | 12(0) | 48(24) |
| CARTGen-IR — v31 | 42(13) | 18(0) | 55(24) | 35(11) | 19(0) | 41(17) |
| CARTGen-IR — v32 | 43(12) | 17(0) | 50(23) | 40(12) | 17(0) | 43(19) |
| CARTGen-IR — v33 | 40(13) | 20(0) | 56(35) | 34(12) | 17(6) | 43(11) |
| CARTGen-IR — v34 | 45(18) | 15(1) | 54(35) | 36(10) | 16(3) | 44(12) |
| CARTGen-IR — v35 | 47(19) | 13(0) | 53(37) | 37(10) | 15(6) | 45(9) |
| CARTGen-IR — v36 | 47(19) | 13(0) | 54(36) | 36(8) | 18(5) | 42(14) |

Table 17: DW-SERA results by (significant) wins/losses vs. baseline scenario

| Strategy | RF (Max. Wins = 60) | | SVR (Max. Wins = 90) | | XGBoost (Max. Wins = 60) | |
|---|---|---|---|---|---|---|
| | Win | Loss | Win | Loss | Win | Loss |
| RU — v1 | 0(0) | 60(60) | 12(4) | 78(74) | 0(0) | 60(60) |
| RU — v2 | 0(0) | 60(60) | 2(0) | 88(86) | 0(0) | 60(60) |
| RO — v1 | 29(7) | 31(4) | 58(39) | 32(17) | 19(0) | 41(18) |
| RO — v2 | 25(7) | 35(6) | 50(27) | 40(23) | 18(1) | 42(20) |
| WERCS — v1 | 22(5) | 38(27) | 48(36) | 42(24) | 6(1) | 54(38) |
| WERCS — v2 | 20(8) | 40(21) | 50(34) | 40(26) | 6(0) | 54(41) |
| WERCS — v3 | 0(0) | 60(37) | 38(19) | 52(39) | 2(0) | 58(51) |
| WERCS — v4 | 3(0) | 57(28) | 34(24) | 56(36) | 2(0) | 58(52) |
| GN — v1 | 5(0) | 55(40) | 29(17) | 61(43) | 2(0) | 58(40) |
| GN — v2 | 7(0) | 53(41) | 27(16) | 63(45) | 2(0) | 58(40) |
| GN — v3 | 3(0) | 57(38) | 25(14) | 65(47) | 2(0) | 58(40) |
| GN — v4 | 0(0) | 60(56) | 6(2) | 84(80) | 0(0) | 60(60) |
| GN — v5 | 0(0) | 60(56) | 6(2) | 84(81) | 0(0) | 60(60) |
| GN — v6 | 0(0) | 60(56) | 4(2) | 86(82) | 0(0) | 60(60) |
| SMOTER — v1 | 25(6) | 35(24) | 52(35) | 38(24) | 2(0) | 58(32) |
| SMOTER — v2 | 22(6) | 38(27) | 42(25) | 48(31) | 4(0) | 56(35) |
| SMOGN — v1 | 1(0) | 59(31) | 32(14) | 58(48) | 1(0) | 59(42) |
| SMOGN — v2 | 3(0) | 57(33) | 32(14) | 58(48) | 2(0) | 58(42) |
| SMOGN — v3 | 2(0) | 58(35) | 32(16) | 58(48) | 1(0) | 59(45) |
| SMOGN — v4 | 20(8) | 40(23) | 37(24) | 53(32) | 7(1) | 53(29) |
| SMOGN — v5 | 20(6) | 40(22) | 37(24) | 53(32) | 6(0) | 54(31) |
| SMOGN — v6 | 20(7) | 40(22) | 39(24) | 51(31) | 5(1) | 55(36) |
| WSMOTER — v1 | 51(27) | 9(0) | 82(65) | 8(0) | 33(6) | 27(6) |
| WSMOTER — v2 | 50(38) | 10(0) | 79(67) | 11(1) | 41(10) | 19(6) |
| WSMOTER — v3 | 47(31) | 13(0) | 78(66) | 12(2) | 39(6) | 21(5) |
| WSMOTER — v4 | 50(37) | 10(0) | 82(62) | 8(0) | 40(9) | 20(5) |
| G-SMOTER — v1 | 42(14) | 18(1) | 81(61) | 9(1) | 36(1) | 24(4) |
| G-SMOTER — v2 | 42(17) | 18(0) | 82(60) | 8(1) | 36(2) | 24(4) |
| G-SMOTER — v3 | 39(13) | 21(3) | 70(52) | 20(14) | 25(8) | 35(7) |
| G-SMOTER — v4 | 40(14) | 20(2) | 70(52) | 20(14) | 29(2) | 31(7) |
| G-SMOTER — v5 | 44(16) | 16(1) | 86(57) | 4(1) | 36(6) | 24(6) |
| G-SMOTER — v6 | 47(20) | 13(0) | 85(58) | 5(1) | 37(7) | 23(6) |
| DAVID — v1 | 3(0) | 57(45) | 10(4) | 74(58) | 0(0) | 56(44) |
| DAVID — v2 | 4(0) | 56(55) | 2(2) | 82(72) | 0(0) | 56(50) |
| KNNOR-REG — None | 43(11) | 17(1) | 64(54) | 26(4) | 25(3) | 35(6) |
| CARTGen-IR — v1 | 23(9) | 37(13) | 54(40) | 36(30) | 22(4) | 38(24) |
| CARTGen-IR — v2 | 26(8) | 34(13) | 54(43) | 36(29) | 19(3) | 41(25) |
| CARTGen-IR — v3 | 30(8) | 30(11) | 52(41) | 38(32) | 20(4) | 40(25) |
| CARTGen-IR — v4 | 36(15) | 24(8) | 54(41) | 36(32) | 18(4) | 42(23) |
| CARTGen-IR — v5 | 31(8) | 29(9) | 51(39) | 39(31) | 11(2) | 49(31) |
| CARTGen-IR — v6 | 29(8) | 31(9) | 51(39) | 39(29) | 11(2) | 49(32) |
| CARTGen-IR — v7 | 33(11) | 27(9) | 50(39) | 40(29) | 11(2) | 49(25) |
| CARTGen-IR — v8 | 34(15) | 26(10) | 49(39) | 41(25) | 13(2) | 47(23) |
| CARTGen-IR — v9 | 28(10) | 32(8) | 47(38) | 43(29) | 17(3) | 43(19) |
| CARTGen-IR — v10 | 27(9) | 33(9) | 47(34) | 43(28) | 14(2) | 46(21) |
| CARTGen-IR — v11 | 31(15) | 29(6) | 44(38) | 46(28) | 19(1) | 41(18) |
| CARTGen-IR — v12 | 33(15) | 27(6) | 48(36) | 42(30) | 17(3) | 43(18) |
| CARTGen-IR — v13 | 33(13) | 27(8) | 55(46) | 35(24) | 16(2) | 44(27) |
| CARTGen-IR — v14 | 33(15) | 27(9) | 55(44) | 35(26) | 16(3) | 44(25) |
| CARTGen-IR — v15 | 40(16) | 20(7) | 56(49) | 34(26) | 20(6) | 40(26) |
| CARTGen-IR — v16 | 38(15) | 22(6) | 55(47) | 35(24) | 19(3) | 41(21) |
| CARTGen-IR — v17 | 25(6) | 35(10) | 47(38) | 43(25) | 14(1) | 46(30) |
| CARTGen-IR — v18 | 21(7) | 39(14) | 44(39) | 46(29) | 12(1) | 48(34) |
| CARTGen-IR — v19 | 30(14) | 30(7) | 48(39) | 42(28) | 17(1) | 43(21) |
| CARTGen-IR — v20 | 29(14) | 31(12) | 45(37) | 45(23) | 16(1) | 44(30) |
| CARTGen-IR — v21 | 29(10) | 31(6) | 50(40) | 40(30) | 17(0) | 43(19) |
| CARTGen-IR — v22 | 31(11) | 29(11) | 51(40) | 39(32) | 13(3) | 47(18) |
| CARTGen-IR — v23 | 33(17) | 27(7) | 48(41) | 42(29) | 19(2) | 41(15) |
| CARTGen-IR — v24 | 36(17) | 24(9) | 51(38) | 39(28) | 17(4) | 43(17) |
| CARTGen-IR — v25 | 29(10) | 31(6) | 60(49) | 30(20) | 14(2) | 46(26) |
| CARTGen-IR — v26 | 29(9) | 31(9) | 58(46) | 32(21) | 15(2) | 45(26) |
| CARTGen-IR — v27 | 38(13) | 22(8) | 58(50) | 32(19) | 15(5) | 45(22) |
| CARTGen-IR — v28 | 37(18) | 23(6) | 61(49) | 29(20) | 17(3) | 43(24) |
| CARTGen-IR — v29 | 27(7) | 33(10) | 49(34) | 41(28) | 11(1) | 49(33) |
| CARTGen-IR — v30 | 22(5) | 38(9) | 47(32) | 43(30) | 7(1) | 53(33) |
| CARTGen-IR — v31 | 29(13) | 31(9) | 44(33) | 46(28) | 18(1) | 42(26) |
| CARTGen-IR — v32 | 27(12) | 33(6) | 48(28) | 42(29) | 17(1) | 43(28) |
| CARTGen-IR — v33 | 30(12) | 30(10) | 51(41) | 39(30) | 13(1) | 47(18) |
| CARTGen-IR — v34 | 31(6) | 29(8) | 50(35) | 40(30) | 12(2) | 48(21) |
| CARTGen-IR — v35 | 32(18) | 28(8) | 50(41) | 40(30) | 17(3) | 43(14) |
| CARTGen-IR — v36 | 37(16) | 23(9) | 54(35) | 36(28) | 19(4) | 41(18) |

### A.1.4 FRIEDMAN/POST-HOC NEMENYI TEST RESULTS AND CRITICAL DIFFERENCE DIAGRAMS

Table 18: Friedman test statistics per metric and learner, with statistically significant p-values ($p < 0.05$) highlighted

| Metric | Learner | Friedman Statistic | p-value | Significance |
|--------|---------|-------------------|---------|--------------|
| RMSE | RF | 123.1373 | 4.2e-21 | True |
| | SVR | 106.6007 | 8.7e-18 | True |
| | XGBoost | 135.0944 | 1.6e-23 | True |
| RW-RMSE | RF | 86.3691 | 8.5e-14 | True |
| | SVR | 75.6623 | 1.0e-11 | True |
| | XGBoost | 104.0617 | 2.8e-17 | True |
| SERA | RF | 88.2847 | 3.6e-14 | True |
| | SVR | 70.9370 | 8.1e-11 | True |
| | XGBoost | 103.3661 | 3.8e-17 | True |
| DW-RMSE | RF | 95.7860 | 1.2e-15 | True |
| | SVR | 84.0775 | 2.4e-13 | True |
| | XGBoost | 106.9165 | 7.6e-18 | True |
| DW-SERA | RF | 109.4205 | 2.4e-18 | True |
| | SVR | 92.3933 | 5.7e-15 | True |
| | XGBoost | 125.1599 | 1.7e-21 | True |

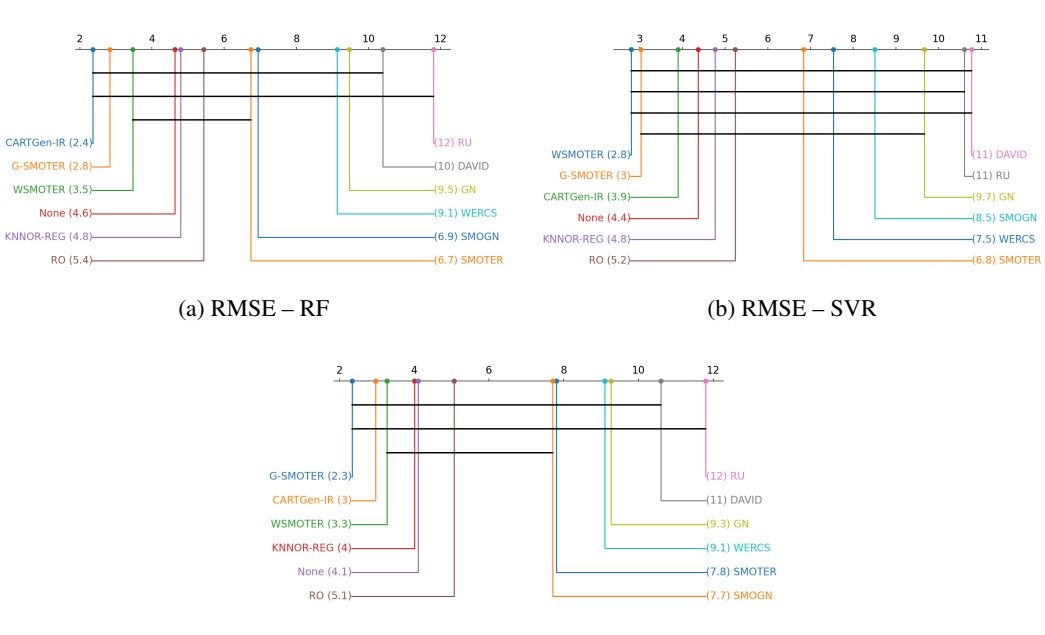

(a) RMSE – RF

(b) RMSE – SVR

(c) RMSE – XGBoost

Figure 9: Critical Difference diagrams for RMSE across learners

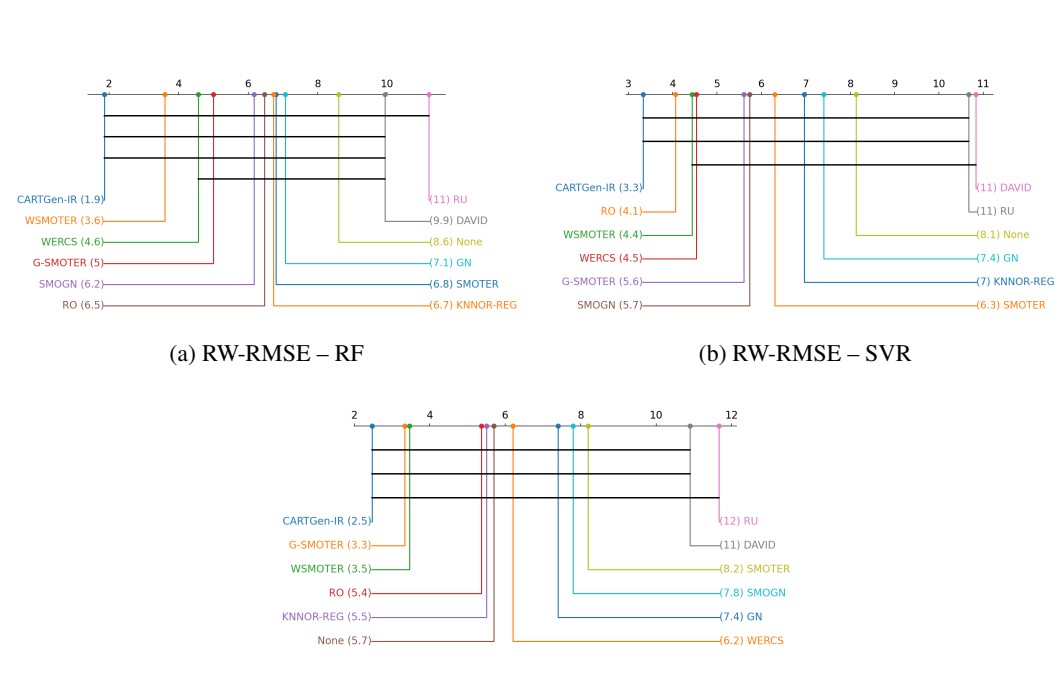

(a) RW-RMSE – RF

(b) RW-RMSE – SVR

(c) RW-RMSE – XGBoost

Figure 10: Critical Difference diagrams for RW-RMSE across learners

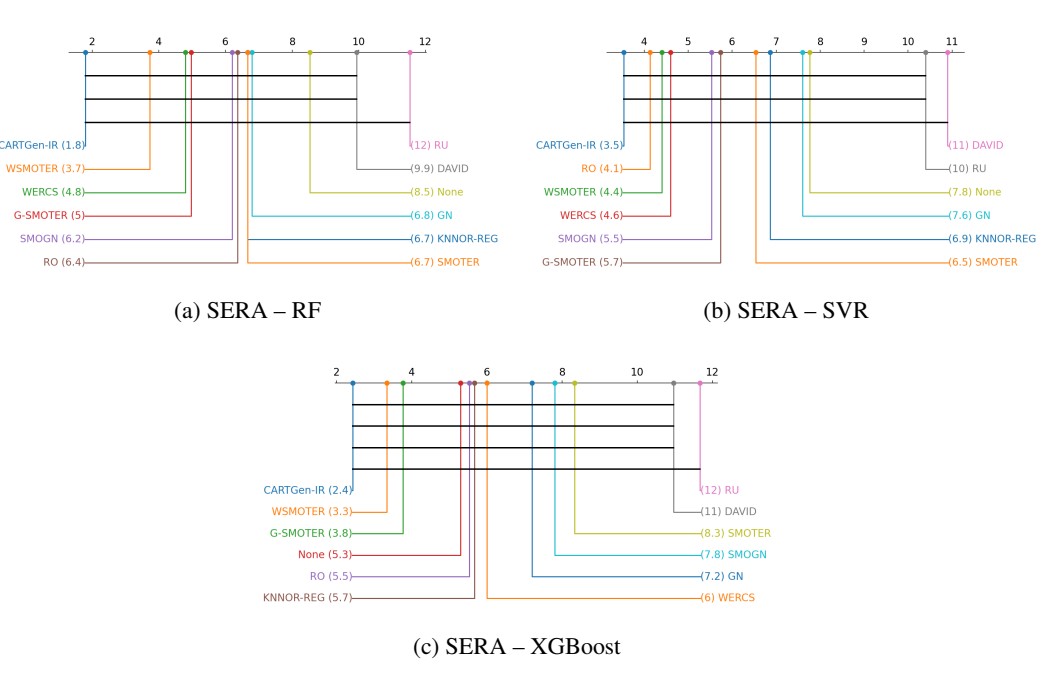

(a) SERA – RF

(b) SERA – SVR

(c) SERA – XGBoost

Figure 11: Critical Difference diagrams for SERA across learners

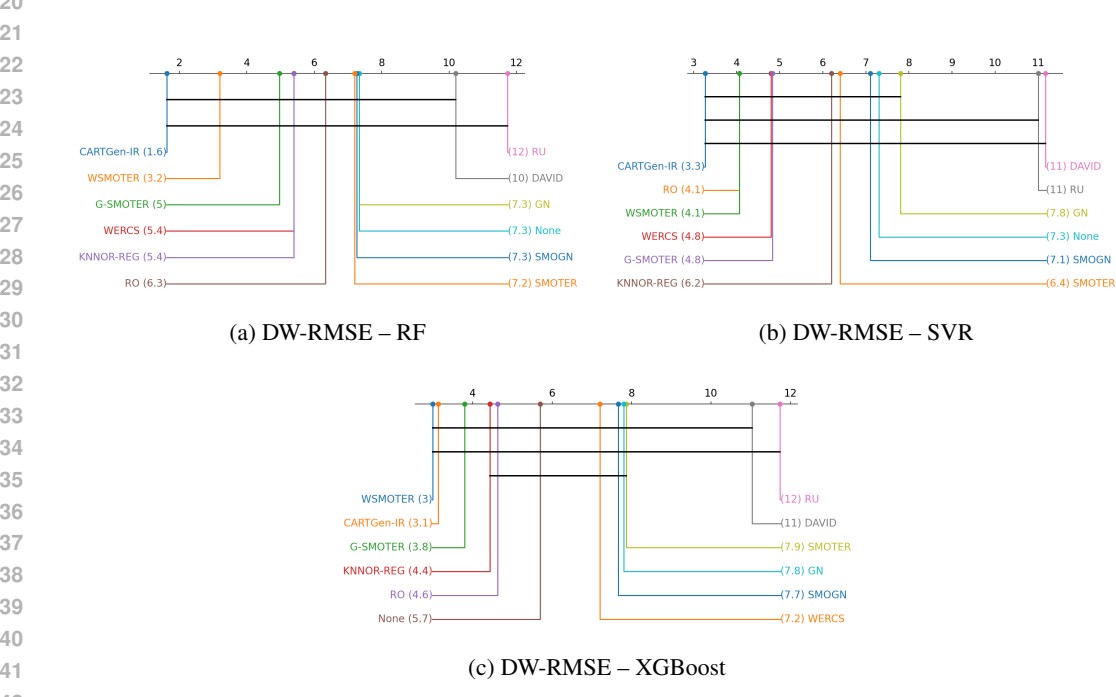

(a) DW-RMSE – RF

(b) DW-RMSE – SVR

(c) DW-RMSE – XGBoost

Figure 12: Critical Difference diagrams for DW-RMSE across learners

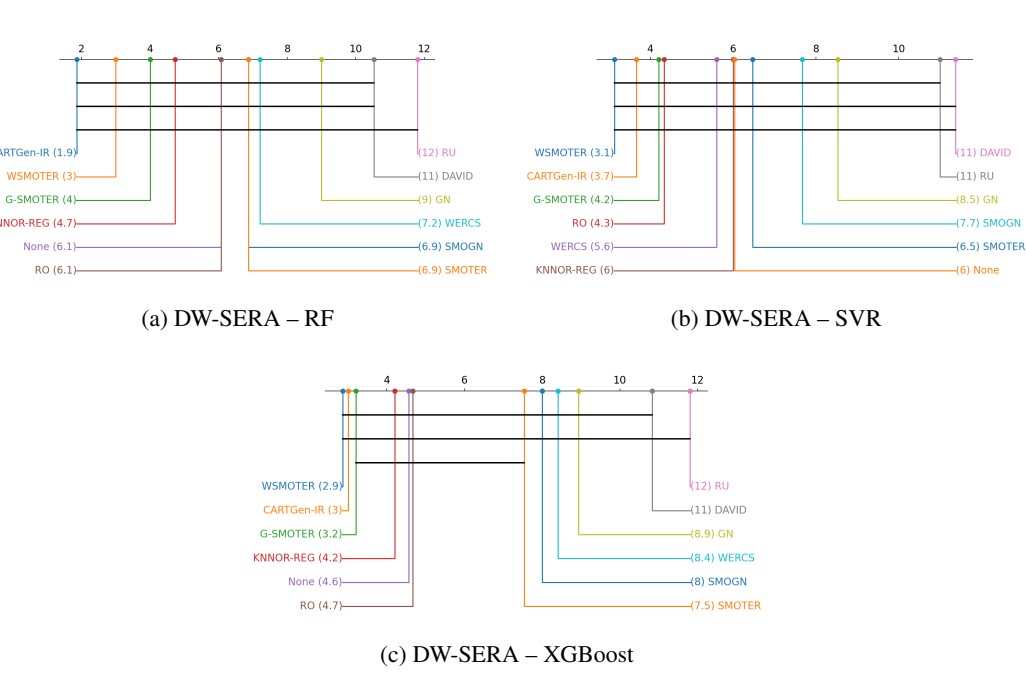

(a) DW-SERA – RF

(b) DW-SERA – SVR

(c) DW-SERA – XGBoost

Figure 13: Critical Difference diagrams for DW-SERA across learners

### A.1.5 BAYESIAN SIGNED-RANK TEST RESULTS

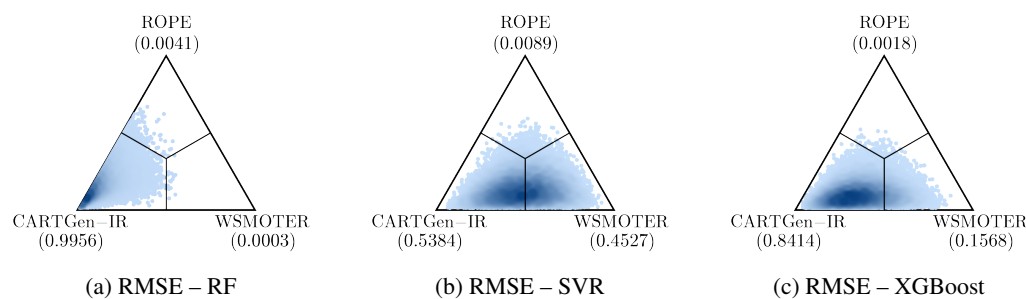

(a) RMSE – RF     (b) RMSE – SVR     (c) RMSE – XGBoost

Figure 14: Bayesian Posterior Ternary Plot for RMSE: CARTGen-IR vs. WSMOTER

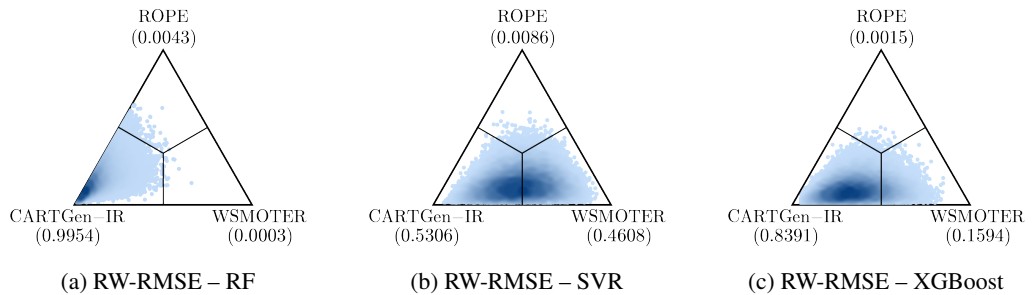

(a) RW-RMSE – RF    (b) RW-RMSE – SVR    (c) RW-RMSE – XGBoost

Figure 15: Bayesian Posterior Ternary Plot for RW-RMSE: CARTGen-IR vs. WSMOTER

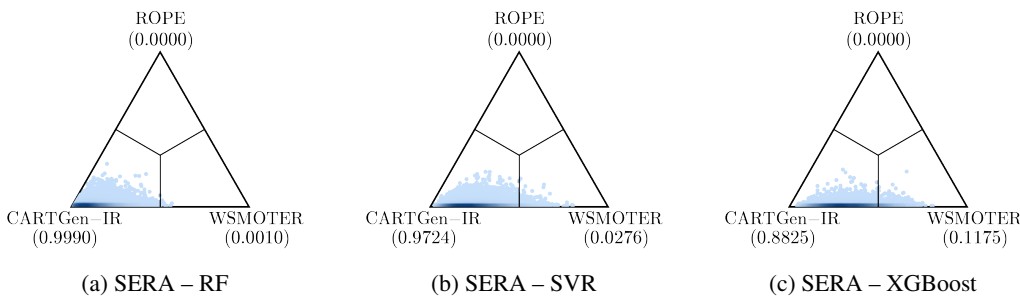

(a) SERA – RF     (b) SERA – SVR     (c) SERA – XGBoost

Figure 16: Bayesian Posterior Ternary Plot for SERA: CARTGen-IR vs. WSMOTER

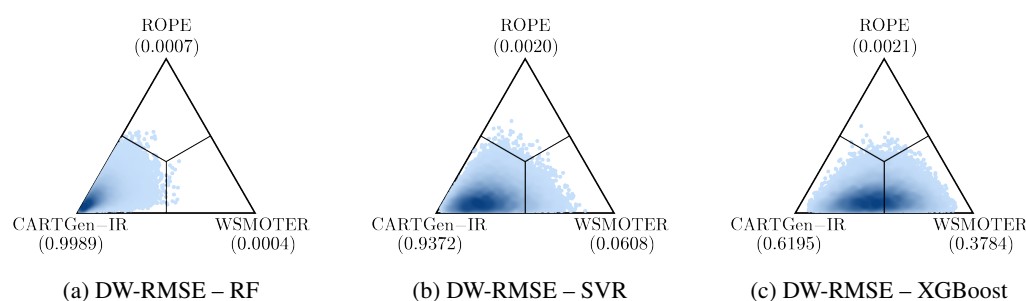

(a) DW-RMSE – RF  (b) DW-RMSE – SVR  (c) DW-RMSE – XGBoost

Figure 17: Bayesian Posterior Ternary Plot for DW-RMSE: CARTGen-IR vs. WSMOTER

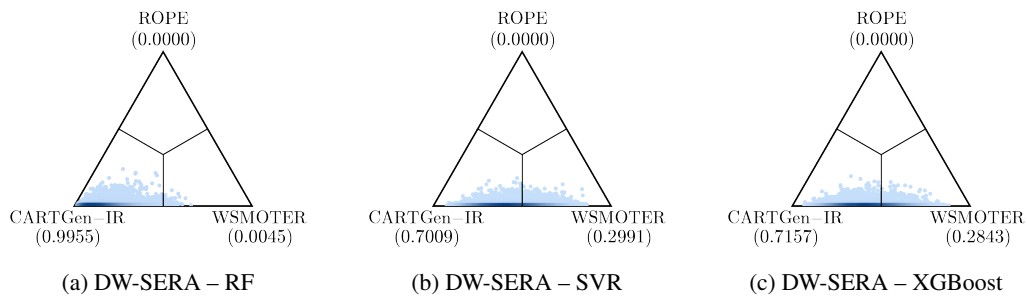

(a) DW-SERA – RF  (b) DW-SERA – SVR  (c) DW-SERA – XGBoost

Figure 18: Bayesian Posterior Ternary Plot for DW-SERA: CARTGen-IR vs. WSMOTER

Table 19: Bayesian Signed-Rank Test: CARTGen-IR vs. WSMOTER

| Metric | Learner | Bayesian Probabilities | | | Number of |
| | | $P$(CARTGen-IR > WSMOTER) | $P$(Equivalent) | $P$(WSMOTER > CARTGen-IR) | Paired Results |
| --- | --- | --- | --- | --- | --- |
| RMSE | RF | 0.9956 | 0.0041 | 0.0003 | 150 |
| | SVR | 0.5384 | 0.0089 | 0.4527 | 150 |
| | XGBoost | 0.8414 | 0.0018 | 0.1568 | 150 |
| RW-RMSE | RF | 0.9954 | 0.0043 | 0.0003 | 150 |
| | SVR | 0.5306 | 0.0086 | 0.4608 | 150 |
| | XGBoost | 0.8391 | 0.0015 | 0.1594 | 150 |
| SERA | RF | 0.9990 | 0.0000 | 0.0010 | 150 |
| | SVR | 0.9724 | 0.0000 | 0.0276 | 150 |
| | XGBoost | 0.8825 | 0.0000 | 0.1175 | 150 |
| DW-RMSE | RF | 0.9989 | 0.0007 | 0.0004 | 150 |
| | SVR | 0.9372 | 0.0020 | 0.0608 | 150 |
| | XGBoost | 0.6195 | 0.0021 | 0.3784 | 150 |
| DW-SERA | RF | 0.9955 | 0.0000 | 0.0045 | 150 |
| | SVR | 0.7009 | 0.0000 | 0.2991 | 150 |
| | XGBoost | 0.7157 | 0.0000 | 0.2843 | 150 |

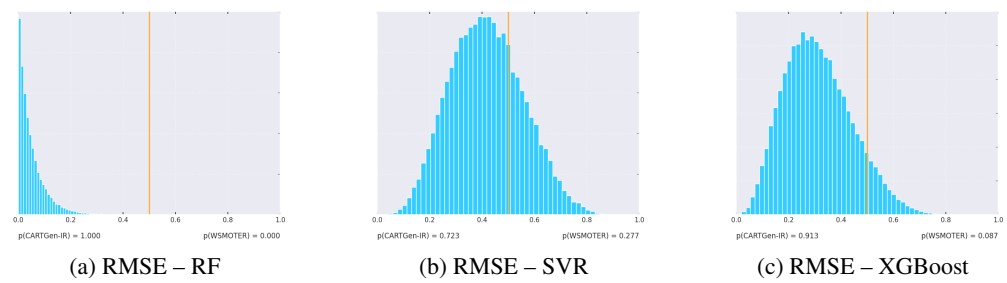

(a) RMSE – RF  (b) RMSE – SVR  (c) RMSE – XGBoost

Figure 19: Bayesian Posterior Distributions for RMSE: CARTGen-IR vs. WSMOTER

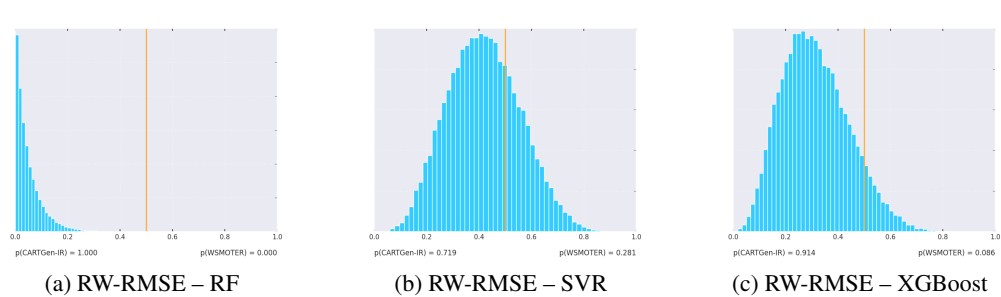

(a) RW-RMSE – RF        (b) RW-RMSE – SVR        (c) RW-RMSE – XGBoost

Figure 20: Bayesian Posterior Distributions for RW-RMSE: CARTGen-IR vs. WSMOTER

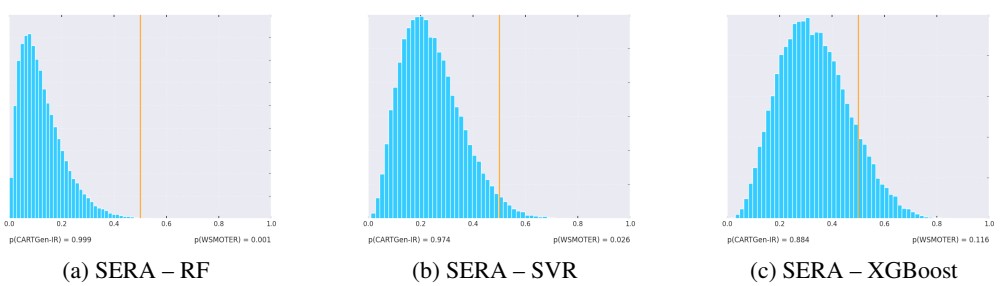

(a) SERA – RF        (b) SERA – SVR        (c) SERA – XGBoost

Figure 21: Bayesian Posterior Distributions for SERA: CARTGen-IR vs. WSMOTER

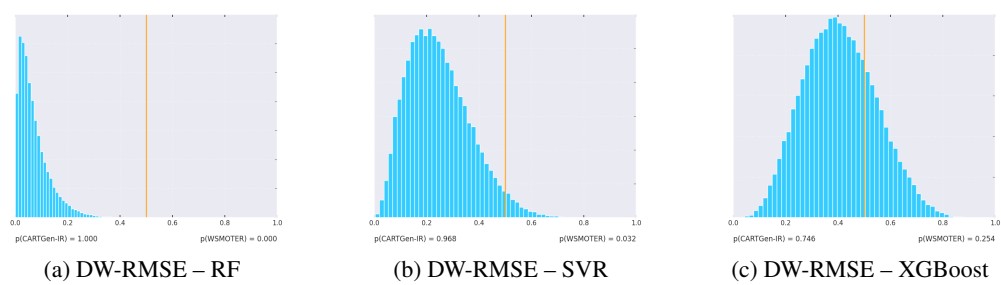

(a) DW-RMSE – RF        (b) DW-RMSE – SVR        (c) DW-RMSE – XGBoost

Figure 22: Bayesian Posterior Distributions for DW-RMSE: CARTGen-IR vs. WSMOTER

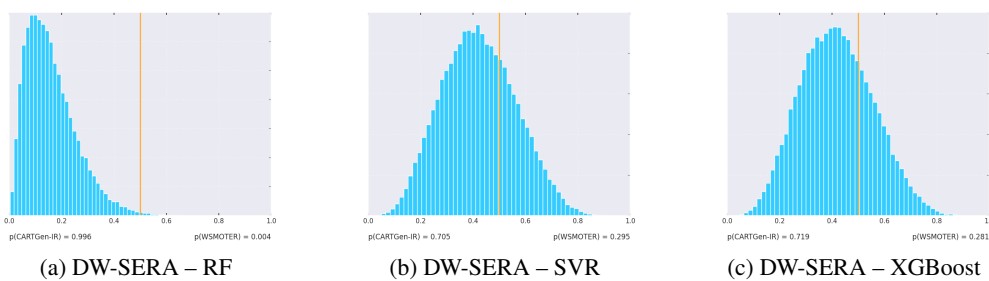

(a) DW-SERA – RF        (b) DW-SERA – SVR        (c) DW-SERA – XGBoost

Figure 23: Bayesian Posterior Distributions for DW-SERA: CARTGen-IR vs. WSMOTER

### A.1.6 RUNTIME RESULTS

Table 20: Runtime average (std. dev.) per preprocessing strategy and dataset

| Dataset | RU | RO | WERCS | GN | SMOTER | SMOGN | WSMOTER | G-SMOTER | DAVID | KNNOR-REG | CARTGen-IR |
|---|---|---|---|---|---|---|---|---|---|---|---|
| abalone | 0.1 (0) | 0.15 (0) | 0.003 (0) | 0.894 (0.002) | 20.576 (0.068) | 20.836 (0.064) | 0.396 (0.008) | 0.392 (0) | 90.638 (0.276) | 0.034 (0.003) | 0.262 (0.001) |
| airfoil | 0.022 (0) | 0.038 (0.001) | 0.001 (0) | 0.193 (0) | 0.416 (0.001) | 0.429 (0.001) | 0.123 (0.002) | 0.982 (0.002) | 10.372 (0.002) | 0.02 (0.001) | 0.093 (0.001) |
| availablePower | 0.043 (0) | 0.118 (0.001) | 0.002 (0) | 0.591 (0.003) | 4.175 (0.014) | 4.362 (0.003) | 0.406 (0.002) | 0.131 (0.001) | 16.357 (0.054) | 0.098 (0.002) | 0.122 (0.002) |
| cpuSm | 0.262 (0.001) | 0.481 (0.001) | 0.004 (0) | 3.361 (0.001) | 9.303 (0.082) | 9.341 (0.065) | 0.693 (0.004) | 0.655 (0.002) | 65.834 (0.004) | 0.039 (0.001) | 0.723 (0.003) |
| ele-1 | 0.006 (0) | 0.01 (0) | 0.001 (0) | 0.034 (0) | 0.075 (0) | 0.077 (0) | 0.047 (0.001) | 0.059 (0) | 0.333 (0.002) | 0.01 (0.001) | 0.053 (0.002) |
| forestFires | 0.016 (0) | 0.029 (0) | 0.001 (0) | 0.131 (0) | 0.13 (0.002) | 0.215 (0.001) | 0.245 (0) | 0.252 (0) | 5.758 (0.035) | 0.009 (0.001) | 0.08 (0) |
| fuelConsumption | 0.083 (0) | 0.282 (0.002) | 0.004 (0) | 1.499 (0.008) | 2.793 (0.008) | 3.704 (0.002) | 0.384 (0.007) | 0.032 (0.002) | 20.292 (0.323) | 0.015 (0.001) | 0.255 (0.002) |
| heat | 0.158 (0.002) | 0.387 (0.002) | 0.005 (0) | 2.421 (0) | 30.896 (0.041) | 31.38 (0.119) | 1.648 (0.003) | 0.484 (0.002) | 83.531 (0.081) | 0.094 (0.066) | 0.457 (0.001) |
| housingBoston | 0.015 (0) | 0.032 (0) | 0.001 (0) | 0.13 (0.001) | 0.391 (0.001) | 0.421 (0.002) | 0.044 (0.001) | 0.023 (0) | 6.482 (0.078) | 0.011 (0.001) | 0.087 (0.001) |
| maxTorque | 0.078 (0.001) | 0.25 (0) | 0.003 (0) | 1.235 (0.003) | 5.746 (0.008) | 6.593 (0.018) | 0.396 (0.003) | 0.38 (0.001) | 20.411 (0.121) | 0.111 (0.003) | 0.249 (0.002) |
| mortgage | 0.034 (0) | 0.058 (0.001) | 0.001 (0) | 0.324 (0.003) | 1.101 (0.003) | 1.116 (0.006) | 0.088 (0.001) | 0.093 (0) | 12.86 (0.125) | 0.016 (0) | 0.113 (0.002) |
| sensory | 0.012 (0.003) | 0.033 (0) | 0.001 (0) | 0.142 (0.001) | 0.25 (0.002) | 0.357 (0) | 0.073 (0.005) | 0.065 (0.001) | 5.227 (0.018) | 0.009 (0.001) | 0.079 (0) |
| servo | 0.006 (0) | 0.01 (0) | 0.001 (0) | 0.017 (0) | 0.1 (0) | 0.11 (0.004) | 0.021 (0.001) | 0.012 (0) | 4.895 (0.015) | 0.01 (0) | 0.053 (0) |
| strikes | 0.01 (0) | 0.02 (0) | 0.001 (0) | 0.086 (0) | 0.104 (0) | 0.144 (0) | 0.062 (0.001) | 0.064 (0) | 5.738 (0.005) | 0.01 (0.001) | 0.066 (0.003) |
| treasury | 0.047 (0.01) | 0.055 (0) | 0.001 (0) | 0.312 (0) | 1.154 (0.002) | 1.166 (0.002) | 0.089 (0.001) | 0.034 (0) | 12.427 (0.1) | 0.016 (0.001) | 0.056 (0.001) |
| Average | 0.059 | 0.130 | 0.002 | 0.758 | 5.147 | 5.350 | 0.314 | 0.244 | 24.077 | 0.034 | 0.183 |
| Std. Dev. | 0.001 | 0.001 | 0.000 | 0.002 | 0.016 | 0.019 | 0.003 | 0.001 | 0.083 | 0.005 | 0.001 |