# OpenReview forum: "CART-Based Synthetic Tabular Data Generation for Imbalanced Regression"
_ICLR.cc/2026/Conference — ICLR 2026 Conference Withdrawn Submission_

### Official Review · Reviewer_hVyD · 2025-10-27

**Soundness:** 3
**Presentation:** 2
**Contribution:** 1
**Rating:** 2
**Confidence:** 4

**Summary:**

This paper addresses the challenge of imbalanced regression in tabular data by introducing a CART-based method that generates synthetic data for rare target values. The authors demonstrate that their approach is highly competitive with state-of-the-art methods on relevance-aware metrics while offering greater computational efficiency than GAN/VAE-based approaches over 15  datasets.

**Strengths:**

- Authors tackle a problem that is VERY important but quite often neglected, as the imbalanced learning literature focuses more on classification than regression.
- Statistical analysis performed comparing methods is quite strong, incorporating ROPE (Region of Practical Equivalence).
- Reproducible manuscript. Datasets, hyperparameters used, and implementation are all presented in the paper or as supplementary material on the open-review system.

**Weaknesses:**

1) The organization needs significant improvement. The first 5 paragraphs of the proposed method sound more like a literature review. They just talk about how past authors used CART for synthetic generation, rather than focusing on what this paper brings to the table. This also raised a question about whether this work makes a significant contribution. Or it consist of just adapting existing approaches to the new setting? At the end, I just didn't see any new meaningful concepts and methods being developed here (more development on that in the other weaknesses, especially the last one).

2) The paper seems a bit outdated in its related work. Intro states: “Standard regression models assume uniform importance across target values and rely on metrics like Mean Squared Error (MSE), which favor average predictions and neglect extremes. This limitation is particularly relevant in domains such as extreme weather forecasting (Schultz et al., 2021), sea surface temperature prediction (Alerskans et al., 2022), oncology drug response (Lenhof et al., 2022), and financial anomaly detection.” But so many new metrics have been proposed like F1-Regression, SERA… Also, there is no development on the relevance of extreme values in these settings.

3) The problem statement and the contributions/value of this work is not clear. At the end of the introduction and related work, I was still not convinced whether the main challenges are found in current data generation or training schemes (like undersampling) for imbalanced regression, why we need a new method, and how the proposed CART-based method improves on it. Nor on the dangers/problems of “user-defined thresholds,” the authors claim several times that they’re solving. Also, given that [1] also proposed a CART model for generation, how does this paper differ from it.

4) The authors start the experiment protocol by talking about a 62 datasets repository. But later, it says only 15 is used. This confuses the paper by first giving the idea we will have a more robust experimental setting to reduce later. Why make this “advertising” regarding these if they are not studied at all in this paper?

5) Lines 253 to 256: “The key attributes of these datasets are summarized in Table 1, which also reports the absolute and relative frequencies of rare instances, defined according to a relevance threshold of 0.8.” First, why this setting at 0.8? Is there a theoretical justification? Also, this statement, in my opinion, contradicts one of the main claims that is stated several times in the paper and the weakness of the previous regarding “user-defined threshold”.

6) Authors consider a single approach based on generative models (DAVID) while 9 over-/and undersampling from different categories. Any justification for this imbalance in SOTA choice?

7) Would like to see an example with a toy/synthetic dataset contrasting the generation done by the proposed CARTGen-IR vs classical methods like SMOTER, etc. This can help show how the proposed method generates more informative samples.

8) The paper lacks a proper ablation study. The method is a pipeline of components (weighting -> resampling -> CART generation -> optional noise). A critical question is: which parts are most important? An ablation study (e.g., CART without weighting, weighting with simple oversampling) is missing. Another question, why CART, which produces axis splits instead of another model like an MLP? Several synthetic data generation uses it as a mapping function between feature interactions [2]

9) Future work statements are quite generic like “broader study” instead of being addressed to confront challenges in the method like its performance with non-extreme rare values, its scalability to very high-dimensional data (as one CART tree is fit per column) etc.

10) The paper introduces several auxiliary contributions only in the concluding paragraph, namely the adaptation of RMSE and SERA to use the DenseWeight mechanism (DW-RMSE, DW-SERA) and the release of a 62-dataset repository. Presenting these as key outcomes is problematic, as they receive no substantive development in the manuscript. The new metrics are not motivated or formally defined.

11)In the same vein as the previous comment, the conclusion also talks about the 62 datasets' contribution out of nowhere. The need for a new 62-dataset repository is not justified, especially considering existing larger public benchmarks (See: [3]) . At the end the work gives the impression of seeking to inflate its perceived contribution with elements that were not core to its research narrative (never mentioned in the intro, abstract etc) or properly validated.

12) The work by Yang et al [4] introduces a framework based on Label Distribution Smoothing (LDS) and Feature Distribution Smoothing (FDS). This approach directly addresses the core problem of imbalanced continuous targets by operating natively in the continuous space, using kernel smoothing to share information across nearby target values without ever requiring crisp partitions or thresholds. Even though it is not oversampling based, it should be discussed in the work as being another one not based on user-defined threshold as is one of the key arguments for the method proposed.

**Minor suggestion & typos:**

- The authors should since the beginning state their CARTGen-IR method (maybe even in the title) as it is a single method being proposed and not a group of cart-based ones. This is not even clear in the intro.
- Line 205: instances.The -> instances. The

**Bibliography:**

[1] Reiter, Jerome P. "Using CART to generate partially synthetic public use microdata." Journal of official statistics 21.3 (2005): 441.

[2] Hollmann, Noah, et al. "Accurate predictions on small data with a tabular foundation model." Nature 637.8045 (2025): 319-326.

[3] Avelino, Juscimara G., George DC Cavalcanti, and Rafael MO Cruz. "Imbalanced regression pipeline recommendation." Machine Learning 114.6 (2025): 1-48

[4] Yang, Yuzhe, et al. "Delving into deep imbalanced regression." International conference on machine learning. PMLR, 2021.

[5] Panagiotou, Emmanouil, Arjun Roy, and Eirini Ntoutsi. "Synthetic tabular data generation for class imbalance and fairness: A comparative study." arXiv preprint arXiv:2409.05215 (2024).

**Questions:**

This section uses the same bibliography as the weakness one.

- The Proposal section heavily reviews existing CART-based generation methods [1,5]. To clarify your contribution, please explicitly state: What is the specific novel component of CARTGen-IR that is not present in these prior works? Currently, the section conflates a literature review with a description of your novel contribution.

-  Include an ablation study quantifying the performance impact of the rarity weighting, the CART-based generation, and the noise injection separately. As the method pipeline has many components, why was it not in the first version?

- The conclusion lists the creation of a 62-dataset repository and new DW-RMSE/DW-SERA metrics as key contributions. As these are not discussed in the body of the paper, could you please either justify their necessity (e.g., over existing larger repositories [4]) and formally define the new metrics? Or just remove these as contributions to this paper.

- Could you provide a qualitative analysis or visualization on a toy dataset, showing how the samples generated by CARTGen-IR differ in the feature space from those generated by a baseline like SMOTER?

- The paper mentions a repository of 62 datasets, but experiments on only 15. Please explain the criteria used to select these 15 datasets and comment on whether the conclusions are likely to hold across the full repository.

- Your work argues against user-defined thresholds. Please discuss how your method relates to other threshold-free paradigms, such as the Label/Feature Distribution Smoothing framework from [4]

---

### Official Review · Reviewer_W6vm · 2025-10-28

**Soundness:** 3
**Presentation:** 3
**Contribution:** 1
**Rating:** 4
**Confidence:** 4

**Summary:**

The paper proposes a method for tabular data generation based on small Bayes'Nets that provide a numerical prediction of how  likely a variable is given others, and sampling from that distribution. The distribution is shaped by a rebalancing of training examples with respect to  learned weights/ importance coeffs that are standard in a re-weighting method for imbalanced classification.

A number od datasets form the evaluation suite, and the paper outperforms SMOTE and being based on trees, is computationally efficient.

**Strengths:**

. All parts constituting the method are those that should naturally have been in their role/s. Conceptual appeal.

**Weaknesses:**

. The paper would appear to represent a good preprocessing pipeline. It does not present a major advance in an area of immediate interest to the ICLR community

**Questions:**

. null

---

### Official Review · Reviewer_HfBX · 2025-10-28

**Soundness:** 2
**Presentation:** 2
**Contribution:** 2
**Rating:** 2
**Confidence:** 3

**Summary:**

This paper proposes CARTGen-IR, a column-wise CART–based data synthesis method for augmenting training data in imbalanced regression, i.e., settings that prioritize rare regions of a continuous target.
Its core novelty is a threshold-free treatment of the target, avoiding the target partitioning required by SMOTER/SMOGN-style methods. The evaluation spans 15 datasets, 3 learners, and 72 preprocessing configurations with 2×5 repeated stratified CV, reporting that CARTGen-IR is broadly superior or on par with alternatives. In terms of runtime, it is the second fastest among generative approaches and exhibits good stability (low standard deviation).

**Strengths:**

- Threshold-free target handling (KDE/DenseWeight/relevance) avoids ad-hoc discretization while remaining interpretable.
- Simple, column-wise CART synthesis: easy to implement, debug, and explain; decoupled from the downstream regressor.
- Heterogeneous datasets, $2\times 5$ repeated stratified CV, Friedman/Nemenyi, Wilcoxon, and Bayesian signed-rank analyses.
- Code and a curated repository; competitive runtime with small variance.

**Weaknesses:**

- The idea of column-wise synthesis with CART follows prior work (Panagiotou et al., 2024); this paper amounts to an adaptation that targets imbalanced regression and links threshold avoidance to weighting plus resampling.
- While DIRVAE/IRGAN are excluded due to unavailable code, the paper lacks practical baselines that apply general-purpose tabular generators (e.g., CTGAN/TVAE) to regression (synthesize → train existing regressors).
- Because the method trains CART models in proportion to the number of columns $p$, its computational and memory characteristics on large, high-dimensional datasets remain unevaluated.
- Only one augmentation-size pattern is used, making it unclear how the method’s effectiveness changes with the imbalance ratio.
- Several presentation issues remain. Please see the items below and revise where applicable.

**Questions:**

- Regarding thresholds, are there truly no other methods that do not require user-defined thresholds?
- While thresholds are indeed avoided, how sensitive is accuracy to $\alpha$ and $\delta$?
- Data augmentation seems feasible with models other than CART (e.g., Random Forest). Is there a compelling reason to insist on CART?
- In Algorithm 1, you add noise only to numeric features. Does this mean the method is inapplicable to datasets in which all features are categorical?
- For Table 4, please discuss which kinds of datasets the proposed method favors or excels on.
- In Figure 2, WSMOTER appears to have a higher proportion of significant wins and wins than the proposed method. How do you interpret this?

(Other Comment)
- Overall: It would be better to end displayed equations with a comma or period (, or .).
- Overall: Please add thousands separators (,) to numbers with three or more digits (especially in Table 1) for readability.
- Line 100: Notation for $t$ is missing.
- Figure 1 is too small and hard to read.
- Algorithm 1: Replace curly quotes ‘’ … ‘’ with TeX quotes `` … ''.
- Algorithm 1: There is no notation/specification for sample(A, B, C, D), CART(A, B), or CARTGen(A, B).
- Algorithm 1: DenseWeight appears in a different font from the surrounding text; please unify it.
- Algorithm 1: ws <- (1/exp() -> ws <- 1/exp() (remove the stray parenthesis).
- Table 1: You categorize Type of Extreme, but the criterion for this categorization is unclear.
- Figure 2: It is small and hard to read; a brief “how to read” explanation would also help.
- Figure 3: Similarly, a short guide on how to interpret the figure would be helpful.
- Line 451: KNORRR -> KNNOR?
- Please standardize the years and formatting in the references.
- It would be better to place the code link in a more prominent location within the main text.

---

### Official Review · Reviewer_kYqh · 2025-11-01

**Soundness:** 2
**Presentation:** 2
**Contribution:** 2
**Rating:** 2
**Confidence:** 4

**Summary:**

This study proposes a method to address data imbalance in regression tasks. For each instance, a rarity score, e.g., analogous to the inverse of data density, is computed. Based on this score, resampling is performed to preferentially select rarer instances. To avoid simple duplication, small perturbations are added as noise when necessary. Subsequently, regression trees are fitted by alternately using different target columns, and new data samples are generated from the resulting predictions.

**Strengths:**

- The paper addresses a practical and often overlooked issue in regression tasks, data imbalance, where some regions of the feature space are densely populated while others are sparsely represented.
- The experimental evaluation is comprehensive, covering various settings and demonstrating the results in various perspectives.

**Weaknesses:**

- The paper provides no theoretical justification for the proposed approach; it appears to be a purely heuristic method.
- All three techniques used to compute the rarity score are existing methods, offering little methodological novelty.
- There is no rationale provided for the parameter settings, such as why exactly five synthetic samples are generated.
- The core of the proposed approach is to alternately treat each variable as the target while using the others as inputs to fit regression trees, and then generate new data based on the fitted trees. However, this idea is very similar to techniques widely used in missing data imputation. Essentially, the method performs interpolation in regions with sparse data, which limits its significance as a novel contribution.
- The paper fails to provide mathematical definitions for key components such as KDE, relevance function, and DenseWeight.
- The justification that some comparison methods could not be reproduced due to unavailable source code is not acceptable; the authors should have re-implemented these methods to ensure a fair comparison.
- Figure 1 is difficult to read and does not effectively help the reader understand the proposed method.

**Questions:**

- Why must the model specifically use CART? Could other tree-based methods or other regression models be applied instead?
- In regression, data imbalance can arise not only from rare instances but also from overly dense regions that bias the model. Why does the proposed method focus solely on rare cases?
- The definition of the CARTGen() function is missing. Since the order in which variables are generated could affect the final synthesized data, wouldn’t the generation order impact the results? If so, which variable should be generated first, and how does this choice affect performance?
- The abstract claims that the proposed approach provides __greater transparency__. What exactly does this transparency mean in this context?

---

### Note · Authors · 2025-11-13

I have read and agree with the venue's withdrawal policy on behalf of myself and my co-authors.